# A Distributional Approach to Controlled Text Generation

**Muhammad Khalifa** * †
Cairo University

**Hady Elsahar**\*
Naver Labs Europe

**Marc Dymetman**\*
Naver Labs Europe

`m.khalifa@grad.fci-cu.edu.eg`

`{hady.elsahar,marc.dymetman}@naverlabs.com`

## Abstract

We propose a Distributional Approach for addressing Controlled Text Generation from pre-trained Language Models (LMs). This approach permits to specify, in a single formal framework, both "pointwise'" and "distributional" constraints over the target LM — to our knowledge, the first model with such generality — while minimizing KL divergence from the initial LM distribution. The optimal target distribution is then uniquely determined as an explicit EBM (Energy-Based Model) representation. From that optimal representation we then train a target controlled Autoregressive LM through an adaptive distributional variant of Policy Gradient. We conduct a first set of experiments over pointwise constraints showing the advantages of our approach over a set of baselines, in terms of obtaining a controlled LM balancing constraint satisfaction with divergence from the initial LM. We then perform experiments over distributional constraints, a unique feature of our approach, demonstrating its potential as a remedy to the problem of Bias in Language Models. Through an ablation study, we show the effectiveness of our adaptive technique for obtaining faster convergence.[1]

## 1 Introduction

Neural language models, such as GPT-2/3 (Radford et al., 2019; Brown et al., 2020a), pretrained on huge amounts of text, have become pre-eminent in NLP, producing texts of unprecedented quality. In this paper, we are concerned with the problem of controlling a generic pretrained LM in order to satisfy certain desiderata. For instance, we may want to avoid toxic content; prevent certain demographic biases; or steer generations towards a certain topic or style. Prior work, taking inspiration from Reinforcement Learning (RL), has aimed at inducing autoregressive models to optimize global objectives using task specific rewards such as BLEU and ROUGE for Machine Translation and Summarization (Ranzato et al., 2016; Bahdanau et al., 2017), or hand crafted rewards (Li et al., 2016b; Tambwekar et al., 2019) to improve certain a priori desirable features.

However, such an optimization process is not infallible; Liu et al. (2016a) noted that it often leads to "degeneration", producing poor examples that improve the average reward but forgo coherence and fluency. This degeneration is often diagnosed as an effect of deviating too much from the original pretrained LM during optimization. Consequently, prior work has regarded proximity to the pretrained model as a prescription for sample quality. This view is most prominent in open-domain generation where no gold references are available for fine-tuning, making the pretrained LM itself the yardstick for fluency. Jaques et al. (2017); Ziegler et al. (2019) propose a conservative fine-tuning approach moderated by a KL penalty between the trained policy and the original LM, discouraging large deviations. A KL penalty was also used by Dathathri et al. (2020), this time in a plug-and-play rather than a fine-tuning context. However, the authors show that balancing policy deviations from the original LM while also satisfying the control conditions is delicate. To combat degeneration they had to combine the KL penalty with post-norm fusion, reranking, and early-stopping procedures.

---

\*Equal Contributions.

†Work done during an internship at NAVER Labs Europe.

[1]Code available on `https://github.com/naver/gdc`

Most of the existing work on Controlled Generation has taken what we refer to as a "pointwise" view, namely focusing on the quality of each *individual* output, a view that is encouraged by the standard RL goal of maximizing rewards computed at the individual level. Such techniques are incapable of enforcing "distributional" conditions, where some collective statistical properties are desired over the set of *all* generations.

Distributional control is key to solving the problem of social biases in LMs trained on large, uncurated Web corpora. Those LMs - dubbed *"Stochastic Parrots"* in (Bender et al., 2021) - tend to encode hegemonic biases that are harmful to marginalized populations. There has been a large body of work analysing these distributional biases (Blodgett et al., 2020; Stanovsky et al., 2019; Prates et al., 2020; Sheng et al., 2019a; Brown et al., 2020b). However, applying distributional control on pretrained models is still an understudied problem. Sheng et al. (2020) introduce a method relying on adversarial triggers (Wallace et al., 2019); this method does not de-bias the whole distribution but only obtains non-biased continuations of given prompts. Bordia & Bowman (2019) introduce a regularization term for reducing gender bias when training a language model from scratch (as opposed to de-biasing a pretrained model).[2]

In this work, we present our *Generation with Distributional Control* (GDC) approach, in which we formalize the problem of controlled text generation as a *constraint satisfaction* problem over the *probability distribution* $p$ representing the desired target LM. Namely, we require the expectations ("moments") relative to $p$ of certain output features to have specific values; this permits for instance to condition all outputs to speak about sports (a *pointwise constraint*), and 50% of them to mention female characters (a *distributional constraint*). Additionally, we require $p$ to have a *minimal KL divergence* $D_{\mathrm{KL}}(p, a)$ from the original pretrained LM $a$. This has the effect that $p$ now inherits favorable linguistic qualities from $a$. As we will explain, this formulation is a generalization of the *Maximum Entropy Principle* and leads to a unique solution $P(x)$. $P(x)$ is an unnormalized distribution, aka an *Energy-Based Model* (EBM) (Hinton, 2002; LeCun et al., 2006; Bakhtin et al., 2020), of which $p(x) = 1/Z\ P(x)$ is the normalized version, where $Z \doteq \sum_x P(x)$ is the partition function of $P$.

Computing the EBM representation $P$ is a crucial step, as it fully determines the *optimal* distribution $p$ we are looking for. However, it is not the end of the story, because the representation thus obtained does not enable us to directly *sample* from $p$, an essential property of any LM.[3] To this end, we introduce *KL-adaptive DPG (Distributional Policy Gradient)*, a variant of an algorithm recently proposed in (Parshakova et al., 2019b). We train the policy $\pi_\theta$ to approximate $p$ in an adaptive way, by speeding up the next round of approximations based on approximations previously obtained. At the end of this process, we obtain a final $\pi_\theta$, our target LM, on which we can estimate diverse metrics, including $D_{\mathrm{KL}}(p, \pi_\theta)$, measuring the approximation quality of $\pi_\theta$ relative to the optimal $p$, and $D_{\mathrm{KL}}(\pi_\theta, a)$, measuring the divergence of $\pi_\theta$ relative to the original LM $a$.

This two-step approach differs from much research in NLP-oriented work with EBMs, which tends to use EBM representations *inside* the training loops of neural networks, blurring different dimensions of the problem. By contrast — similarly to Parshakova et al. (2019a;b) in a different context — we clearly *decouple* the relatively simple problem of determining a "pivot" optimal EBM from the more difficult problem of exploiting this EBM at inference time, Such decoupling is valuable, because it permits to better diagnose the important challenges to focus on.

Overall, our contributions can be summarized as follows:

1. We introduce a Distributional View for controlled text generation formalized as a constraint satisfaction problem combined with a divergence minimization objective, providing a single framework both for "distributional" constraints (collective statistical requirements) and for "pointwise" constraints (hard requirements on each individual) (**§2.1**). To our knowledge, this is the first framework with such generality for controlled text generation.

2. We show how these constraints lead to an optimal EBM for the target model (**§2.2**), propose the KL-Adaptive DPG algorithm for approximating the optimal EBM distribution by

---

[2]Additional Related Work is provided in §E. We use §A, §B ... to refer to sections in the Appendix.

[3]One possible sampling approach here would be to employ MCMC techniques, such as Metropolis-Hastings (Robert & Casella, 2005). These come with theoretical convergence guarantees in the limit but in practice convergence can be very difficult to assess, and furthermore, obtaining samples can be extremely slow.

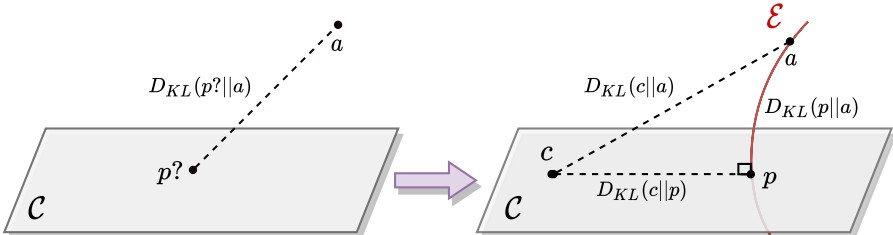

**Figure 1:** From MaxEnt to EBM through Information Geometry. The Generalized MaxEnt specification (left panel) is looking for a distribution $p$ that lies on the moment constraints manifold $\mathcal{C}$ and that minimizes the forward KL $D_{\mathrm{KL}}(p, a)$. The solution is provided by Information Geometry: (1) build the exponential family $\mathcal{E}$ determined by $a$ and $\phi$, (2) $p$ lies at the intersection between $\mathcal{C}$ and $\mathcal{E}$, (3) for any distribution $c$ satisfying the constraints, the "Pythagorean identity" holds: $D_{\mathrm{KL}}(c||a) = D_{\mathrm{KL}}(c||p) + D_{\mathrm{KL}}(p||a)$; in particular $p$ is unique.

    an autoregressive policy **(§2.3)**, and show the effectiveness of this adaptive technique for obtaining faster convergence **(§B.2)**.

3. We conduct experiments in a number of pointwise and distributional conditions, assessing results in terms of divergence from GPT-2, fluency and diversity, with better performance than strong baselines. The distributional experiments show the potential of our approach as a remedy to the current and important problem of bias in pretrained language models, providing a novel direction for addressing it **(§3)**.

## 2 FORMALIZATION

We denote by $X$ the set of all sequences $x$ of bounded length $L_{max}$, by $a$ the initial pretrained model and by $p$ the desired target model. The probabilities of $x$ according to each model are $a(x)$ and $p(x)$. Our approach consists in expressing our desiderata through constraints on the desired values $\bar{\mu}_i$ of the *expectations* (aka *moments*) $\mu_i \doteq \mathbb{E}_{x \sim p} \phi_i(x)$ of certain predefined real-valued feature functions $\phi_i(x)$, for $i \in \{1, \ldots, k\}$.

To illustrate, the previous example can be expressed by using two binary features, $\phi_1(x) = 1$ iff $x$ is classified as speaking about sports, $\phi_2(x) = 1$ iff $x$ mentions a female character. Then our "moment constraints" take the following form: $\mu_1 = \mathbb{E}_{x \sim p} \phi_1(x) = 1.0, \quad \mu_2 = \mathbb{E}_{x \sim p} \phi_2(x) = 0.5$.
The first (pointwise) constraint implies that each individual $x$ has to speak about sports (otherwise $\mu_1$ could not reach its maximum value 1.0), the second (distributional) constraint that 50% of the $x$'s have to mention a female character.[4]

Let $\mathcal{C}$ be the set of all distributions $c$ over $X$ that satisfy the moment constraints. We then propose to specify $p$ as a distribution respecting the constraints, but also minimizing KL divergence from $a$:

$$p \doteq \arg\min_{c \in \mathcal{C}} D_{\mathrm{KL}}(c, a), \tag{1}$$

Equation (1) is a generalization of the *Maximum Entropy Principle* of Jaynes (1957), which corresponds to the limit case where $a$ is the uniform $u$ distribution over $X$, noting that minimizing $D_{\mathrm{KL}}(c, u)$ is equivalent to maximizing the entropy of $c$ under the constraints — in other words, trying to find the least "specific" distribution satisfying the constraints.

### 2.1 CONSTRAINTS, INFORMATION GEOMETRY, EXPONENTIAL FAMILIES

To recap our formal approach, we have a finite set $X$, a distribution $a$ over $X$ s.t. $a(x) > 0, \forall x \in X$, and real functions $\phi_1, ..., \phi_k$ over $X$. We specify moment constraints $\mu_i = \bar{\mu}_i$ on distributions $c$ over $X$, where $\mu_i \doteq \mathbb{E}_{x \sim c} \phi_i(x)$ and the $\bar{\mu}_i$'s are given targets; the set of distributions satisfying these constraints is denoted by $\mathcal{C}$. Our Problem is to find a $p$ such that $p = \arg\min_{c \in \mathcal{C}} D_{\mathrm{KL}}(c, a)$.

We follow Csiszár & Shields (2004) on this question, a problem that is at the core of the field of Information Geometry (Nielsen, 2018; Amari & Nagaoka, 2000). Under the assumption that $\mathcal{C} \neq \emptyset$, they prove the following result (also see §A.1):

---

[4]This example uses only binary features, but real-valued features can also be used, for instance scores returned by a soft classifier.

**Theorem 1** **(A)** *There exists a unique solution $p$ to the problem above, obtained as $p(x) \propto P(x)$ where $P$ is in* exponential family *form:*

$$P(x) = a(x) \; \mathbb{1}[x \in X_{\mathcal{C}}] \; e^{\sum_i \lambda_i \phi_i(x)}. \tag{2}$$

*In other words $p(x) = 1/Z \, P(x)$, with $Z = \sum_{x \in X} P(x)$; $P$ is an unnormalized distribution, i.e. an* EBM. *Here $X_{\mathcal{C}} = \{x \in X \mid \exists c \in \mathcal{C} \text{ s.t. } c(x) > 0\}$ is the "support set" associated with $\mathcal{C}$. The $\lambda_i$'s are real numbers called the* natural parameters *associated with the moments $\mu_i$.*

**(B)** *$p$ can be approximated to arbitrary precision by distributions $p_\epsilon$ of the form:*

$$p_\epsilon(x) \propto a(x) \, e^{\sum_i \lambda_{\epsilon,i} \phi_i(x)} \tag{3}$$

*for appropriate real values of the $\lambda_{\epsilon,i}$.*

**(C)** *$p$ satisfies the* Pythagorean Identity*: $D_{\mathrm{KL}}(c, a) = D_{\mathrm{KL}}(c, p) + D_{\mathrm{KL}}(p, a), \forall c \in \mathcal{C}$ (see Fig 1).*

The advantage of this version of the connection between Generalized Maximum Entropy and Exponential Families is its generality, which distinguishes it from other presentations, and which makes it ideal for unified application to pointwise, distributional or hybrid constraints.

In the special case of only pointwise constraints, of the form $\mathbb{E}_{x \sim c} \phi_i(x) = 1.0, i \in [1, k]$, with $\phi_i(x) \in \{0, 1\}$, let's define the predicate $b(x)$ to be 1 iff $x$ satisfies all the constraints. Then, using the (A) form of the result, it is an easy exercise (see §A.2) to prove that $X_{\mathcal{C}} = \{x \in X \mid b(x) = 1\}$ and that one has $p(x) \propto a(x)b(x)$. In this case $P(x) = a(x)b(x)$ is a very simple EBM that does not involve an exponential part; this is the EBM form that we use for experiments involving only pointwise constraints.

In the general case where some constraints are distributional, the determination of $X_{\mathcal{C}}$ is not as direct, and we prefer to use the approximation provided by (B), which permits a generic implementation. With only distributional constraints, an exact solution is typically obtained with finite $\lambda$'s. With hybrid constraints, some of the $\lambda$'s may tend to infinite (positive or negative) values but thresholding them suffices to get a good approximation.

## 2.2 From Moment Constraints to EBM

Let's now consider a set of desired moment constraints $\bar{\mu}$.[5] In the general case (i.e., when some constraints are distributional), we use Theorem 1.(B), which says that the desired energy-based model $P$ can be approximated arbitrarily closely in the following form:

$$P(x) \doteq a(x) e^{\boldsymbol{\lambda} \cdot \boldsymbol{\phi}(x)}. \tag{4}$$

---

**Algorithm 1** Computing $\boldsymbol{\lambda}$

**Input:** $a$, features $\boldsymbol{\phi}$, imposed moments $\bar{\boldsymbol{\mu}}$
1: sample a batch $x_1, \dots, x_N$ from $a$
2: for each $j \in [1, N]$: $w_j(\boldsymbol{\lambda}) \leftarrow e^{\boldsymbol{\lambda} \cdot \boldsymbol{\phi}(x_j)}$
3: $\hat{\boldsymbol{\mu}}(\boldsymbol{\lambda}) \leftarrow \frac{\sum_{j=1}^N w_j(\boldsymbol{\lambda}) \, \boldsymbol{\phi}(x_j)}{\sum_{j=1}^N w_j(\boldsymbol{\lambda})}$
4: solve by SGD: $\arg\min_{\boldsymbol{\lambda}} ||\bar{\boldsymbol{\mu}} - \hat{\boldsymbol{\mu}}(\boldsymbol{\lambda})||_2^2$
**Output:** parameter vector $\boldsymbol{\lambda}$

---

This EBM defines the desired normalized distribution $p(x) \doteq \frac{P(x)}{Z}$, where $Z \doteq \sum_x P(x)$. What is left is to learn appropriate values for the parameter vector $\boldsymbol{\lambda}$ s.t.:

$$\mathbb{E}_{x \sim p} \boldsymbol{\phi}(x) \simeq \bar{\boldsymbol{\mu}}. \tag{5}$$

We address this problem through Algorithm 1. First, we sample a large number $N$ of sequences $x_1 \dots x_j \dots x_N$ from $a$. On line 2, we define "importance weights" $w_j(\boldsymbol{\lambda}) \doteq \frac{P(x_j)}{a(x_j)} = \exp \langle \boldsymbol{\lambda}, \boldsymbol{\phi}(x_j) \rangle$. On line 3, we then use SNIS (Self Normalized Importance Sampling) (Kim & Bengio, 2016; Parshakova et al., 2019a) to estimate $\boldsymbol{\mu}(\boldsymbol{\lambda}) \doteq \mathbb{E}_{x \sim p} \boldsymbol{\phi}(x)$. SNIS consists in computing:

$$\hat{\boldsymbol{\mu}}(\boldsymbol{\lambda}) = \frac{\sum_{j=1}^N w_j(\boldsymbol{\lambda}) \, \boldsymbol{\phi}(x_j)}{\sum_{j=1}^N w_j(\boldsymbol{\lambda})}, \tag{6}$$

---

[5] Boldface $\boldsymbol{\phi}$ and $\boldsymbol{\mu}$ represents vectors of real values (features and moments).

and it can be shown that $\hat{\boldsymbol{\mu}}(\boldsymbol{\lambda}) \simeq \boldsymbol{\mu}(\boldsymbol{\lambda})$, with convergence in the limit (Owen, 2013).

Note that the estimate $\hat{\boldsymbol{\mu}}(\boldsymbol{\lambda})$ is obtained not as a single number, but as a parametric function of the variable $\boldsymbol{\lambda}$. We want to find $\boldsymbol{\lambda}$ such that $\hat{\boldsymbol{\mu}}(\boldsymbol{\lambda}) = \bar{\boldsymbol{\mu}}$, a question that we handle on line 4 by performing an SGD optimization over the objective $\min ||\bar{\boldsymbol{\mu}} - \hat{\boldsymbol{\mu}}(\boldsymbol{\lambda})||_2^2$.[6]

At the end of this process, we obtain an estimated value for the parameter vector $\boldsymbol{\lambda}$, and a representation $P(x) = a(x) \exp \langle \boldsymbol{\lambda}, \boldsymbol{\phi}(x) \rangle$. While $a(x)$ is a normalized distribution by construction, the introduction of the second factor loses this normalization property, making $P(x)$ an EBM.[7] [8]

## 2.3 From EBM to Autoregressive Policy

The EBM representation just obtained for $P$ defines the optimal $p = Z^{-1}P$ unambiguously, a crucial intermediate step in the solution of our problem. From it we can immediately compute ratios of the form $p(x)/p(x')$ for two sequences $x, x'$, but without knowing $Z$, we cannot compute $p(x)$ and, *even* with such a knowledge, we cannot produce samples from $p$.

This problem is typical of EBMs at large: they provide a rich and flexible mechanism for specifying models, but they leave a gap between representation and exploitation. A range of techniques, from sophisticated MCMC approaches (especially for continuous models in vision) to contrastive learning techniques, have been developed for bridging this gap.

> **Algorithm 2** KL-Adaptive DPG
>
> **Input:** $P$, initial policy $q$
> 1: $\pi_\theta \leftarrow q$
> 2: **for** each iteration **do**
> 3:     **for** each episode **do**
> 4:         sample $x$ from $q(\cdot)$
> 5:         $\theta \leftarrow \theta + \alpha^{(\theta)} \frac{P(x)}{q(x)} \nabla_\theta \log \pi_\theta(x)$
> 6:     **if** $D_{\mathrm{KL}}(p||\pi_\theta) < D_{\mathrm{KL}}(p||q)$ **then**
> 7:         $q \leftarrow \pi_\theta$
> **Output:** $\pi_\theta$

One technique that is suitable for our objective here, namely sampling from a sequential EBM that includes an autoregressive component $a(x)$, is the DPG ("Distributional Policy Gradient") algorithm (Parshakova et al., 2019b).

The objective of DPG is to obtain an autoregressive policy $\pi_\theta$ that approximates $p$, where approximation is formalized in terms of making the cross-entropy $CE(p, \pi_\theta) = -\sum_x p(x) \log \pi_\theta(x)$ as small as possible.[9] DPG exploits the fact that, for any "proposal" distribution $q$ whose support contains the support of $p$, we have

$$\nabla_\theta CE(p, \pi_\theta) = -\nabla_\theta \mathbb{E}_{x \sim p} \log \pi_\theta(x) = -\mathbb{E}_{x \sim p} \nabla_\theta \log \pi_\theta(x) = -\mathbb{E}_{x \sim q} \frac{p(x)}{q(x)} \nabla_\theta \log \pi_\theta(x)$$

where the last equality is an instance of importance sampling.

Our "KL-adaptive" version of DPG is shown in (Algorithm 2). We start from an input EBM $P$, along with an initial policy $q$ which is a proxy to $p$; in our case we take $q = a$. During an iteration (think minibatch or set of minibatches), we sample a number of sequences from $q$, do an SGD update of $\theta$ (line 5), where $P$ is used instead of $p$ (noting that they only differ by a multiplicative constant), and where $\alpha^{(\theta)}$ is a learning rate. The efficiency of the algorithm is related to how close the proposal $q$ is to the target $p$,[10] The algorithm is *adaptive* in the sense that it modifies $q$ periodically to take advantage of the evolving approximations $\pi_\theta$. On line 6, we we test whether the current $\pi_\theta$ is closer

---

[6]$\boldsymbol{\mu}(\boldsymbol{\lambda})$ can approximate $\bar{\boldsymbol{\mu}}$ arbitrarily closely, and we know from SNIS theory that with increasing $N$, $\hat{\boldsymbol{\mu}}(\boldsymbol{\lambda})$ will become arbitrarily close to $\boldsymbol{\mu}(\boldsymbol{\lambda})$. In our experiments we stop the SGD optimization when $||\bar{\boldsymbol{\mu}} - \hat{\boldsymbol{\mu}}(\boldsymbol{\lambda})||_2^2$ becomes smaller than 0.01.

[7]The class of Energy-Based Models (EBMs) (LeCun et al., 2006) is much larger than the exponential family models we are considering in this paper. An EBM $P(x)$ is just any unnormalized distribution over an input space $X$, in other words a mapping $P$ from $X$ to the non-negative reals. The terminology comes from physics, and corresponds to writing $P(x)$ in the form $P(x) = e^{-E(x)}$, $E$ being called the "energy" associated with $x$.

[8]A question was raised by an anonymous reviewer about the viability of adding new constraints incrementally. The answer is yes, more details provided in the Appendix, §A.3.

[9]This is equivalent to minimizing $D_{\mathrm{KL}}(p, \pi_\theta) = CE(p, \pi_\theta) - H(p)$.

[10]In the limit where $q$ were equal to $p$, the algorithm would be identical to standard supervised training, except that samples would be obtained directly from the underlying process $p$ rather than a training set of samples.

than $q$ to $p$ in terms of KL-divergence, and if so we update $q$ to $\pi_\theta$ on line 7.[11] §B.2 provides an ablation study showing the effectiveness of this adaptive step for obtaining faster convergence.

# 3 EXPERIMENTS, RESULTS, AND EVALUATION

In this section we describe our evaluation methodology and perform experiments on pointwise constraints (§3.2) and on distributional and hybrid constraints (§3.3). The Appendix contains a detailed view of evaluation (§H), comparison with extra baselines (§D.2), and an ablation study (§B.2).

## 3.1 EVALUATION METRICS

The main metrics we report are: (1) $\mathbb{E}_{x \sim \pi_\theta} \phi_i(x)$, assessing the ability of $\pi_\theta$ to reach the expectation goal on the $i$-th constraint, (2) $D_{\mathrm{KL}}(p||\pi_\theta)$, the forward KL divergence from the optimal distribution (which should be as close to 0 as possible), (3) $D_{\mathrm{KL}}(\pi_\theta||a)$, the reverse KL divergence from the original GPT-2; for details on the estimation of these metrics see §B.1.

Previous work has mostly focused on the diversity of each individual output using Dist-1,2,3 scores (Li et al., 2016a) to measure repetitions within a *single* generated sequence. However, the shortcomings in terms of *sample* diversity, of optimization techniques when training generative models for text, has recently been documented in (Caccia et al., 2020). So additionally, we report Self-BLEU-3,4,5 (Zhu et al., 2018) to measure repetitions at a distributional level across the whole set of generated samples, and also provide a token/type frequency analysis (see Fig. 4 and §H.4).

Note that KL divergence from the original GPT-2 also implicitly captures sample diversity: a distribution that focuses all its probability mass on a few sequences typically displays high divergence from GPT-2. Implementation details and hyper-parameters are available in the Appendix (§ F).

## 3.2 POINTWISE CONSTRAINTS EXPERIMENTS

Pointwise constraints are of the form $\mathbb{E}_p \phi_i(x) = 1$, with $\phi_i$ a binary feature. Contrarily to distributional constraints, they can be directly associated with a "reward", namely $\phi_i$ itself. RL-inspired baselines can then be introduced naturally, and this is what we do here.

**Single-Word constraints:** Here we constrain the presence of a specific word $w$ in the generated text i.e. $\phi(x) = 1$ iff $w$ appears in the sequence $x$. We use 9 single-word constraints of different rarity levels: "US" (original frequency: $7 \cdot 10^{-3}$), "China" ($4 \cdot 10^{-3}$), "Canada" ($2 \cdot 10^{-3}$), "amazing" ($1 \cdot 10^{-3}$), "Paris" ($5 \cdot 10^{-4}$), "restaurant" ($6 \cdot 10^{-4}$), "amusing" ($6 \cdot 10^{-5}$), "Vampire" ($9 \cdot 10^{-5}$), "Wikileaks" ($8 \cdot 10^{-5}$).

**Word-list constraints:** We use 4 different word lists among those proposed in (Dathathri et al., 2020), covering the following topics: "kitchen", "fantasy", "politics", and "computers". We set $\phi_l(x) = 1$ if $x$ contains at least one one word from the word list $l$.

**Classifier-based constraints:** We use pre-trained classifiers from (Dathathri et al., 2020), which consist of a linear head on top of GPT-2. We select 4 classes and define corresponding pointwise constraints: "very positive", "positive", "very negative" and "Clickbait". See §F for details on constraint computations.

**Baselines:** We compare our method *GDC* to three baselines: (1) *REINFORCE* (Williams, 1992b), using the reward $\phi(x)$, i.e. trying to maximize $\mathbb{E}_{\pi_\theta} \phi(x)$; (2) *REINFORCE*$_{P(x)}$ : Reinforce again, but now using the reward $P(x)$ based on our energy model $P$, i.e. maximizing $\mathbb{E}_{\pi_\theta} P(x)$; this baseline starts from the same optimal EBM $P$ representation as GDC but with a standard optimization objective rather than a distributional one; in other words, while GDC tries to get a similar *sampling* distribution to $p$, this baseline tries to get sequences of *maximal* probability $p(x)$. (3) *ZIEGLER* (Ziegler et al., 2019): an approach relying on the RL Proximal Policy Optimization (PPO) algorithm (Schulman et al., 2017) and which tries to maximize the objective $\mathbb{E}_{\pi_\theta} \phi(x) - \beta D_{\mathrm{KL}}(\pi_\theta, a)$, which *interpolates* the reward $\phi(x)$ with a KL-divergence penalty from the pretrained model, but where the goal is not explicitly to satisfy a constraint; for a geometric illustration of the differences with

---

[11]In the original DPG, the superiority test is done on the basis of the log-likelihood on a validation set. Here we are in the more demanding situation where no validation set is available. To directly estimate the KL divergence from $p$ (line 6), we exploit the identity $D_{\mathrm{KL}}(p||\pi) = -\log Z + 1/Z\, \mathbb{E}_{x \sim q(x)} \frac{P(x)}{q(x)} \log \frac{P(x)}{\pi(x)}$. See §B.1 for derivations and a comparison with using Total Variation Distance (TVD) for assessing divergence.

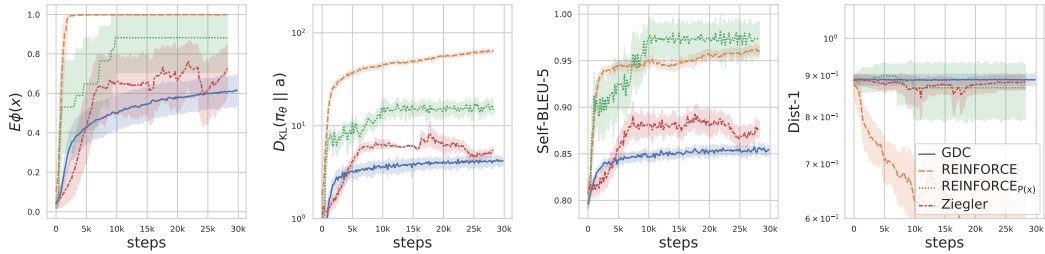

**Figure 2:** Eval. metrics $\mathbb{E}\phi(s)$, $D_{\mathrm{KL}}(\pi_\theta\|a)$ ($\downarrow$ better), Self-BLEU-5 ($\downarrow$ better), and Distinct-1 ($\uparrow$ better), aggregated across 17 point-wise experiments (single words, wordlists, discriminators), performed at each 10 gradient updates, for policies obtained from GDC against three training baselines REINFORCE , REINFORCE$_{P(x)}$ and ZIEGLER . See Appendix H for a detailed view for each experiment and more evaluation metrics.

GDC see §D.1. §D.2 provides a comparison of GDC with two additional baselines.

**Results:** Figure 2 shows the evolution of the metrics over training steps, aggregated across the $9 + 4 + 4 = 17$ experiments. We observe the following: the baseline RE-INFORCE , which does not have any explicit link in its objective to the pretrained GPT-2, converges very early in the training, reaching a maximum value of $\mathbb{E}_{\pi_\theta}\phi(x)$ at the expense of a very large deviation from the original GPT-2. High values of $D_{\mathrm{KL}}(\pi_\theta|a)$, are translated into low Dist-1 and very high Self-BLEU-5 indicating degeneration and lack of diversity. REINFORCE$_{P(x)}$ maximizes the energy model $P$ by peaking on a few sequences only; this can yield high values of $\mathbb{E}_{\pi_\theta}P(x)$, at the expense of low sample diversity as demonstrated in the highest values of SELF-BLEU-5 scores among baselines.[12]

In the case of ZIEGLER we can see a positive effect of the interpolation factor $\beta$ between the reward and the KL penalty in the objective function. In the aggregated experiments reported here, the reward is slightly better than with GDC, but with inferior diversity scores (see also Fig. 4, showing that GDC produces richer vocabulary), and the stability is much worse (a detailed view of each experiment is provided in §H, showing more clearly the instability of this baseline). A complementary evaluation is provided by Figure 3, focusing on the ability of $\pi_\theta$ to

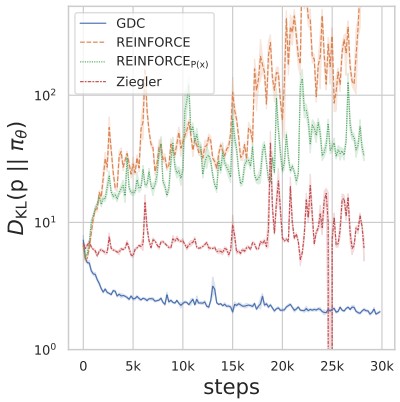

**Figure 3:** GDC steadily decreases the KL deviation between the trained policy $\pi_\theta$ and the target distribution $p$. The Figure is aggregated across 17 point-wise constraints experiments, see Appendix H for a separate view of each experiment.

converge to the optimal distribution $p$. We see that GDC is superior to all baselines in terms of $D_{\mathrm{KL}}(p\|\pi_\theta)$ and also much more stable.

In summary, in these experiments, we see that with GDC the constraint expectation $\mathbb{E}_{\pi_\theta}\phi(x)$ smoothly increases while $\pi_\theta$ maintains the lowest divergence from GPT-2, becomes closest to the optimal $p$, and has the best diversity scores overall. On the other hand, we also note that at the point where we stop training (30K steps), the average over experiments of $\mathbb{E}_{\pi_\theta}\phi(x)$, while still increasing, does not reach $100\%$, an issue that we discuss at the end of the paper (§4).

### 3.3 DISTRIBUTIONAL AND HYBRID CONSTRAINTS EXPERIMENTS

As formalized in §2, GDC permits to define pointwise and distributional constraints as well as any mix between them. This unique feature makes it very suitable to remedy biases that the text generation model may have, a problem identified in several previous works (Sheng et al., 2019b).

---

[12]The difference with REINFORCE makes sense if one observes that $\phi(x)$ can be maximized on many sequences, while $P(x)$ tries to maximize $a(x) \cdot \phi(x)$, which is typically maximized on only one sequence.

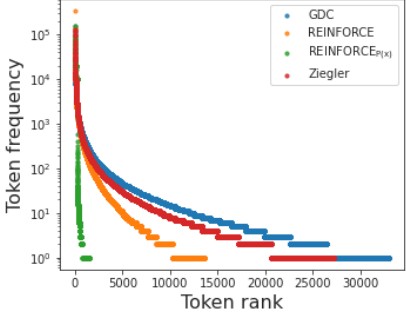

**Figure 4:** "Zipf-like" token frequency analysis on sets of 68000 generated samples from each method (only samples strictly satisfying the constraints are kept, for fair comparison). Longer tails mean a lower concentration of mass on the high frequency tokens, and therefore indicate more vocabulary richness. See Appendix H.4 for details.

| Reps | $\phi(x)$ | |
|---|---|---|
| | | **GDC** |
| 1 | 1 | "Thank you all for the service this site gives me , " he said. ... |
| 1 | 1 | This book is incredibly rich , entertaining , and extremely enjoyable... |
| | | **REINFORCE** |
| 1 | 1 | Featuring the highest quality performance performance performance... |
| 1 | 1 | This beautiful beautiful quality production quality high quality... |
| 1 | 1 | High quality performance high quality performance product ... |
| | | **REINFORCE_P(x)** |
| 10k | 1 | Thank you for supporting the journalism that our community needs! ... |
| | | **ZIEGLER** |
| 4418 | 1 | Thank you for supporting the journalism that our community needs! ... |
| 3560 | 1 | Be the first to know. No one covers what is happening in our... |

**Table 1:** Examples of generations controlled by a discriminator on the class label "*very positive*". Reps is the frequency of the whole sequence in a corpus of 10k samples. Tokens highlighted in yellow with different intensities indicates their overall frequencies in the generated corpus. Generations are trimmed to 15 tokens for display purposes. See §H.5 a full list of generations .

We employ GDC to balance gender and profession distributions across biographies generated by a GPT-2 model fine-tuned on Wikipedia Biographies (Lebret et al., 2016) *(henceforth GPT-2$^{bio}$)* (§G gives additional details). The bias in GPT-2$^{bio}$ is significant: we calculated that this model generates only around 7% female biographies. It also displays a large imbalance between professions related to "Science" (1.5%), "Art" (10.0%), "Business" (10.9%) and "Sports" (19.5%).

**Experiment 1: Single Distributional Constraint** We use the distributional constraint $\mathbb{E}_{x \sim p}\phi_{female}(x) = 0.5$; GDC is able to reduce the bias of GPT-2$^{bio}$ to obtain 35.6% female biographies rather than only 7.4% (see Fig. 2 for this experiment and the next ones).

**Experiment 2: Multiple Distributional Constraints** We then test our framework with several distributional constraints of different values and control directions. We specify four distributional constraints all at once with the goal of *increasing* the expectations of "science" and "art" to 40% and *decreasing* those of "sports" and "business" to 10%. GDC is able to increase the expectations of the first two professions respectively from 1.5% to 20.3% and from 10 to 31.6% and to decrease those of "business" and "sports" respectively from 10.9% to 10.2% and from 19.5% to 11.9%, reaching expectations close to the desired ones for all features using a single training method.

**Experiments 3,4,5,6: Hybrid Constraints** Here we want to de-bias the model as in the previous case but we single out biographies of scientists, artists, etc. Formally, our requirements become $\mathbb{E}_{x \sim p}\phi_{profession}(x) = 1.0$, a pointwise constraint, and $\mathbb{E}_{x \sim p}\phi_{female}(x) = 0.5$, a distributional constraint. In those 4 hybrid experiments we can clearly see that GDC can address both pointwise and distributional constraints increasing each simultaneously with just the right amount to reach the desired expectations. Appendix §G further elaborates Fig. 2 (convergence curves).

## 4 DISCUSSION

Our approach to controlled text generation is distinguished by its breadth — the first one to handle distributional along with pointwise constraints, with applications to the important problem of Bias in pretrained LMs — and by the transparency of the supporting formalism. It decouples the training objective along two different dimensions. The first consists in solving the initial constraints specification, and leads through a direct algorithm to an optimal solution in EBM format. The second, where the real computational difficulty lies, consists in approximating this EBM with an autoregressive policy for use at inference time.

Sampling from an EBM is an important, hard, and well-identified challenge in the literature. Our approach there consists in proposing a KL-adaptive version of the DPG algorithm, which exploits ascertained improvements of the trained policy to speed up convergence.

This is an effective method for rare events, as we show in an ablation study (§B.2). In the case of pointwise constraints, where comparisons with baselines can be done, our experiments show the

| | Aspect | Desired | Before | After |
|---|---|---|---|---|
| | **Single Distributional constraint** | | | |
| 1 | Female | 50% | 07.4% | 36.7% |
| | **Multiple distributional constraints** | | | |
| 2 | Art | 40% ↑ | 10.9% | ↑31.6% |
| | Science | 40% ↑ | 01.5% | ↑20.1% |
| | Business | 10% ↓ | 10.9% | ↓10.2% |
| | Sports | 10% ↓ | 19.5% | ↓11.9% |
| | **Hybrid constraints** | | | |
| 3 | Female | 50% | 07.4% | 31.9% |
| | Sports | 100% | 17.5% | 92.9% |
| 4 | Female | 50% | 07.4% | 36.6% |
| | Art | 100% | 11.4% | 88.6% |
| 5 | Female | 50% | 07.4% | 37.7% |
| | Business | 100% | 10.1% | 82.4% |
| 6 | Female | 50% | 07.4% | 28.8% |
| | Science | 100% | 01.2% | 74.7% |

**Table 2:** Distributional and hybrid constraints experiments demonstrating the generality of GDC in dealing with this mixed type of constraints. ↑/↓ indicates which direction (increasing/decreasing) improves the target expectation. See Appendix §G for convergence curves.

method's superiority in satisfying the constraints while avoiding degeneration. Reaching close to 100% samples meeting the constraints, can sometimes be obtained in these baselines, but only at a severe cost in terms of quality and sample diversity. Of course, if we do not care about such aspects, obtaining 100% constraint satisfaction is trivial: just generate *one* sentence satisfying the pointwise constraint!

Our method does not suffer from degeneration, but our end policies still generate a number of samples not satisfying the constraints. A possibility, left for future work, might consist in filling the moderate residual gap with MCMC techniques, which would be guaranteed to reach our optimal $p$ in the limit. We do not go this route here, but conduct an experiment (see §C) to better understand the nature of the problem. In the simple case of a single-word constraint ($x$ includes *"amazing"*), we sample directly 1M samples from GPT-2 and keep the roughly 5K samples containing *amazing* (a variant of rejection sampling, taking two processing days). We then do a standard supervised fine-tuning of GPT-2 with these samples, stopping training when the CE validation loss starts to increase, and observe that this model exhibits a worse constraint satisfaction rate than ours. This experiment does not mean that a much larger fine-tuning dataset, obtained in this slow, non-adaptive way, would not reach better statistics, but it raises doubts about the ability of the GPT-2 architecture to fine-tune over such a non-standard constraint as containing a given word *somewhere* in its output.

Overall, we believe that the proposed decomposition into two sub-problems is a methodological advantage compared to most other works, which directly aim at training a policy with the goal of improving certain evaluation metrics, but without clearly defining what qualifies as an optimal solution. The computational challenge of fully bridging the gap between the optimal EBM and an efficient sampling engine remains, and we hope that the formalism we propose, along with initial applications and experimental validations, will motivate further research along these lines.

ACKNOWLEDGMENTS

We would like to thank the anonymous reviewers for their insightful feedback that helped enhancing the final version of this manuscript. We also thank Germán Kruszewski, Laurent Besacier, Matthias Gallé and Christopher Dance for providing technical feedback on this work and proof-reading the manuscript, as well as Tetiana Parshakova and Jean-Marc Andreoli for their work on the original versions of the SNIS and DPG algorithms.

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

# Appendix

## A  DETAILS ON FORMALIZATION (§2)

### A.1  COMMENTS ON THEOREM 1

Our statement of Theorem 1 is actually a reformulation of two results in section 3 of Csiszár & Shields (2004). Our property (A) is a simple notational transposition of their Remark 3.1 (p. 444). Property (C) is the Pythagorean Identity in their Theorem 3.2 (p. 442). Property (B) reformulates the last part of the same Theorem "... and in general $\mathcal{L} \cap \mathrm{cl}(\mathcal{E}_Q) = \{P^*\}$" in terms of a limit of a sequence of distributions.

Note: Csiszár & Shields (2004) assume a finite $X$ here, but generalizations to infinite (countable and/or continuous) $X$ spaces are possible, see (Csiszar, 1975).

### A.2  THE CASE OF POINTWISE CONSTRAINTS IN §2.2

In the case of purely pointwise constraints, if $b(x) = 1$, then the distribution $c = \delta_x$ is in $\mathcal{C}$, hence $x \in X_\mathcal{C}$. Conversely, if $x \in X_\mathcal{C}$ then there is some $c \in \mathcal{C}$ such that $c(x) > 0$, implying that $b(x) = 1$. Hence $X_\mathcal{C} = \{x \in X \mid b(x) = 1\}$. Thus, in equation (2), $P(x) = a(x)b(x) \exp \sum_i \lambda_i \phi_i(x)$; but for $b(x) \neq 0$, $\phi_i(x) = 1$, so the exponential factor is a constant, which proves that $P'(x) = a(x)b(x)$ is proportional to $P(x)$, and therefore $p(x) \propto P'(x)$.

### A.3  INCREMENTALLY ADDING NEW CONSTRAINTS

An interesting question[13] is whether the process explained in §2 can be made incremental: if one has already computed a $p$ and a $\pi_\theta$ relative to a certain number of constraints, can one add a new constraint without restarting the whole process from scratch? The answer is yes, and here we provide some formal elements to understand why.

#### A.3.1  TRANSITIVITY PROPERTY OF GENERALIZED MAXENT

According to (Csiszár, 1996), the Generalized MaxEnt of sections §2.1 and §2.2 has the "Transitivity property". In our notation, this says that if we have $k' > k$ constraints, with $C$ the manifold of distributions respecting only the first $k$ constraints, $C'$ the manifold respecting all $k'$ constraints (hence $C' \subset C$), then the maxent projection $p'$ of $a$ onto $C'$ can be obtained by first projecting $a$ onto $C$, obtaining $p$, and then projecting $p$ onto $C'$, obtaining $p'$. In particular, the $k$ lambdas associated with $p$ can be directly reused as the first lambdas of the $k'$ lambda's associated with $p'$.

(Csiszár, 1996) gives only a minimal proof sketch, but it is instructive to provide the details, as we do now, because the proof is a neat illustration of the power of information geometry for problems of the kind we consider. The proof, illustrated in Figure 5, is very similar to one of the proofs for the transitivity of the orthogonal projection in Euclidean geometry.

---

[13] raised by an anonymous reviewer of our ICLR submission.

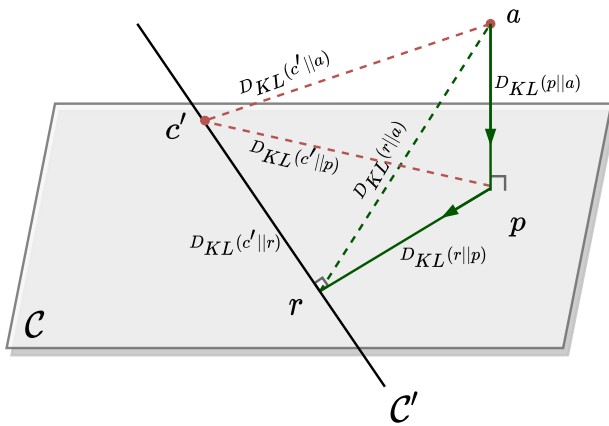

**Figure 5:** Transitivity of Information Projection (aka Generalized MaxEnt).

**Proof.** In the Figure, $p$ is the information projection (Csiszar's terminology for the Generalized Maxent) of $a$ onto $\mathcal{C}$, as before. Let's define $r$ to be the projection of $p$ onto $C'$. We need to prove that $r$ is identical to the projection $p'$ of $a$ onto $C'$. We consider an arbitrary distribution $c'$ in $C'$, and apply the Pythagorean Identity of Theorem 1 three times. Because $p$ is the projection of $a$ onto $C$, we have $D_{\mathrm{KL}}(r, a) = D_{\mathrm{KL}}(r, p) + D_{\mathrm{KL}}(p, a)$ and also $D_{\mathrm{KL}}(c', a) = D_{\mathrm{KL}}(c', p) + D_{\mathrm{KL}}(p, a)$. Because $r$ is the projection of $p$ onto $C'$, we have $D_{\mathrm{KL}}(c', p) = D_{\mathrm{KL}}(c', r) + D_{\mathrm{KL}}(r, p)$, hence $D_{\mathrm{KL}}(c', p) \geq D_{\mathrm{KL}}(r, p)$. Putting these three facts together, we find that $D_{\mathrm{KL}}(c', a) \geq D_{\mathrm{KL}}(r, a)$. As $c'$ is an arbitrary point of $C'$, this proves that $r$ is the projection of $a$ onto $C'$, in other words, $r = p'$.

### A.3.2    TRANSITIVITY AND AUTOREGRESSIVE POLICY

Due to the Transitivity property, when calculating the EBM representation, it is possible to start from $p$ without re-fitting $p'$ from scratch. However the move from EBM to autoregressive policy of §2.3 remains to be discussed. The question now is the following. We have already obtained a policy $\pi_\theta$ approximating $p$, and we are interested in obtaining a policy $\pi_{\theta'}$ approximating $p'$: is it advantageous to start Algorithm 1 with $q = \pi_\theta$, rather than starting "from scratch" and taking $q = a$ ? Intuition says "yes, very probably", because $\pi_\theta$ is by construction an approximation to $p$, which is closer than $a$ to $p'$ (formally, $D_{\mathrm{KL}}(p', p) \leq D_{\mathrm{KL}}(p', a)$, see Fig. 5, where $p' = r$). Due to the approximation, we only have $D_{\mathrm{KL}}(p', \pi_\theta) \simeq D_{\mathrm{KL}}(p', p)$ , so a formal proof that $\pi_\theta$ is superior to $a$ as a starting point is impossible, but we expect that further experiments would confirm the improvement.

## B    MORE ON ADAPTIVITY

### B.1    DETAILS ON KL-ADAPTIVITY

In this section we provide details on the comparison step in our KL-Adaptive version of the DPG Algorithm, introduced in section 2. We want to assess whether the current $\pi_\theta$ is closer than $q$ to $p$, and if the test is positive, we set $\pi_\theta$ as the new proposal, hoping to make the proposal more effective for importance sampling.

There are several ways to compute similarity between distributions, two of the most popular ones being on the one hand KL-divergence and on the other hand Total Variation Distance (TVD) — where $\mathrm{TVD}(p\|p') \doteq 1/2 \sum_x |p(x) - p'(x)|$ — which is often used in probability and MCMC theory.[14] Calculation of these metrics relative to $p$ is not straightforward since the distribution $p \propto P$ is only implicitly represented by the unnormalized EBM $P$, and we cannot easily obtain direct samples from $p$. In this section we describe a workaround.

---

[14]Both metrics are equal to 0 only if the distributions are equal everywhere (in the case of discrete distributions, which are our focus here, otherwise almost everywhere). To our knowledge, there is no obvious best metrics to use when assessing a proposal in importance sampling, leading us to conduct an ablation experiments with both metrics (Appendix 2)

Given $P$ and a proposal distribution $q$ that we can sample from, using importance sampling (Owen, 2013), one can calculate the partition function $Z$ as follows:

$$Z = \sum_x P(x) = \sum_x q(x) \, P(x)/q(x)$$
$$= \mathbb{E}_{x \sim q(x)} \, P(x)/q(x)$$

$$(7)$$

We can then compute $D_{\mathrm{KL}}(p||\pi)$ as:

$$D_{\mathrm{KL}}(p||\pi) = \sum_x p(x) \log \frac{p(x)}{\pi(x)} = \sum_x p(x) \log \frac{P(x)}{Z\pi(x)}$$
$$= -\log Z + \sum_x p(x) \log \frac{P(x)}{\pi(x)} = -\log Z + \sum_x q(x) \frac{p(x)}{q(x)} \log \frac{P(x)}{\pi(x)}$$
$$= -\log Z + 1/Z \, \mathbb{E}_{x \sim q(x)} \frac{P(x)}{q(x)} \log \frac{P(x)}{\pi(x)}$$

$$(8)$$

Similarly, for $\mathrm{TVD}(p||\pi)$:

$$\mathrm{TVD}(p||\pi) = 1/2 \sum_x |p(x) - \pi(x)|$$
$$= 1/2 \sum_x q(x) \left| \frac{\pi(x)}{q(x)} - \frac{p(x)}{q(x)} \right| = 1/2 \sum_x q(x) \left| \frac{\pi(x)}{q(x)} - \frac{P(x)}{Z \, q(x)} \right|$$
$$= 1/2 \, \mathbb{E}_{x \sim q(x)} \left| \frac{\pi(x)}{q(x)} - \frac{P(x)}{Z \, q(x)} \right|$$

$$(9)$$

In §B.2 we run an ablation study to compare the use of $D_{\mathrm{KL}}$ on line 6 of Algorithm 2) or its replacement by TVD.

For both metrics, we need an estimate of $Z$. The precision of this estimate depends on the sample size and the quality of the proposal distribution $q$. We calculate a moving average estimate $Z_{\mathrm{MA}}$ of $Z$ is used inside the estimations of $D_{\mathrm{KL}}(p||\pi_\theta)$ and $D_{\mathrm{KL}}(p||q)$ (Algorithm 3, lines 7 and 8). $Z_{\mathrm{MA}}$ is updated at each iteration of the training, and the moving average estimate is valid due to the fact that $\hat{Z}_i$, based on $K$ samples, is an unbiased estimate of $Z$, and therefore so is $Z_{\mathrm{MA}}$. In this way, the estimate benefits from *all* the samples being produced during the course of the training; and also because the proposal distribution $q$ evolves and gets closer to the target distribution $p$, the quality of the estimates of both $D_{\mathrm{KL}}(p||\pi_\theta)$ and $Z_{\mathrm{MA}}$ through importance sampling increases (equation 7). A similar approach is taken in the case of TVD (not shown).

---

**Algorithm 3** KL-Adaptive DPG (detailed)

---

**Input:** $P$, initial policy $q$

1: $\pi_\theta \leftarrow q$

2: $Z_{\text{MA}} \leftarrow 0$            $\triangleright$ Initialize Moving Average estimate of Z

3: **for** each iteration $i$ **do**

4:      **for** each step $k \in [1, K]$ **do**

5:          sample $x_k$ from $q(\cdot)$

6:          $\theta \leftarrow \theta + \alpha^{(\theta)} \frac{P(x_k)}{q(x_k)} \nabla_\theta \log \pi_\theta(x_k)$

7:      $\hat{Z}_i \leftarrow K^{-1} \sum_k P(x_k)/q(x_k)$           $\triangleright$ Estimate on the $K$ samples

8:      $Z_{\text{MA}} \leftarrow \frac{i * Z_{\text{MA}} + \hat{Z}_i}{i+1}$           $\triangleright$ Update moving average estimate of Z

9:      $\hat{D}_{\text{KL}}(p||\pi_\theta) \leftarrow -\log Z_{\text{MA}} + (K\, Z_{\text{MA}})^{-1} \sum_k \frac{P(x_k)}{q(x_k)} \log \frac{P(x_k)}{\pi_\theta(x_k)}$      $\triangleright$ Estimate on the $K$ samples

10:      $\hat{D}_{\text{KL}}(p||q) \leftarrow -\log Z_{\text{MA}} + (K\, Z_{\text{MA}})^{-1} \sum_k \frac{P(x_k)}{q(x_k)} \log \frac{P(x_k)}{q(x_k)}$      $\triangleright$ Estimate on the $K$ samples

11:      **if** $\hat{D}_{\text{KL}}(p||\pi_\theta) < \hat{D}_{\text{KL}}(p||q)$ **then**

12:          $q \leftarrow \pi_\theta$

**Output:** $\pi_\theta$

---

## B.2 ABLATION ON ADAPTIVITY

Here we run an ablation experiment on the adaptivity step of KL-Adaptive DPG (§2). We compare three variants of our proposed method: **DPG-KLD**, which uses KL divergence from the target distribution $p$ to measure the quality of the trained policy $\pi_\theta$ i.e. if $D_{\mathrm{KL}}(p\|\pi_\theta) < D_{\mathrm{KL}}(p\|q)$ we update the proposal distribution $q \leftarrow \pi_\theta$. **DPG-TVD** is similar but with the total variation distance instead (TVD). In **non-Adaptive** the initial proposal $q$ is kept fixed during training.

We run 3 point-wise experiments with single word constraints of three rarity levels in the original GPT-2 distribution, namely: "Vampire" $(1/10^4)$,"Paris" $(1/10^3)$,"US" $(1/10^2)$ .For each we use 3 different seeds and train for $10k$ gradient updates.

Figure 6 shows training trends of the three ablations. We find a significant difference in convergence speed in favour of the adaptive methods. The efficiency gap between Adaptive and non-Adaptive methods becomes larger the more rare the constraints are. i.e. the proposal distribution $q$ starting point is very far from the target distribution $p$, as the efficiency of the DPG algorithm is related to how close the proposal $q$ is to the target $p$. When $q$ is continuously adapted, the proposal distribution becomes closer to $p$ and the training becomes efficient regardless of how far the initial proposal distribution is from $p$. We observe similar convergence rates for DPG-KLD and DPG-TVD.

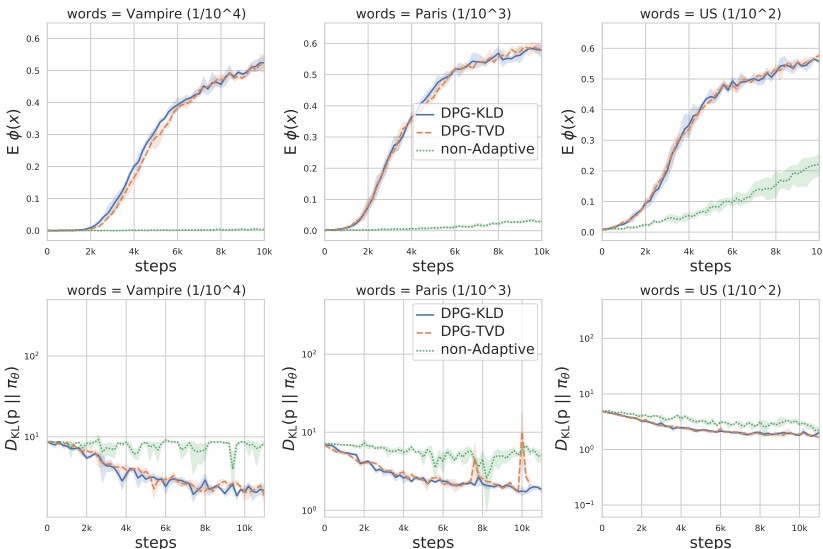

**Figure 6:** Ablation experiment elaborating the effectiveness of the adaptive step in the DPG algorithm explained in section 2. We compare three adaptivity variants, based on the KL divergence (DPG-KLD), on the TVD distance (DPG-TVD) and with no adaptation. We find similar convergence rates for both KLD and TVD adaptive DPG compared to a much slower convergence without adaptation.

## C    CAN STANDARD SUPERVISION FULLY SATISFY THE CONSTRAINTS?

In this section, we try to better understand potential difficulties of autoregressive models to fully satisfy constraints such as the ones illustrated in our pointwise experiments.

To this end, we consider whether a standard fully supervised fine-tuning of GPT-2 can achieve that objective while keeping a minimal distance from the initial model. To answer the question, we carry out an experiment where we fine-tune GPT-2 on a collection of samples satisfying the desired constraint. Our goal here is to investigate whether GPT-2 can fully satisfy the constraint without overfitting the fine-tuning data, since overfitting (memorizing) the training data basically means high KL-divergence from the initial model.

For this experiment, we choose a single-word constraint with the word "amazing". We start by sampling 1M sequences from GPT-2 small — a process that took us roughly 48 hours — and keeping only the ones containing "amazing" (this filtration process can be seen as a variant of rejection sampling (Casella et al., 2004)). We end up with a total of 4600 samples out of which we use 500 for validation and the rest for fine-tuning.

Figure 7 shows evolution of both validation loss and constraint satisfaction $\mathbb{E}\phi(x)$ on samples generated from the model during fine-tuning. Interestingly, the lowest validation loss corresponds to only $\mathbb{E}\phi(x) \approx 0.56$. Higher values of $\mathbb{E}\phi(x)$ correspond to higher validation loss i.e. to overfitting.

This result suggests a relationship between training a policy reaching 100% and overfitting the training data. This hints at the difficulty of strictly imposing certain types of constraints on pre-trained language models without moving far away from the initial model.[15]

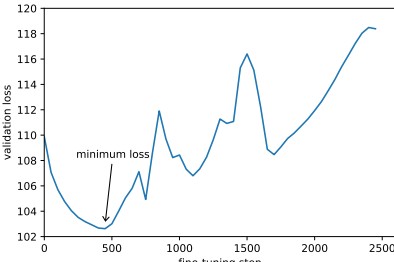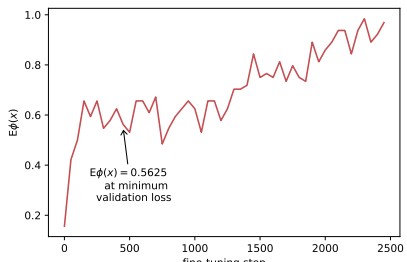

**Figure 7:**    Supervised experiment when fine-tuning GPT-2 on a corpus of sentences containing the word "amazing". **Left:** validation loss development during fine-tuning. **Right:** percentage of samples generated using the fine-tuned model and containing the word "amazing". Here, the best model according to the validation loss is only able to achieve $\mathbb{E}\phi(x) = 0.5625$. Higher values of $\mathbb{E}\phi(x)$ tend to occur with higher validation loss, i.e when overfitting.

---

[15]Note how very difficult the job would be in the extreme case of a constraint was based on a hash-based predicate filtering on average one sentence out of two.

## D  MORE COMPARISONS

### D.1  ILLUSTRATION COMPARING GDC, REINFORCE, AND ZIEGLER

The figure below illustrates the difference between GDC, the RL-based REINFORCE and ZIEGLER baselines for a pointwise constraint. The main points to note are: (1) REINFORCE is trying to find a distribution $p_R$ maximizing $r(x)$ (meaning that $p_R$ lies on the $\mathcal{C}$ manifold), but this $p_R$ is free to land anywhere on this manifold, and (2) ZIEGLER is trying to find a distribution $p_Z$ that interpolates (with a weight $\beta$) between a high average $r(x)$ and the KL divergence from $a$; unless $\beta = 0$, in which case we are back to REINFORCE, $p_Z$ does not satisfy the constraint and falls outside of the manifold.

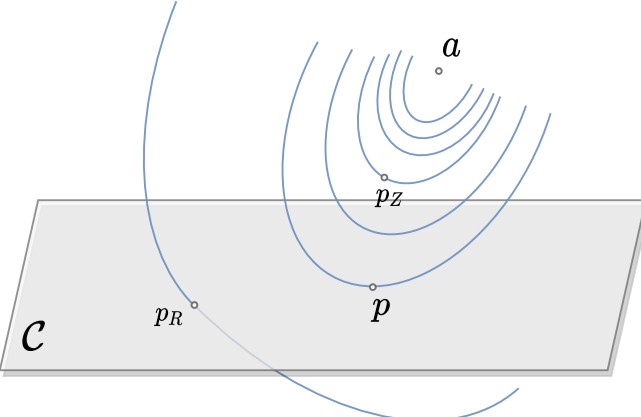

**Figure 8:** Case of a pointwise binary requirement $r(x) = 1$: comparison with Reinforce and Ziegler. The curves correspond to different $D_{\mathrm{KL}}(\cdot, a)$ levels. The manifold $\mathcal{C}$ is the set of distributions $c$ s.t. $c(x) > 0 \rightarrow r(x) = 1$, or, equivalently s.t. $\mathbb{E}_{x \sim c} r(x) = 1$. The curved lines represent increasing levels of the KL divergence $D_{\mathrm{KL}}(q, a)$. According to Reinforce, any distribution $p_R$ s.t. $\mathbb{E}_{x \sim p_R} r(x) = 1$, that is, any distribution on $\mathcal{C}$, is optimal. According to Ziegler, to each temperature $\beta > 0$ is associated an optimal distribution $p_Z = \arg\min_q \beta D_{\mathrm{KL}}(q, a) - \mathbb{E}_{x \sim q} r(x)$, which does not directly lie on $\mathcal{C}$ — this is because, as indicated in (Ziegler et al., 2019), this distribution is of the form $p_Z(x) \propto a(x) e^{r(x)/\beta}$, giving positive probability to all $x$'s in the support of $a$, including to points not lying on $\mathcal{C}$. Our own optimal $p$ does lie on $\mathcal{C}$ by definition, while minimizing the KL divergence from $a$.

### D.2  COMPARISON AGAINST FURTHER BASELINES

Here we compare GDC to other baselines, namely Plug and Play (PPLM) (Dathathri et al., 2020) and CTRL (Keskar et al., 2019) for sentiment control. PPLM works by updating the hidden states of GPT-2 for a given prefix in order to derive the generation towards the desired attributes. Unlike GDC, PPLM needs a prefix to perform its hidden-state updates. Thus, our approach is more general in the sense that any prefix can be used on the trained model at test time, rather than requiring prefix-specifc fine-tuning. CTRL is a large-scale language model (1.63 billion parameters and ~14x larger than GPT-2 small) based on control codes for steering text style and content. For the purpose of generating positive/negative sentiments using CTRL, we use its positive/negative reviews control codes as done in (Dathathri et al., 2020). The control codes used are "`Reviews Rating:  5.0`" and "`Reviews Rating:  1.0`" for positive and negative sentiment control, respectively. We use five different prefixes *(or prompts)* and generate 100 continuations given each prefix obtaining a total of 500 samples. It is worth noting that GDC is trained in the same way as described in the main text, i.e. without any knowledge of prefixes, and that we only use prefixes at test time with the saved checkpoint. The five prefixes used come from (Dathathri et al., 2020): "The chicken ", "The potato ", "The lake ", "The pizza ", and "The horse ".

We use the same sampling parameters across all approaches by setting the temperature $T = 1.0$, using top-k sampling with $k = 10$, and removing the repetition penalty used in CTRL (Keskar et al., 2019). However, we notice that CTRL does not work well with higher $T$ values (apparent in the

samples in Table 3), therefore we report also CTRL evaluation with lower temperature $T = 0.5$ and a repetition penalty $\lambda_{rep} = 1.2$ as reported in their paper.

As metrics, we use sentiment class expectation $\mathbb{E}\phi(x)$, the perplexity according to an external GPT-2 small architecture as in (Li et al., 2018), and the diversity metrics introduced in section §3.1. We average all these metrics across the 500 continuations generated. Table 3 shows the results for positive and negative sentiment control experiments. As shown, GDC is able to achieve better positive/negative sentiment with lower perplexity than both PPLM and CTRL. As for diversity, GDC achieves comparable diversity to the other two approaches and even outperforms PPLM on the Dist-n metrics in the positive sentiment task.

Table 4 shows sample continuations from all three approaches. Clearly, PPLM and CTRL exhibit some form of degeneration and repetition in many of the continuations (highlighted in light red), which is reflected in their very high perplexity score compared to GDC, which produces much more natural text with minimum repetitions without requiring a repetition penalty as CTRL.

It is also worth noting here that CTRL (and other control code methods) is very much limited in terms of its applications. For instance, to generate positive/negative sentiment text as we do in this experiment, we are required to use the ``Reviews Rating...'' control code, using control codes outside of those CTRL was fine-tuned on leads to very bad generations. This, in turn, restricts the generated text to positive/negative reviews although we may desire different types of positive/negative text (e.g. news reports). We can observe this effect[16] in some of the samples in Table 4 such as "The chicken we just ordered from Amazon.com..." and "The pizza works no matter what settings you use it on.

| Method | $\mathbb{E}\phi(x)\uparrow$ | Perplexity $\downarrow$ | Dist-1 $\uparrow$ | Dist-2 $\uparrow$ | Dist-3 $\uparrow$ | SB-3 $\downarrow$ | SB-4 $\downarrow$ | SB-5 $\downarrow$ |
|---|---|---|---|---|---|---|---|---|
| **Positive Sentiment** | | | | | | | | |
| P&P | 0.52 | 29.26±22.07 | 0.72 | 0.89 | 0.91 | **0.98** | 0.96 | 0.92 |
| CTRL | 0.28 | 76.52±90.51 | **0.82** | **0.95** | **0.94** | **0.98** | **0.95** | **0.90** |
| GDC | **0.56** | **13.53±3.18** | 0.76 | 0.91 | 0.92 | 0.99 | 0.97 | 0.95 |
| CTRL* | 0.78 | 26.80±11.89 | 0.90 | 0.97 | 0.95 | 0.99 | 0.98 | 0.97 |
| **Negative Sentiment** | | | | | | | | |
| P&P | 0.14 | 27.72±23.95 | 0.73 | 0.90 | 0.92 | 0.98 | 0.95 | 0.92 |
| CTRL | 0.16 | 82.05±54.74 | **0.82** | **0.95** | **0.94** | **0.97** | **0.94** | **0.90** |
| GDC | **0.51** | **13.59±3.84** | 0.73 | 0.87 | 0.88 | 0.98 | 0.97 | 0.94 |
| CTRL* | 0.44 | 28.50±12.86 | 0.90 | 0.97 | 0.95 | 0.99 | 0.98 | 0.96 |

**Table 3:** Comparison against PPLM (Dathathri et al., 2020) and CTRL (Keskar et al., 2019) on positive and negative sentiment control. We generate 100 samples for each prefix obtaining a total of 500 samples. All metrics shown are averaged across the 500 samples obtained. CTRL refers to the shared setting across all approaches with temperature $T = 1.0$ and repetition penalty $\lambda_{rep} = 1.0$ and CTRL* refers to having $T = 0.5$ and $\lambda_{rep} = 1.2$. Here, we see a clear advantage of GDC in terms of constraint satisfaction and perplexity and a comparable performance in terms of diversity against PPLM and CTRL.

---

[16]With lower temperatures, this behaviour becomes even worse and CTRL mostly generates reviews.

| | **GDC** |
|---|---|
| Positive | The chicken is so tasty! This recipe has been in my freezer for about 2 months now. I have always had good quality chicken breasts, so I had a huge amount of chicken and
The potato is my favourite part of the recipe, and it is also my biggest problem. It is a good addition to anything you try to add to your meal. I love potato soup
The lake has been the most active place in the country for visitors since it opened last summer, and it's not hard to see why. The lake has been a place to stay for years,
The pizza place is great for a good time! They have all kinds of amazing sandwiches. My favorite sandwich is the "Chicken Pizza" which is the best I've ever had on my way to I don't think the pizza is better than any other pizza. It's not a
The horse is very well balanced with the horse's head and ears. It is a great horse to have for the family. The horse is quite large and the tail is long enough to accommodate the", |
| Negative | The chicken is so bad it's literally going to die. It's like the dog is dying from a bad diet. I'm not even sure I want it in my kitchen"
The potato was really bad, but it's the worst potato that I've ever tasted. It was really hard for me to put my finger on. I was very disappointed with the flavor of the
The lake was just too cold for the water and the wind to blow. I couldn't get out of it. I couldn't see anything. The wind blew through my windshield and my car windshield"
The pizza is so bad that I've had to buy it again, even if I was just going to order one at a time. I'm not going to be buying one for the next week
The horse in the back row is not going to win the race. It is going to go up in flames, and the race will end in a heap of dust. The winner will be thrown |

| | **PPLM** |
|---|---|
| Positive | The chicken and the egg story: the extraordinary rise of the powerful man as he transforms the lives of ordinary people in a world in crisis The story of the A man dressed in a white suit
The potato. It is the world's most awesome people, and and and the that the the the a the a the a the a the a the , "and the the
The lake is a great spot to enjoy the spectacular views of the Great Lakes. This is also a great place to take the children's swim. The lake is also a great place to hike in the beautiful mountains
The pizza is a delight! I have never had this before. I am a fan of Italian, and I have not had it before in the States. I will be back! It was a great experience
The horse is a powerful, beautiful, and extremely intelligent animal., (.C,:,.,.,../.../.'.,,' (., ". |
| Negative | The chickenpox epidemic of 1918-1920 in Britain was an acute and deadly disease that killed about 100,000 people worldwide, most of them infants. The 1918-1919 epidemic was caused by the
The potato is one of those things we all dream of. I think the most common thing that people come up with when I say I have the perfect one is the idea of a "salt water" version
The lake is one one one. <endoftext>The United Nations (UN) and the European Union (EU) are among a number of the world's most in the state and,, on the House vote for
The pizza crust is anvil, which is what the British have for a long time. The British Empire, the French, the the the the the a in the that is a a it is called and it
The horse is in the saddle. That's how he's been for the last four years. The Tampa Bay Lightning won a series of three games in a row to begin the new year and into January we were |

| | **CTRL** |
|---|---|
| Positive | The lake I am looking forward to seeing in September! The sea scene alone would have me watching again! Rating: 5.0 One of the best comedies I've seen. We will definitely watch it again. Smart and funny
The horse for this ones lines is:"The road to Hell is paved with good intentions. All roads to Hell end in Hell themselves." Rating: 5.0 I live in a small
The potato were "seeded" during a European settlement. What the characters have gone through is inevitable, but extremely rare. (And the potato has the honor of being the world's oldest potato. For that honor, we have a nickname: "@ @
The chicken we just ordered from Amazon.com has not yet arrived and I am EXTREMELY EXCITED! The seller has the finest poultry in the market....plus, it is DELICIOUS!Thank you so
The pizza has been around for decades. Now that time has been added to it, all of us can appreciate it better, and enjoy it the way we have always enjoyed.PERFECT Pie:(The second listen) And it |
| Negative | The pizza works no matter what settings you use it on. The icecream maker always leaks out around the spout and onto the base (gross) - finally stopped working. I only wish I had spent more for a
The horse can not be found. Characters whose names show up in the battle screen:EXE: SRMX&OY; SQX the knight ¿QWOKB SKOZY the warrior!A useful upgrade for a
The lake has been made, but it's far from Earth 5. The ship has disappeared but they continue to radio.Ignoring the plot, which the Star Trek series never bothered with, Spock says that "we should have followed up. There is
The chicken died on me after 8 months. I don't think the unit is compatible with young chickens. Not recommended. Rating: 1.0 the plates didn't last long enough for me.I bought two of these plates and they
The potato does not start from eggplants, it starts from the start of generation! How stupid is that! :( I bought this and many others to try with my toddler for his preschool class. I want him to get |

**Table 4:** Samples generated from GDC, Plug and Play (Dathathri et al., 2020) and CTRL (Keskar et al., 2019) for both positive and negative experiments. Control codes are omitted for CTRL. Prefixes are underlined. Repetitions are highlighted in light red. As shown, PPLM and CTRL produce more repetitions compared to GDC.

## E    RELATED WORK EXTENDED

**Optimizing global rewards for Text Generation**    There is a large reinforcement learning inspired literature about steering an autoregressive sequential model towards optimizing some global reward over the generated text. This includes REINFORCE (Williams, 1992a) for Machine translation (MT) Ranzato et al. (2016), actor critic for Abstractive Summarization (Paulus et al., 2018), Image-to-Text Liu et al. (2016b), Dialogue Generation Li et al. (2016b), and Video Captioning (Pasunuru & Bansal, 2017). With respect to rewards, some approaches for Machine Translation and Summarization (Ranzato et al., 2016; Bahdanau et al., 2017) directly optimize end task rewards such as BLEU and ROUGE at training time to compensate for the mismatch between the perplexity-based training of the initial model and the evaluation metrics used at test time. Some others use heuristic rewards as in (Li et al., 2016b; Tambwekar et al., 2019), in order to improve certain a priori desirable features of generated stories or dialogues. Other non-RL techniques for approximating the global sequence constraints $\phi(x)$ by a biased estimator $\phi(x_t|x_{:t-1})$. These techniques usually referred to as weighted decoding Holtzman et al. (2018); See et al. (2019) this however still requires a heavy search procedure and this biased estimation of sequences that satisfy the global constraint compromises fluency and coherence. Continuous approximation using the Gumbel Softmax was developed for the training of Variational Autoencoders but several works have implemented it for natural language generation Shetty et al. (2017); Chu & Liu (2019); Kusner & Hernández-Lobato (2016).

**Competing Degeneration in Controlled Text Generation**    When using such approaches, one needs to take care of not forgetting too much of the original LM policy ("degeneration"): Liu et al. (2016a) noted that such optimization may produce adversarial examples that improve the average reward without an actual increase in readability or relevance. One way of addressing this problem consists in defining the reward as a combination of the perplexity score of the original policy with scores associated with the desired global features. Wu et al. (2016); Paulus et al. (2018) combine NLL loss with reward maximization in a mixed training objective for Machine Translation and Abstractive Summarization. Yang et al. (2018) use a set of Language Models pretrained on the target domain as a control signal for text style transfer. As a proxy to perplexity, Holtzman et al. (2018) design hand-crafted rewards using a set of discriminators to ensure the quality of generated text in open-ended text generation. Liu et al. (2016a), however, show that defining a combination reward accounting for text fluency is highly non-trivial and the results of directly optimizing it cannot be fully trusted.

**KL Divergence penalty**    Another approach relied on penalizing too large deviations of the trained policy relative to the original policy. Jaques et al. (2017; 2019) propose a conservative fine-tuning approach with a KL penalty between the trained policy and the original auto-regressive model. This penalty acts as a regularizer to the optimization process that prevents the trained policy from deviating too much from the original policy. Ziegler et al. (2019) follow a similar approach for fine tuning a language model based on human preferences, in this case a proximal policy algorithm (Schulman et al., 2017) is used to maximize the combined reward. PPLM (Dathathri et al., 2020), this time in a plug-and-play rather than a fine-tuning context, also use KL divergence to penalize deviations from the initial policy.

**Pointwise vs. Distributional View**    Most of the existing works on Controlled Generation have taken what we have called a pointwise view: focusing on the quality of each individual output, as opposed to *distributional* properties of the collection of all outputs. And in fact, the standard objective of RL is to *optimize* a pointwise reward. Even when policy-gradient methods do consider distributions over outputs, they only do as a tool towards producing maximal rewards; and in fact, it is a side effect of the limited capacity of the policy networks that such distributions do not peak on a single output, as would be the optimal outcome in cases of real-valued rewards with no ties.[17] By contrast to this usual optimization "intent", our own intent here is explicitly distributional, and the policies we are looking for are not simply tools towards maximizing scores, but actual objectives in their own right.

---

[17] In which cases the distribution $q$ maximizing $\mathbb{E}_{x \sim q} R(x)$ would be $q = \delta_{x^*}$ for $x^* = \arg\max_x R(x)$.

Such a change of perspective might be argued against in the case of conditional seq2seq problems, such as Machine Translation, where focusing on a single good output for a given input makes sense, but is clearly in-adapted when focusing on language models where sample diversity is a requirement.

**Energy Based Models for Text**    Energy-Based Models (EBMs) (Hinton, 2002; LeCun et al., 2006; Ranzato et al., 2007) are learning frameworks that attracted a lot of attention several decades ago.[18] There has been a recent surge of interest in these types of models across a variety of fields. Some early NLP-related EBM research is concerned with neural-based sequence labelling problems (e.g. tagging) exploiting the global sequence (Andor et al., 2016; Belanger & McCallum, 2016). Some current applications to text generation include Parshakova et al. (2019a) and Deng et al. (2020), who augment a standard autoregressive LM with an additional global factor in order to get a lower perplexity on the training data. Tu et al. (2020) propose an energy-based method to perform inference networks from pretrained Non-Autoregressive Machine Translation models. A recent survey of EBMs for text is provided in Bakhtin et al. (2020).

---

[18]The early work on "Whole sentence exponential models" by (Rosenfeld et al., 2001) — which only came to our attention when preparing the final version of this paper — can be considered as a form of EBM over texts. While it does not utilize neural networks, it does exploit, as we do, the exponential family in order to provide a global form of control over texts.

## F  Hyperparameters and Training Details

We implement GDC and all baselines using the PyTorch framework (Paszke et al., 2019). For all experiments we start from a pretrained GPT-2 small (117M parameters) obtained from the Hugging-Face library (Wolf et al., 2019) and fine-tune for 3K gradient-update steps. Each training required 2 Nvidia V100 GPUs, the longest model took $\sim 72$ hours to train.

A list of the hyperparameters used for GDC and baselines is given in table 5. $K$ refers to the number of gradient steps per iteration in Algorithm 2.

$N$ refers to the number of samples required and $\mu_{tolerance}$ to the minimum tolerated error $||\bar{\boldsymbol{\mu}} - \hat{\boldsymbol{\mu}}(\boldsymbol{\lambda})||_2^2$ while optimizing $\boldsymbol{\lambda}$, and $\boldsymbol{\lambda}_{learning}$ is the SGD step size for updating $\boldsymbol{\lambda}$ in Algorithm 1.

During training of the policy $\pi_\theta$, we perform periodic evaluation as follows: every 10 minibatch gradient updates, we sample 2048 sequences of 40 tokens long, using *nucleus sampling* with $top_p = 0.9$ (Holtzman et al., 2020) and estimate diversity metrics on these samples. On the other hand, for accurate estimations of $D_{\text{KL}}$ based metrics we perform pure sampling on another set of 2048 sequences of 40 tokens long.

For word-lists in the pointwise experiments in section 3.2, we used the 4 word lists from the Plug and Play (Dathathri et al., 2020) repository[19]. As for the sentiment and clickbait classifiers, we used their pre-trained classifier heads over GPT-2 medium[20].

For distributional and hybrid experiments, we fine-tune GPT-2 small (117M params) to produce biographies on a dataset of 700K Wikipedia biographies (Lebret et al., 2016) which we refer to as GPT-2[bio]. To detect if a given text is about a *female* gender, we construct $\phi_{female}(x)$ as a simple rule-based discriminator that depends on the percentage of female personal pronouns (she, her, hers, herself) w.r.t. all mentioned pronouns. We define four types of professions "Art", "Science", "Business and Politics", and "Sports". To detect them, we define a wordlist for each type as shown in table 6.

| Training Method | Constraint | Hyperparameters |
|---|---|---|
| $\forall$ | $\forall$ | `steps=3K, top_p=0.9, warmup=10,`
`dropout=0.1, lr= 0.0000141,`
`optimizer=adam.` |
| $\forall$ | Single word word-list/classifier | `gen_length=25`
`gen_length=40` |
| REINFORCE | Word-list/classifier | `batch_size=256` |
| ZIEGLER | $\forall$ | `batch_size=256, γ=1.0, λ=0.95,`
`clip_range=0.2, target_KL=6.0,`
`horizon=10000, initial_KL_coefficient=0.2` |
| GDC | All Pointwise | `batch_size=2048, `$K$`=20480` |
| | Distributional | $N$`=20k, batch_size=2048, `$K$`=20480,`
$\mu_{tolerance} = 0.01, \boldsymbol{\lambda}_{learning} = 0.5$ |

**Table 5:** Hyperparameters used throughout all experiments. $\forall$ denotes common parameters between all training methods or constraints.

| Profession | Word-List |
|---|---|
| **Art** | `storyteller, author, poet, actor, artist, actress, sculptor, screenwriter,`
`singer, musician, composer, conductor, songwriter, designer` |
| **Science** | `scientist, sociologist, philosopher, inventor, student, astronomer, historian,`
`academic, researcher, chemist` |
| **Business/Politics** | `businessman, businesswoman, entrepreneur, chairman, chairwoman, governor,`
`politician, journalist, ambassador, communist, liberal, officer, lawyer, queen,`
`king` |
| **Sports** | `footballer, trainer , player, swimmer, cyclist, athlete , wrestler, golfer,`
`cricketer` |

**Table 6:** Words in each profession word list used in the distributional constraints experiments.

---

[19]https://github.com/uber-research/PPLM/tree/master/paper_code/wordlists
[20]https://github.com/uber-research/PPLM/tree/master/paper_code/discrim_models

## G    DISTRIBUTIONAL AND HYBRID CONTROL EXPERIMENTS FOR DEBIASING LANGUAGE MODELS

Large pretrained Language Models are often trained on uncurated data from the internet, where several demographics are severely underrepresented. One of those demographics is women, whose biographies make up only $18.58\%$ of English Wikipedia's biographies (Graells-Garrido et al., 2015). It is expected that such bias is transferred if not amplified by Language Models. Previous work has suggested associations of certain demographics with certain professions, sentiments and stereotypes (Sheng et al., 2019b; Brown et al., 2020b; Nadeem et al., 2020). This shows thaat Bias in LMs also shows up in different forms than just under-representation, and the task of debiasing LMs could require more a complex control method. GPT-2$^{\text{bio}}$ demonstrates a large initial bias: over a large sample of size 20480 examples using top-p sampling ($p = 0.9$), it generates only around 7% female biographies. and a large imbalance between profession types "Science" ($1\%$), "Art" ($10\%$), "Business&Politics" ($10\%$) and "Sports" ($20\%$).

In this set of experiments, we demonstrate the potential of GDC as flexible general framework that can control pretrained Language Models to impose pointwise, distributional constraints, or even a mix between them (hybrid constraints). We design a set of 6 experiments whose descriptions and results are displayed in the figures below. Generation examples are provided in Table 7.

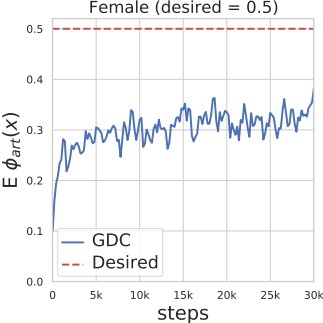

**Figure 9:** *Exp1: Single Distributional Constraint.* Balancing demographics can be represented easily through distributional constraints. By using a constraint such as $\mathbb{E}_{x \sim p}\phi_{female}(x) = 0.5$, we can target balancing the female biographies in the distribution of all generations. Note that a point-wise objective $\mathbb{E}_{x \sim p}\phi_{female}(x) = 1.0$ would maximize the presence of female biographies at the expense of other demographics, inducing bias in the opposite direction. The plot shows how $\mathbb{E}_{x \sim p}\phi_{female}(x)$ evolves towards the defined expectation: GDC is able to reduce the bias of GPT-2$^{\text{bio}}$ to obtain 36.7% female biographies rather than just 7%.

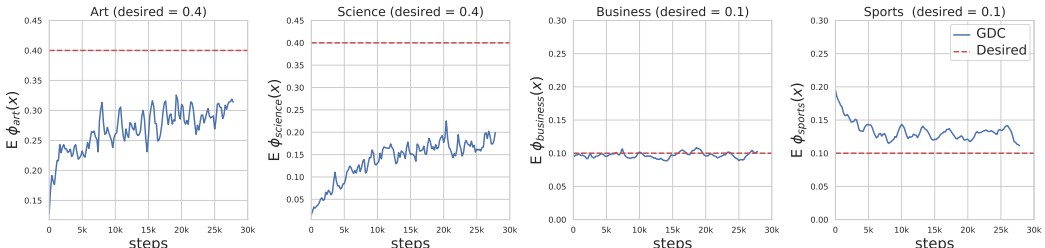

**Figure 10:** *Exp2: Multiple Distributional Constraints* This experiment demonstrates the flexibility of GDC in dealing with several distributional constraints at once, even when these constraints have different objectives (increase, decrease, or keep fixed). We challenge the flexibility of GDC by setting four distributional constraints with four arbitrary expectation values targeting $\mathbb{E}\phi_{science}$ and $\mathbb{E}\phi_{art}$ at 40% and $\mathbb{E}\phi_{sports}$ and $\mathbb{E}\phi_{business}$ at 10%. In the figure, from left to right, we can note the increase of $\mathbb{E}\phi_{science}$ and $\mathbb{E}\phi_{art}$ from 1.5% to 20.3% and from 10% to 31.6% respectively. Interestingly, the initial $\mathbb{E}\phi_{business}$ of GPT-2[bio] (10.9%) is already very close to the desired expectation (10%), and we can see that during the course of the training, GDC keeps this value fixed as it is already satisfying the corresponding target distributional constraint. $\mathbb{E}\phi_{sports}$ initially starts higher than the target distributional constraint 10%, and we can note that GDC succeeds to reduce it from 19.6% to 11.9%.

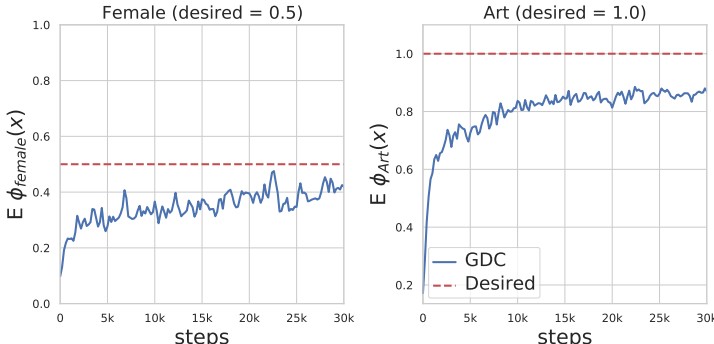

**Figure 11:** *Exp3: Hybrid constraints* In this experiment, we specify two types of constraints: pointwise with $\mathbb{E}\phi_{art}(x) = 1.0$ and distributional with $\mathbb{E}\phi_{female}(x) = 0.5$ (henceforth Hybrid). GDC in a single training procedure is able to increase the expectation of biographies about females from 7.4% to 36.6% and Art professions from 11.4% to 88.6%.

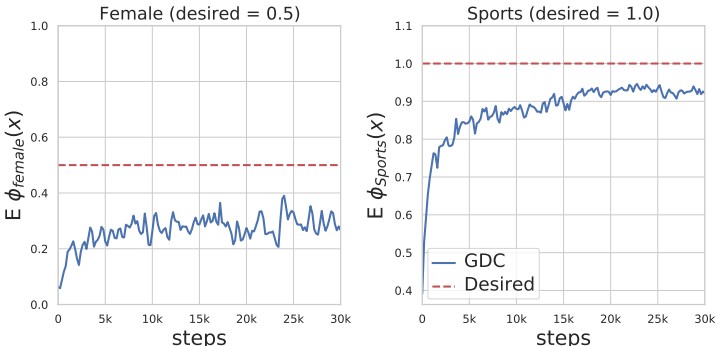

**Figure 12:** *Exp4: Hybrid constraints.* In this experiment, we specify two types of constraints: pointwise with $\mathbb{E}\phi_{sports}(x) = 1.0$ and distributional with $\mathbb{E}\phi_{female}(x) = 0.5$. GDC in a single training procedure is able to increase the expectation of biographies about females from 7.4% to 31.9% and Sports professions from 17.5% to 92.9%.

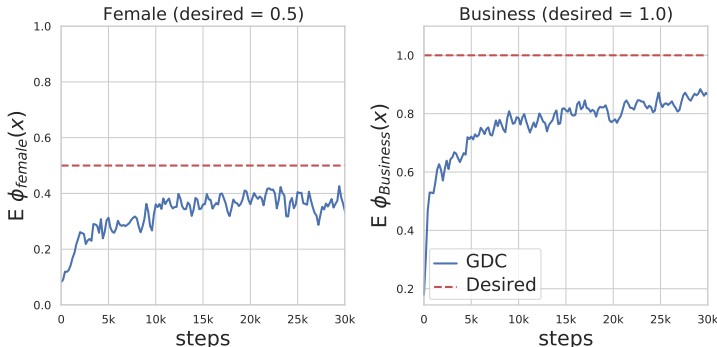

**Figure 13:** *Exp5: Hybrid constraints.* In this experiment, we specify two types of constraints: pointwise with $\mathbb{E}\phi_{business}(x) = 1.0$ and distributional with $\mathbb{E}\phi_{female}(x) = 0.5$. GDC in a single training procedure is able to increase the expectation of biographies about females from $7.4\%$ to $37.7\%$ and Business professions from $10.1\%$ to $82.4\%$.

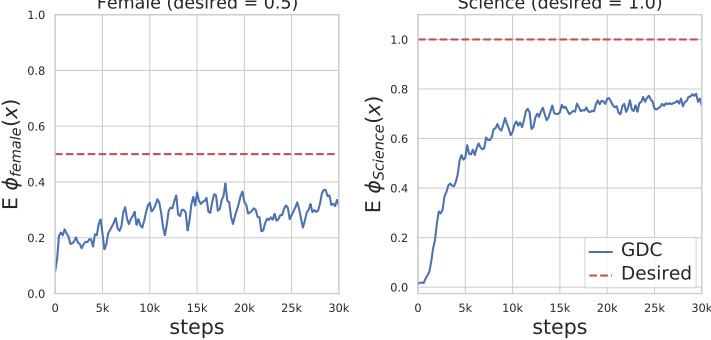

**Figure 14:** *Exp6: Hybrid constraints.* In this experiment, we specify two types of constraints: pointwise with $\mathbb{E}\phi_{science}(x) = 1.0$ and distributional with $\mathbb{E}\phi_{female}(x) = 0.5$. GDC in a single training procedure is able to increase the expectation of biographies about females from $7.4\%$ to $28.8\%$ and Science professions from $1.2\%$ to $74.7\%$.

| | |
|---|---|
| **Art Professions Biographies** | |
| F | oraci martínez rubin ( born october 24, 1982 ) is a puerto rican actress, dancer and model. she was the first puerto ... |
| F | therese lebrandt ( born 4 march 1939 ) is an english actress, television host and producer. she is known for her roles as lily lenox... |
| - | , better known by his stage name zac banezi, is an israeli singer and songwriter. the producer of many artists, as well as the keyboardist of heavy metal band the.. |
| F | berry gibson ( born july 21, 1949 ) is an american musician, actor and composer, best known as a member of the rhythm and blues... |
| - | balkrishnan dev is an indian actor who is known for his roles in telugu movies. he began his career with a short supporting role in " sapikaya ". later he played .. |
| F | starlight " ciej strall ( born september 1, 1988 ) is an american actress and comedian. she is best known for her role as el ... |
| - | quentin brantley ( born april 27, 1973 ) is a canadian actor, composer, director, writer and producer. he is best known for his work.. |
| - | "Álvaro olajerra " is an argentine comedian and actor. in 1983, he won an episode of céspedes justicialiste de bolaños.. |
| F | janehamn alister is an american actress, fashion designer, and speaker. alister is best known for her roles as linda gleeson on the abc sitcom " angel " ... |
| - | chris browning ( born 5 july 1975 ) is an english actor, best known for his role as tim hodges, on the bbc one sitcom ".. |
| - | andy papadelaspe ( born 9 july 1973 ) is a french actor and director. he is known for his performances in several feature films including " bern .. |
| **Science Professions Biographies** | |
| - | ters g. g. engeland ( 14 april 1914 – 4 september 2002 ) was an american astronomer and senior researcher in astrophysics. he.. |
| F | thene ted ( born april 4, 1967 ) is an american science educator, student, medical research scientist and medical researcher. she is a director of mls ... |
| F | alexandra martin thomas ( born march 2, 1978 ) is a nigerian scientist and sociologist, researcher and writer. she is the current president of ... |
| - | quentin jacobsen ( born 1952 ) is a philosopher. he is a senior fellow at the center for progressive studies, where he teaches philosophy and is responsible for.. |
| - | edgar yanowicz ( born 26 july 1940 ) is a philosopher, sociologist and translator who lives and works in new york city. yanowicz is.. |
| F | antosia rose ( born 4 april 1962 ) is an english philosopher. she is a fellow of the royal society and a visiting fellow of the royal academy of engineering... |
| F | cornelius roberts ( 25 october 1756 – 17 december 1818 ) was a philosopher of science, well known as a marxist during the .. |
| - | mathias friedrich attelet ( 4 may 1916 – 11 november 2010 ) was a german philosopher. he was a specialist on number theory, algebraic.. |
| F | helped moore ( february 27, 1918 – january 25, 1980 ) was a historian and college president who was active in the civil rights movement and has written.. |
| - | mathias friedrich attelet ( 4 may 1916 – 11 november 2010 ) was a german philosopher. he was a specialist on number theory, algebraic.. |
| - | themen, jimmy and charles " ( december 25, 1960 ) is an american philosopher. he is a visiting professor at the university of mich.. |
| **Business & Politics Professions Biographies** | |
| - | said thai khalid (, born 1947 ) is a burmese novelist, journalist and politician. his career began in 1962 and he has become a leader of the .. |
| - | viscount knippenstern ( 14 november 1737 – 5 august 1792 ) was an austrian-born german jurist and politician. .. |
| F | alfreda rochelle, ( may 10, 1877 – november 4, 1965 ) was a canadian lawyer, judge and judge. she served as... |
| - | theodor radulović ( ; 30 october 1873 – 18 november 1960 ) was a croatian statesman, diplomat, and military officer, .. |
| - | charles lawrence ( april 19, 1807 – april 30, 1876 ) was an american politician and soldier. he served as a union general during .. |
| F | i subon ( ; born january 18, 1982 ) is an israeli journalist, writer, columnist and journalist. she is known as the first women writer to... |
| F | hiyat haza (, born 1959 ) is a somali politician. she has been a member of the parliament of somalia from june 2009 to april... |
| - | erik wiemens ( born 11 october 1957 ) is a german politician. as a youth, he participated in a number of parties, most notably.. |
| F | atalie castillo gonzález ( born 26 april 1957 ) is a mexican politician affiliated to the institutional revolutionary party. as of 2014 she served as deputy... |
| - | ashaun " tom " hicks ( born july 28, 1986 ) is an american actress, singer, and beauty pageant contestant. he is also a journalist and .. |
| - | izhev, born " yuri aleksandrovich isov " ( ; ), was a writer, journalist and politician. isov first became active in.. |
| **Sports Professions Biographies** | |
| F | isaba aguirre ( born 10 february 1983 in Éixidat, france ) is a female volleyball player from spain. she is a... |
| F | hanyu pratak ( born 11 june 1993 ) is a female badminton player from bangladesh. she is also an eventer and former world... |
| - | alexandre nicolau ( born 16 february 1989 in travancore ) is an italian professional footballer who plays for serie b club acf.. |
| - | yury novoshenko ( ; born march 14, 1987 in tokushima ) is a russian professional football player. in 2011, he played in the.. |
| F | eina jena ( born july 12, 1981 ) is an american soccer player currently playing for ca pei in the chinese super league. she also formerly... |
| F | chiyo zuai ( born 18 april 1979 in taipei ) is a retired taiwanese tennis player. she is the 1996 olympic... |
| F | patti ann rakic ( born 23 february 1990 ) is an australian former synchronized swimmer who competed at the 2012 summer olympics. her... |
| - | christopher " chris " saul ( born 31 march 1964 ) is a scottish former professional footballer and manager, who managed.. |
| F | katja shearer ( born 7 july 1994 ) is a swedish footballer who plays as a goalkeeper for grödig tyngall. shearer started.. |
| - | was an indian chinese footballer who played for hong kong first division league team shandong luneng f.c. during the 1980s. he was regarded as the best right-.. |
| - | andrey sivchenko ( born 8 july 1983 ) is a russian swimmer. he competed in the men's 200m butterfly event at the 2012 summer.. |
| - | campbell anderson ( born september 10, 1951 ) is a former professional american football player. he played four seasons with the indianapolis colts of .. |
| - | amın güntur ( born 26 may 1987 ) is a turkish professional footballer who plays as a goalkeeper for kozlu kiznevetspor.. |

**Table 7:** Randomly selected generations from the hybrid Experiments (3,4,5,6). **F** indicates that the generation is about a female character. The imposed distributional constraint is $\mathbb{E}\phi_{female}(x) = 0.5$, while the pointwise constraint is $\mathbb{E}\phi_{art}(x) = 1.0$, $\mathbb{E}\phi_{science}(x) = 1.0$, etc.

# H    EXTRA DETAILS ON POINTWISE EXPERIMENTS

## H.1    APPROXIMATING THE DESIRED $p$ DISTRIBUTION

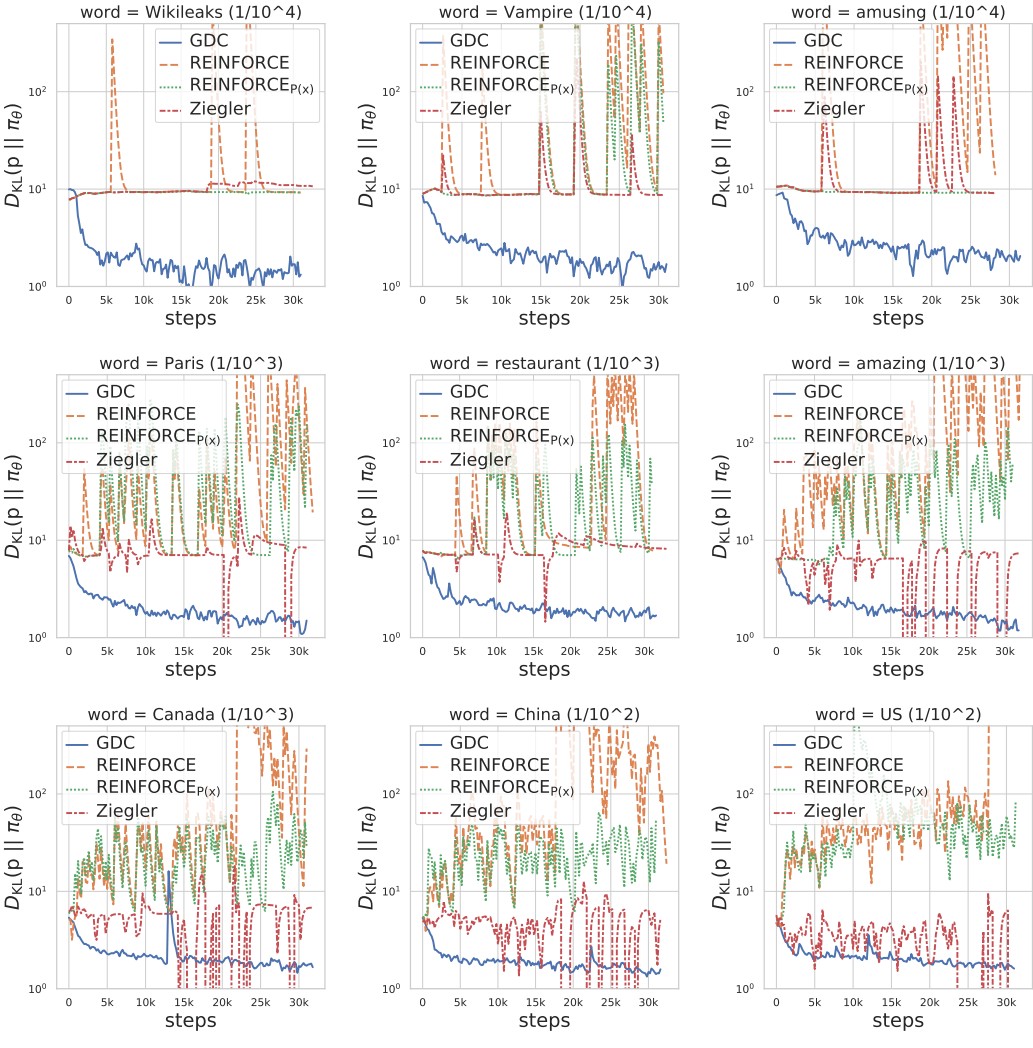

**Figure 15:** $D_{\mathrm{KL}}(p, \pi_\theta)$ against the training steps for GDC and the three baselines introduced in section §3.2 for the single-word control task. Curves are displayed for nine different single-word constraints of varying rarity levels (1/100, 1/1000, 1/10000). GDC exhibits much better convergence behaviour than the other baselines, showing its superiority in approximating the desired distribution $p$.

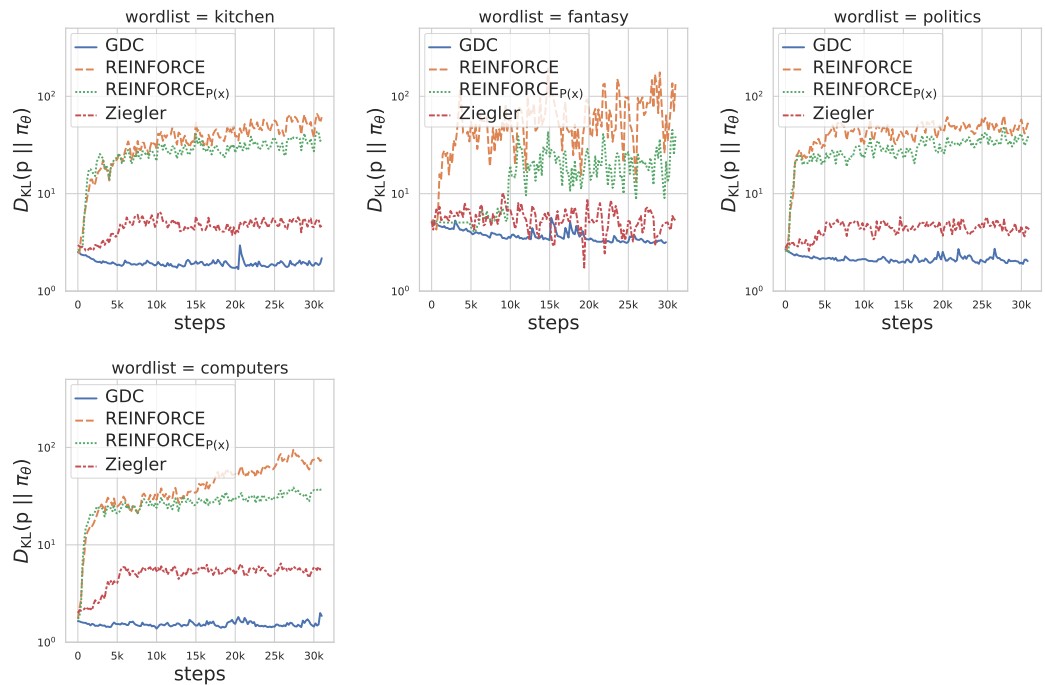

**Figure 16:** $D_{\mathrm{KL}}(p, \pi_\theta)$ against the training steps for GDC and the three baselines introduced in section 3.2 for word-list constraints. Curves are displayed for 4 word-lists: kitchen , fantasy, politics, computers. GDC exhibits much better convergence behaviour than the other baselines, showing its superiority in approximating the desired distribution $p$.

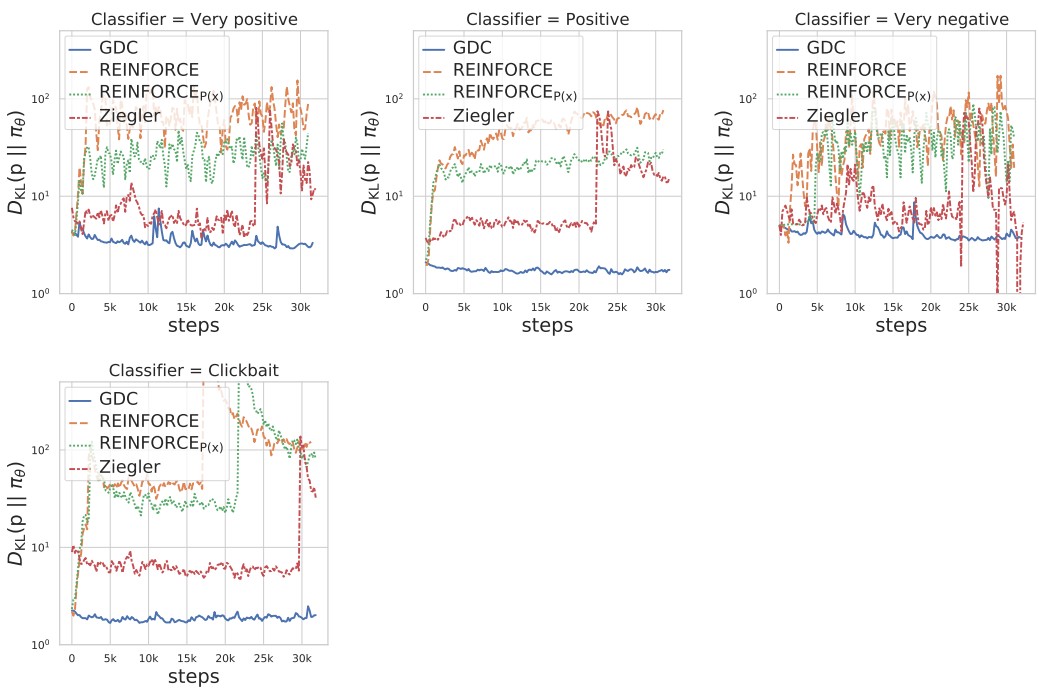

**Figure 17:** $D_{\mathrm{KL}}(p, \pi_\theta)$ against the training steps for GDC and the three baselines introduced in section 3.2 for classifier-based control. Curves are displayed using 4 different classifiers: very positive, positive, and very negative sentiment, and click-bait. GDC exhibits much better convergence behaviour than the other baselines, showing its superiority in approximating the desired distribution $p$.

## H.2 More Details on Point-wise Constraints Experiments

## H.3 Pointwise Constraints

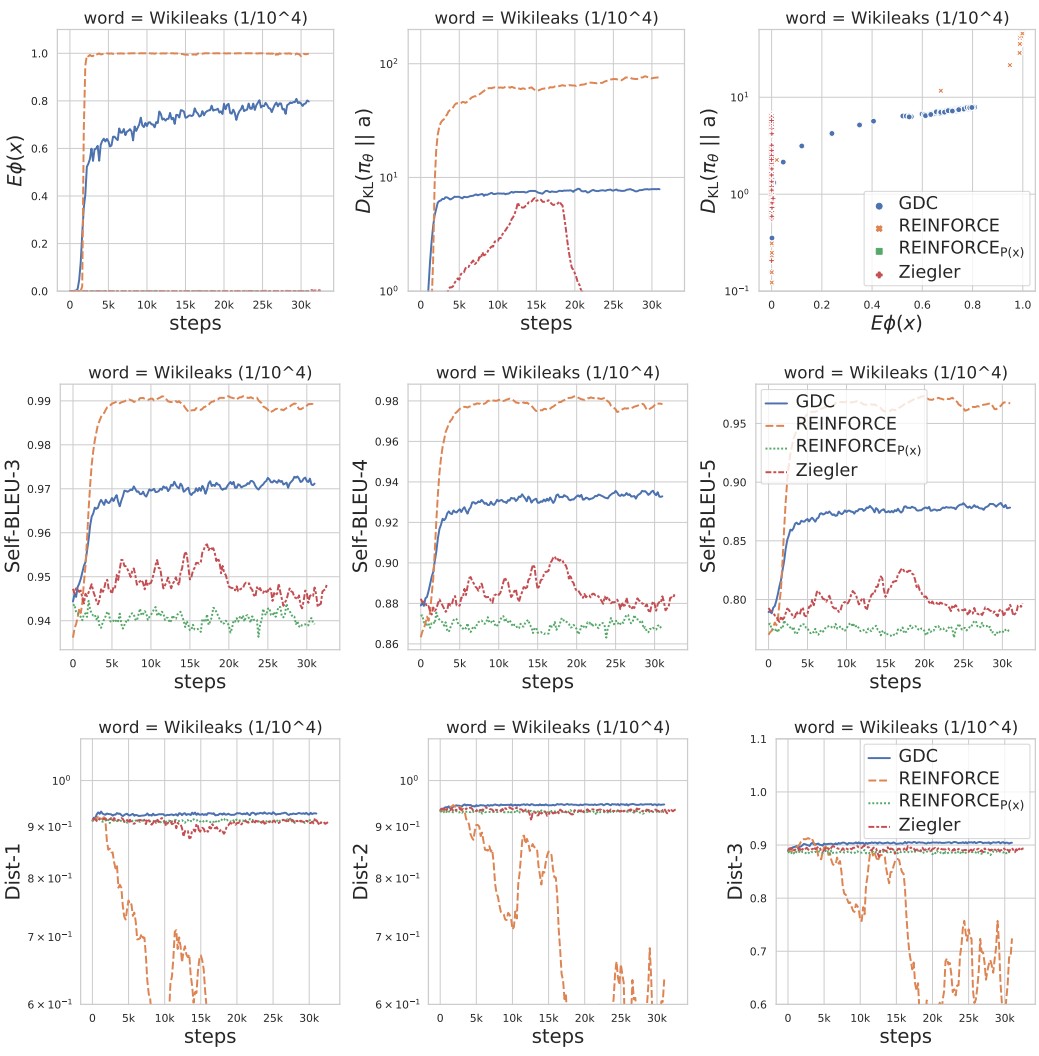

**Figure 18:** Line plot of different evaluation metrics against the training steps when controlling for the word "wikileaks" (with initial occurrence probability of 1/10000) as a single-word constraint.

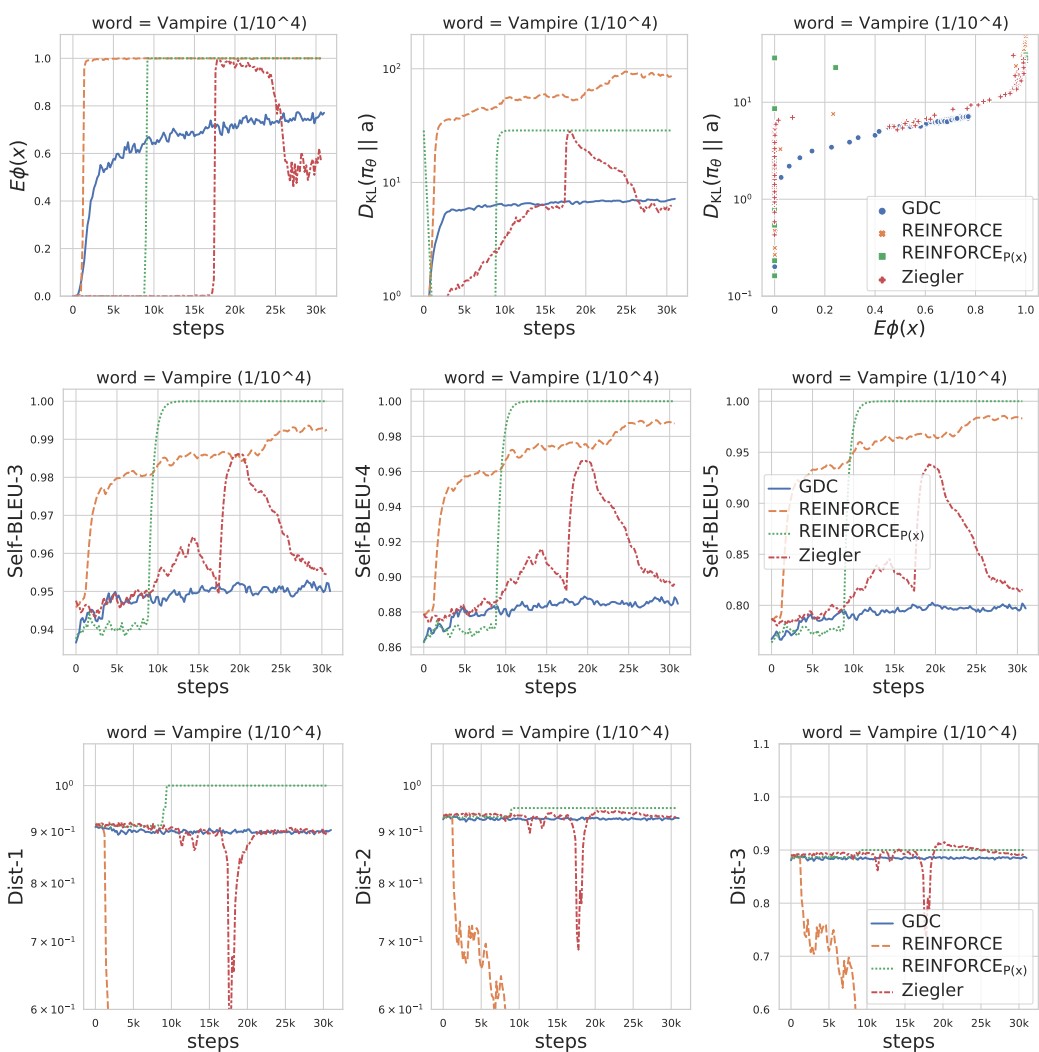

**Figure 19:** Line plot of different evaluation metrics against the training steps when controlling for the word "vampire" (with initial occurrence probability of 1/10000) as a single-word constraint.

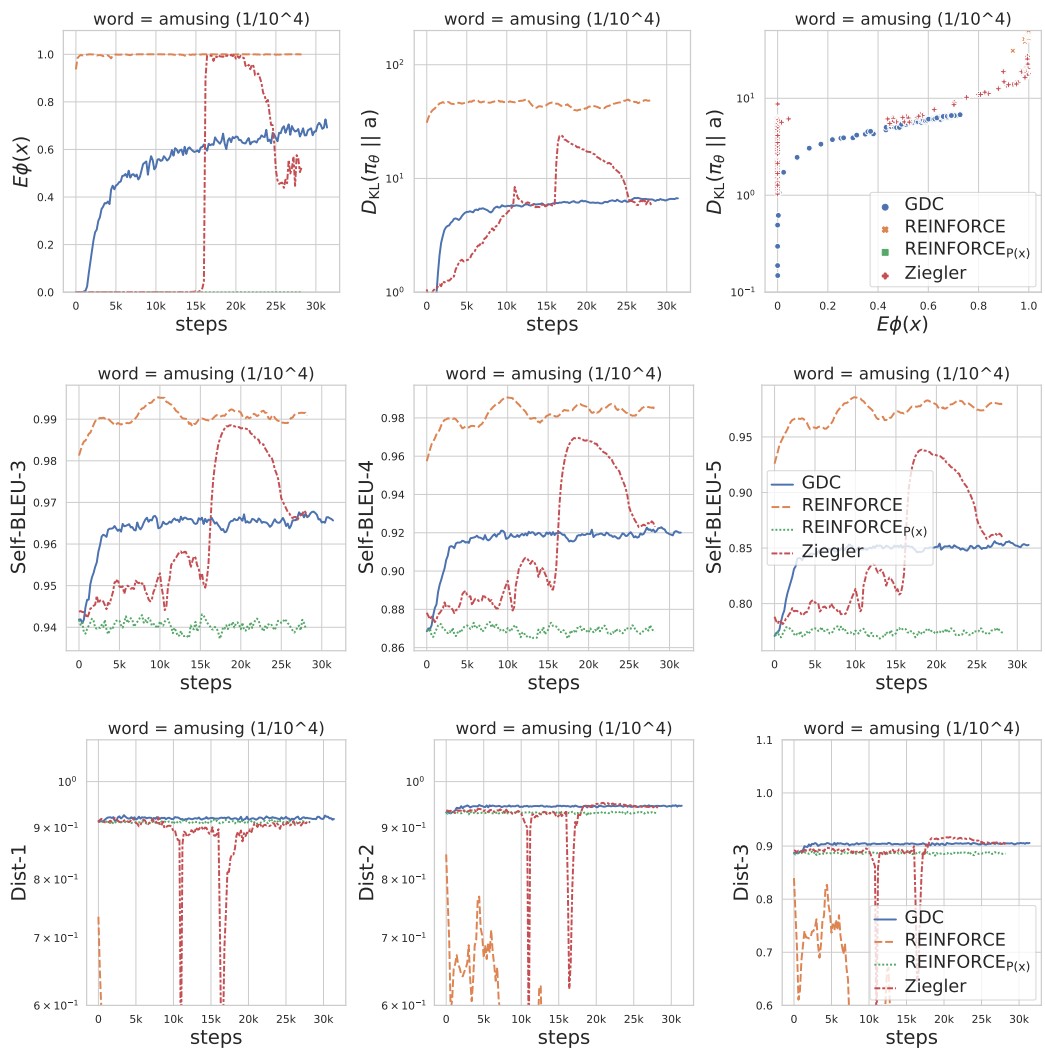

**Figure 20:** Line plot of different evaluation metrics against the training steps when controlling for the word "amusing" (with initial occurrence probability of 1/10000) as a single-word constraint.

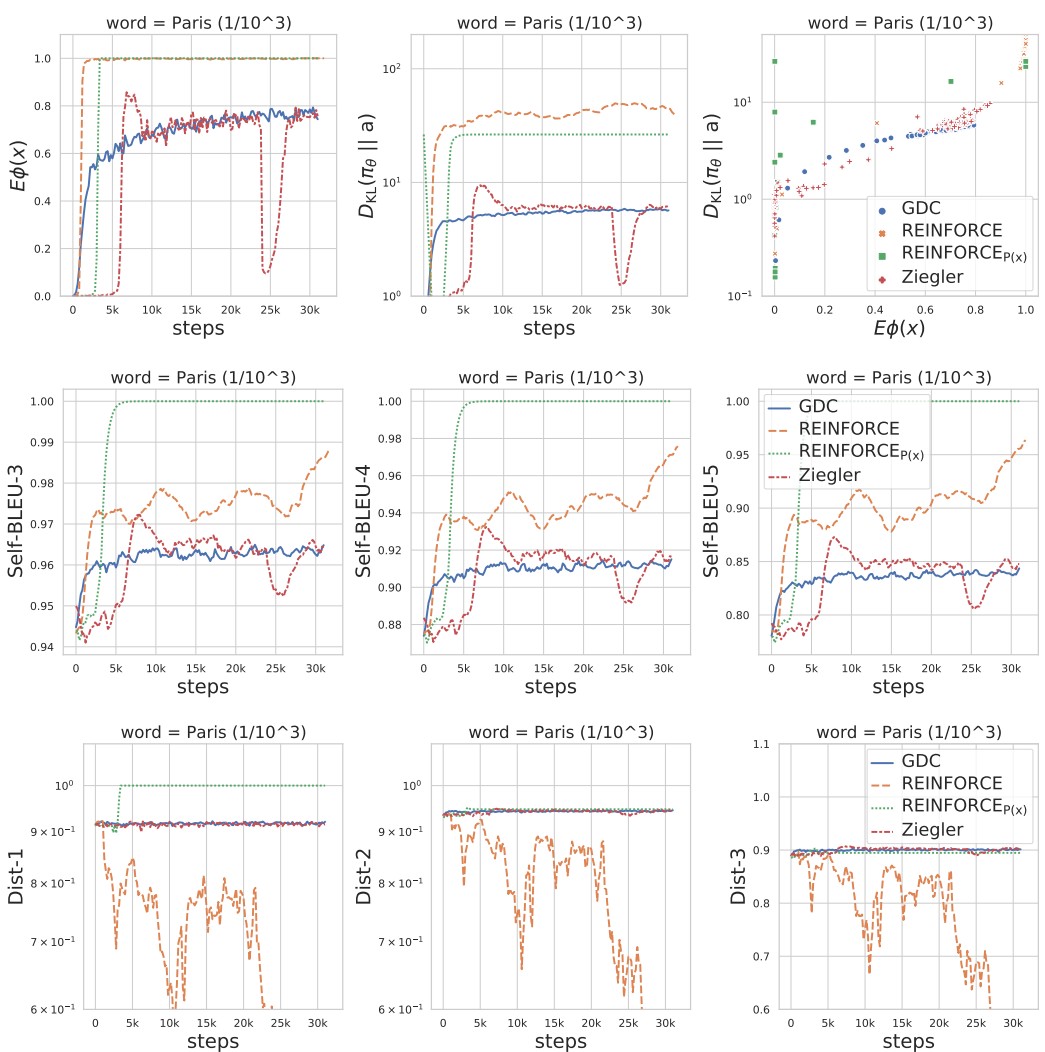

**Figure 21:** Line plot of different evaluation metrics against the training steps when controlling for the word "Paris" (with initial occurrence probability of 1/1000) as a single-word constraint.

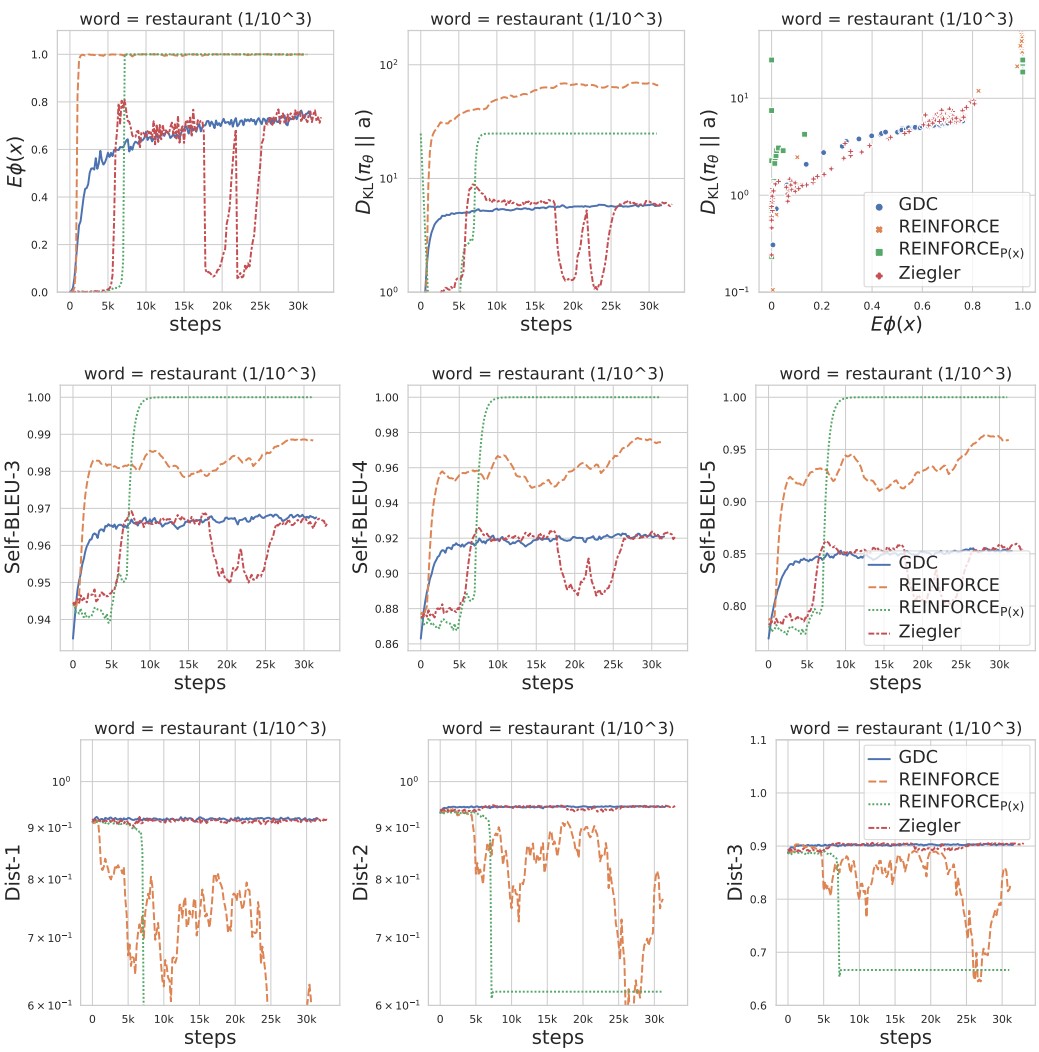

**Figure 22:** Line plot of different evaluation metrics against the training steps when controlling for the word "restaurant" (with initial occurrence probability of 1/1000) as a single-word constraint.

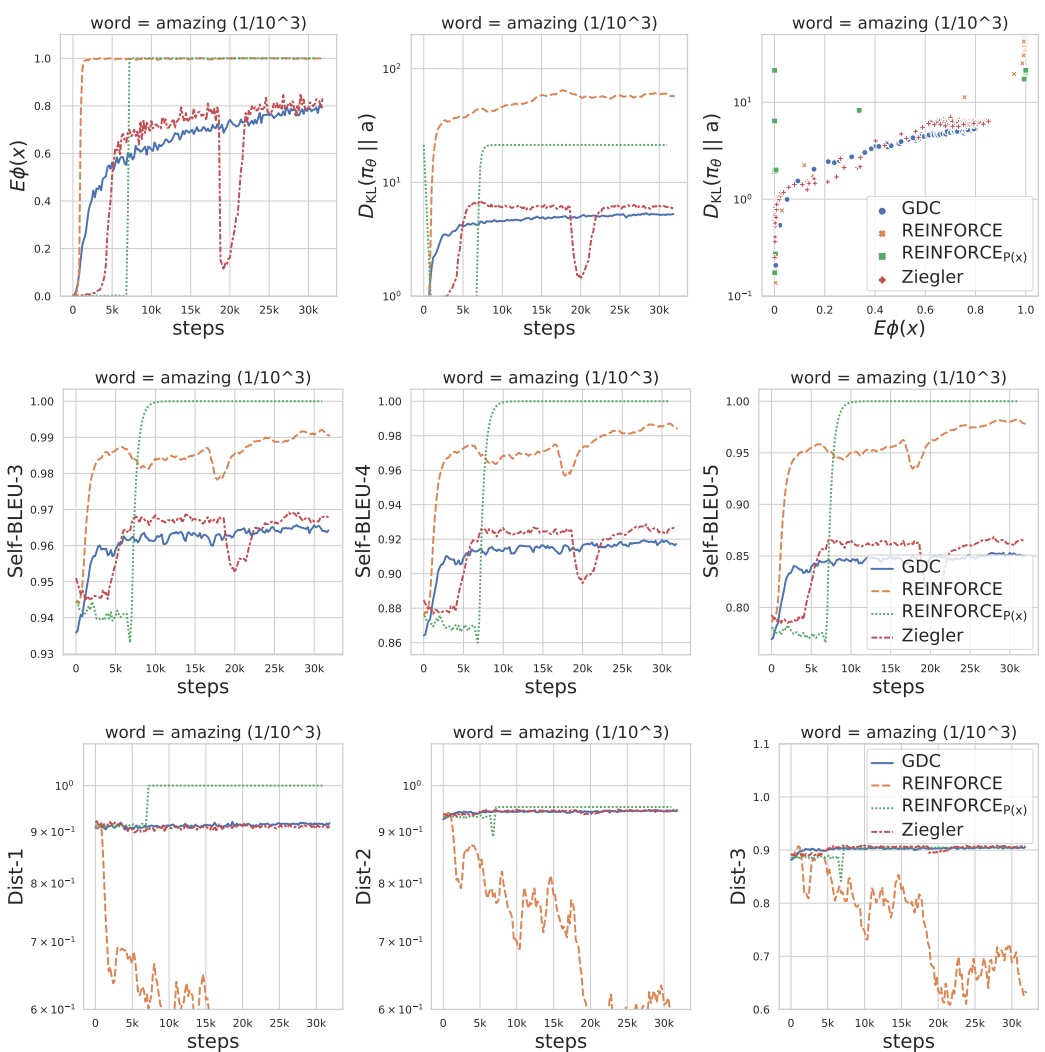

**Figure 23:** Line plot of different evaluation metrics against the training steps when controlling for the word "amazing" (with initial occurrence probability of 1/1000) as a single-word constraint.

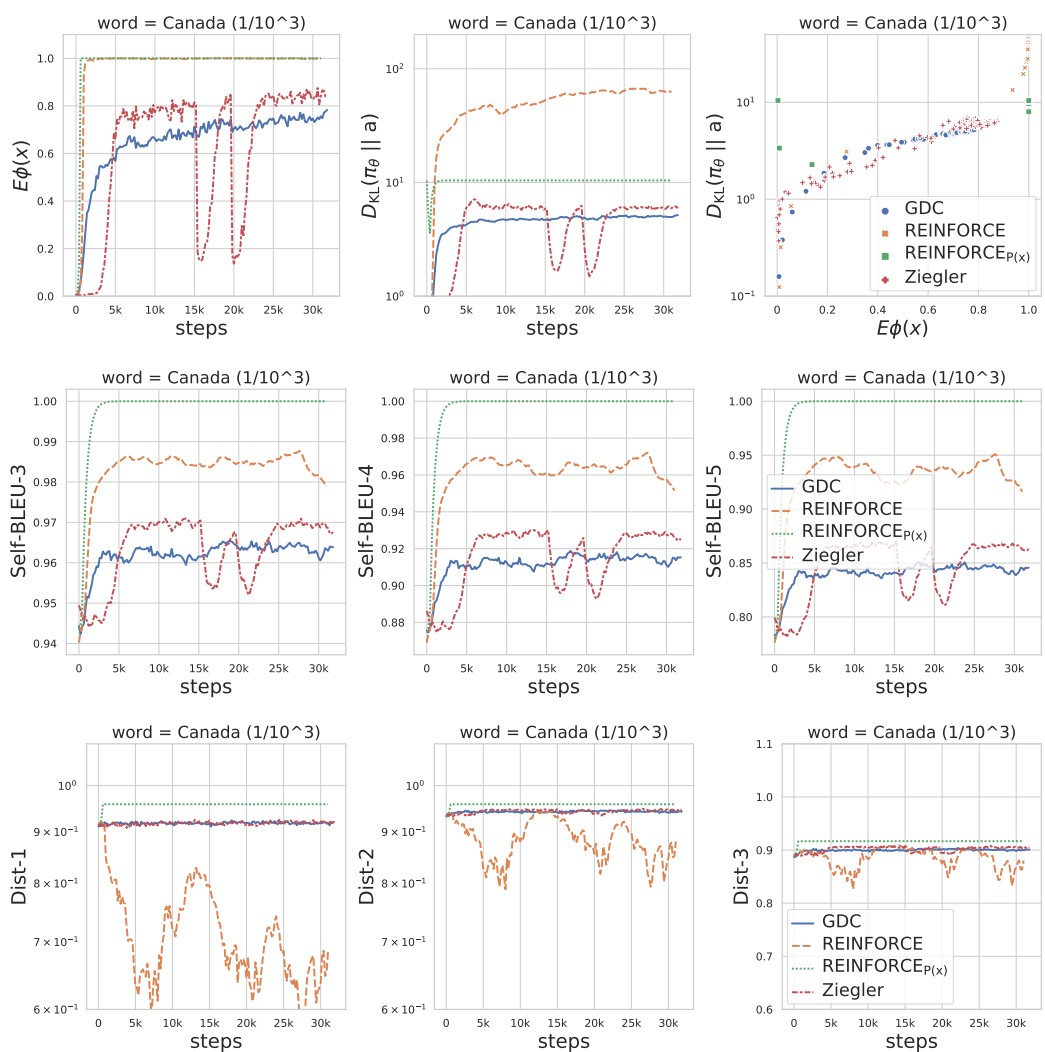

**Figure 24:** Line plot of different evaluation metrics against the training steps when controlling for the word "Canada" (with initial occurrence probability of 1/1000) as a single-word constraint.

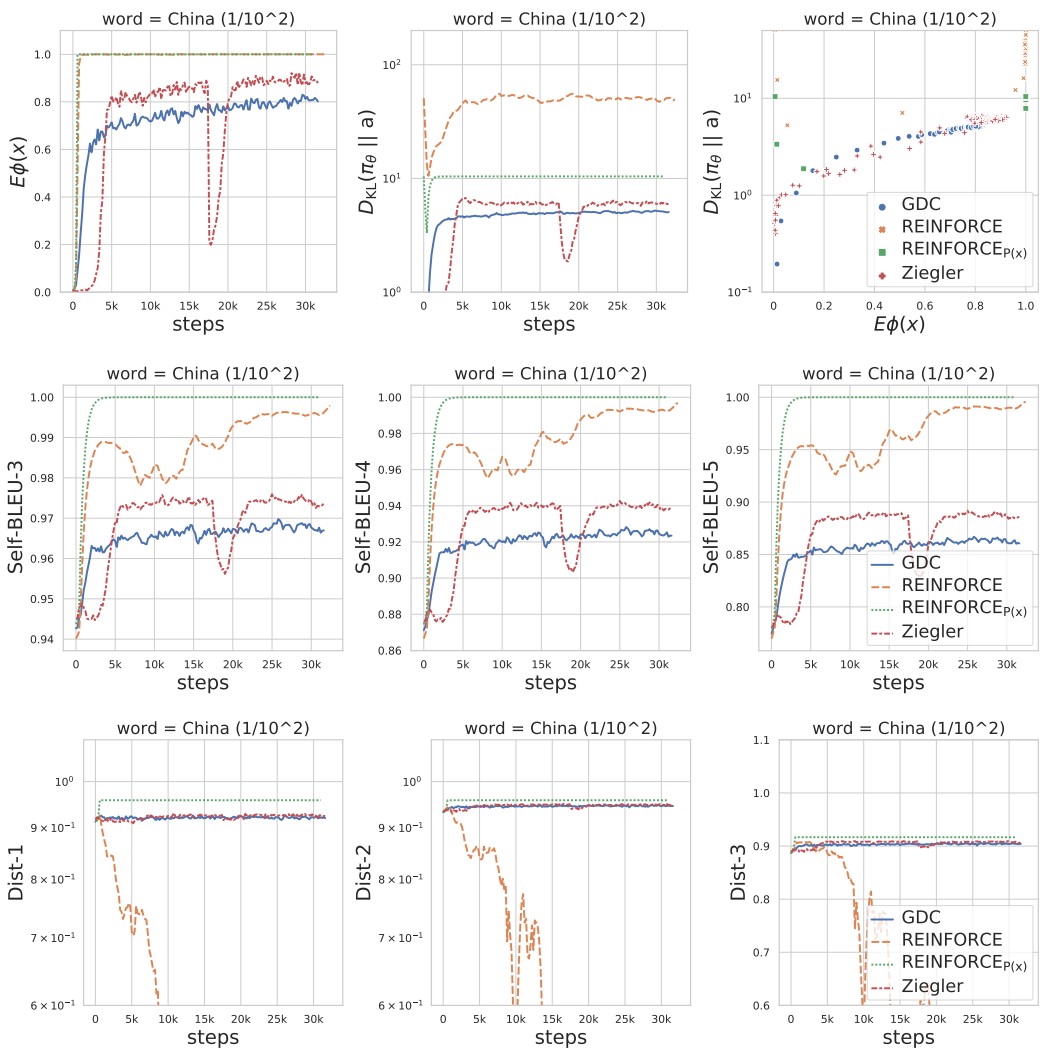

**Figure 25:** Line plot of different evaluation metrics against the training steps when controlling for the word "China" (with initial occurrence probability of 1/100) as a single-word constraint.

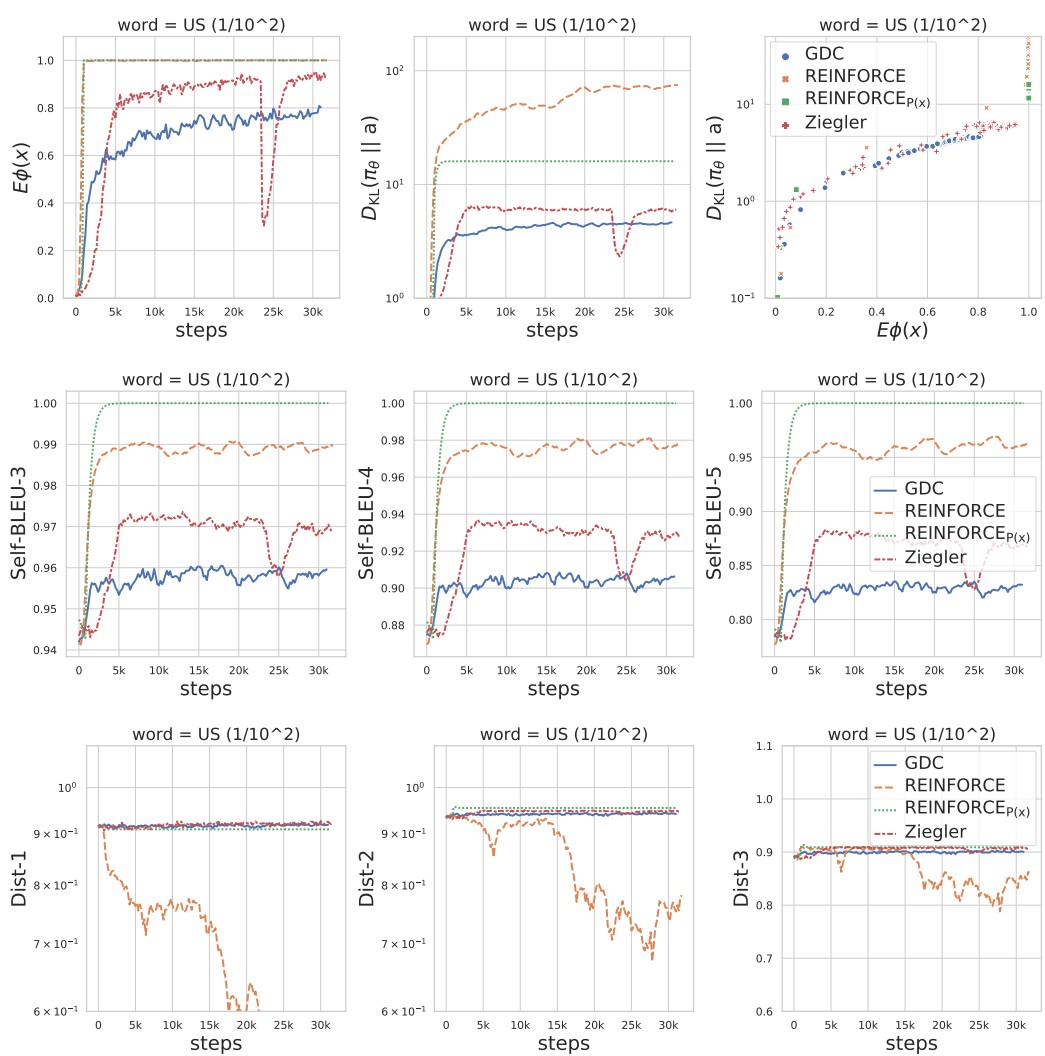

**Figure 26:** Line plot of different evaluation metrics against the training steps when controlling for the word "US" (with initial occurrence probability of 1/100) as a single-word constraint.

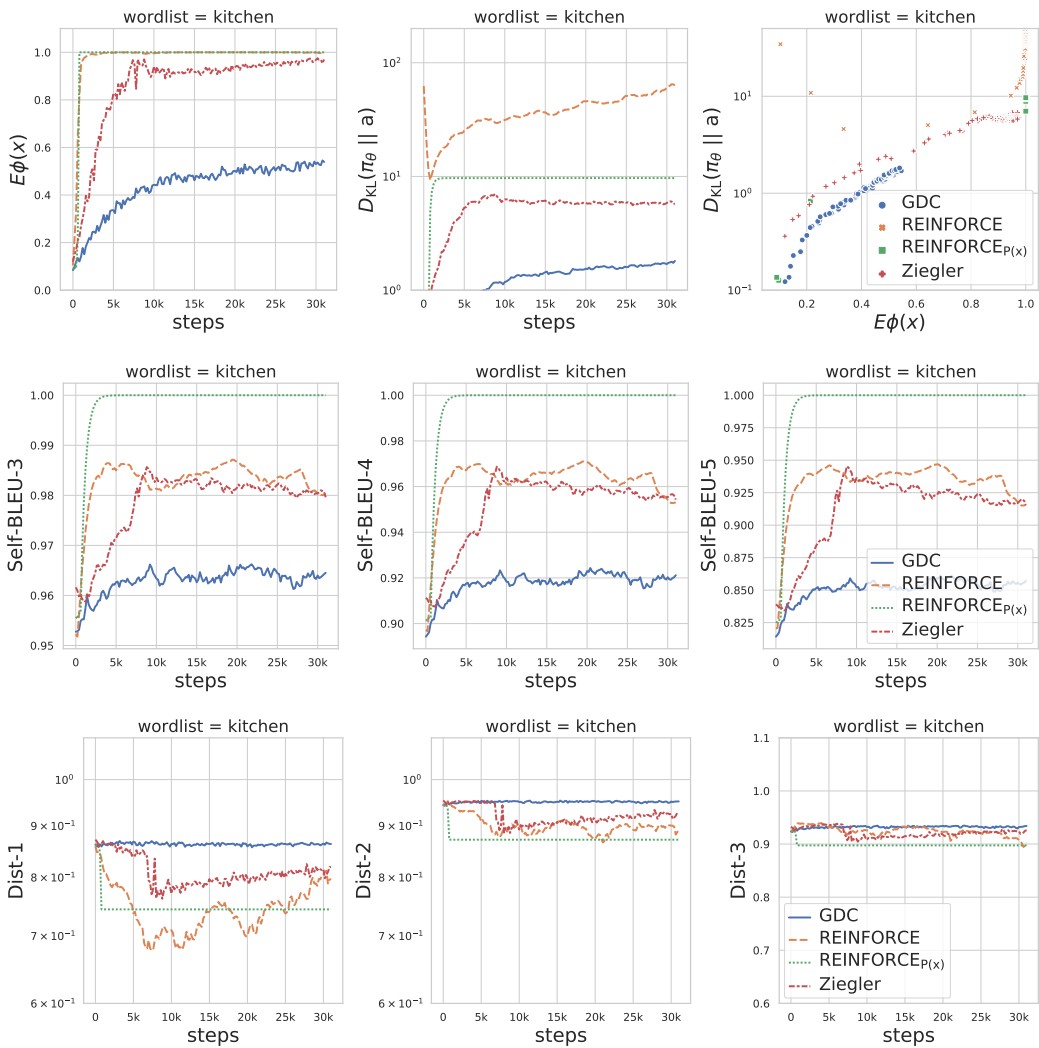

**Figure 27:** Line plot of different evaluation metrics against the training steps when controlling for the **kitchen** word-list.

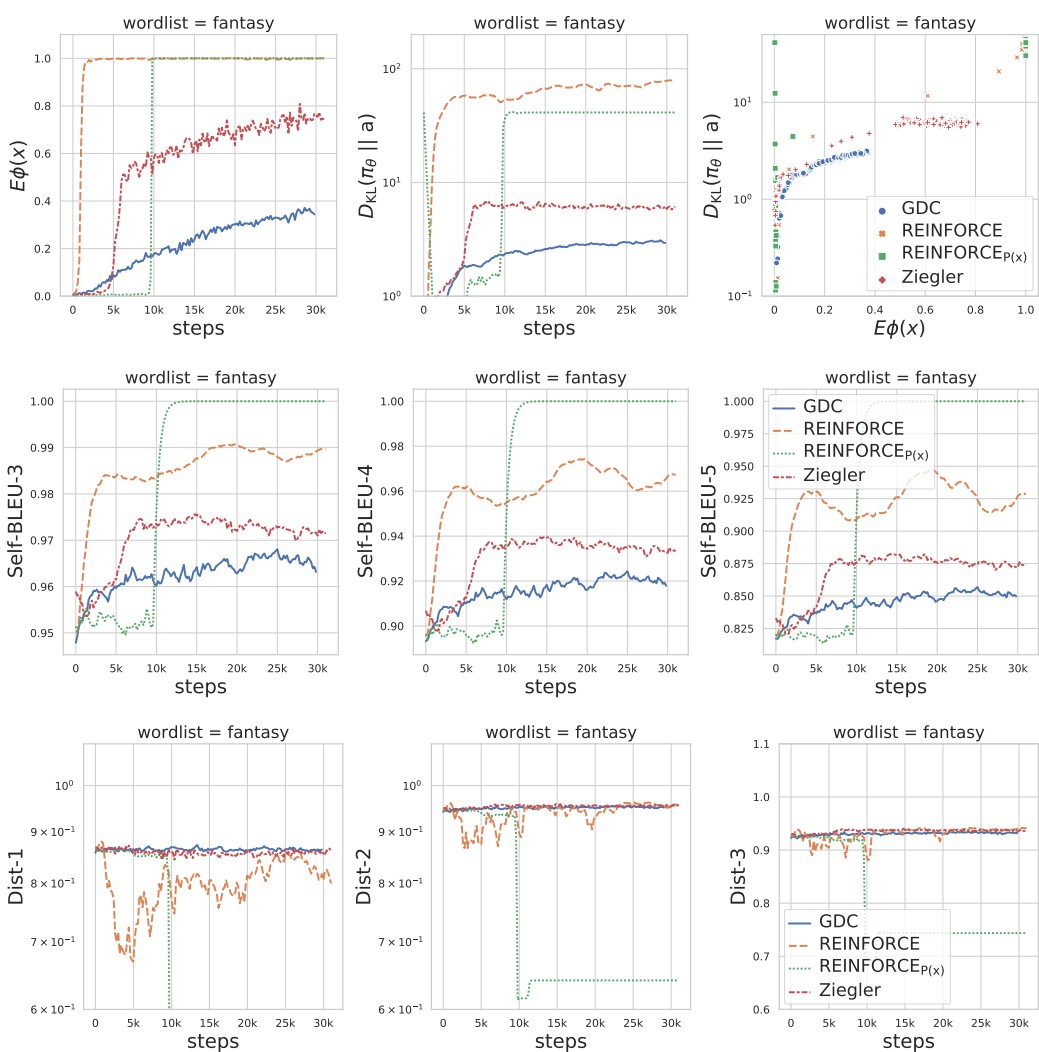

**Figure 28:** Line plot of different evaluation metrics against the training steps when controlling for the **fantasy** word-list.

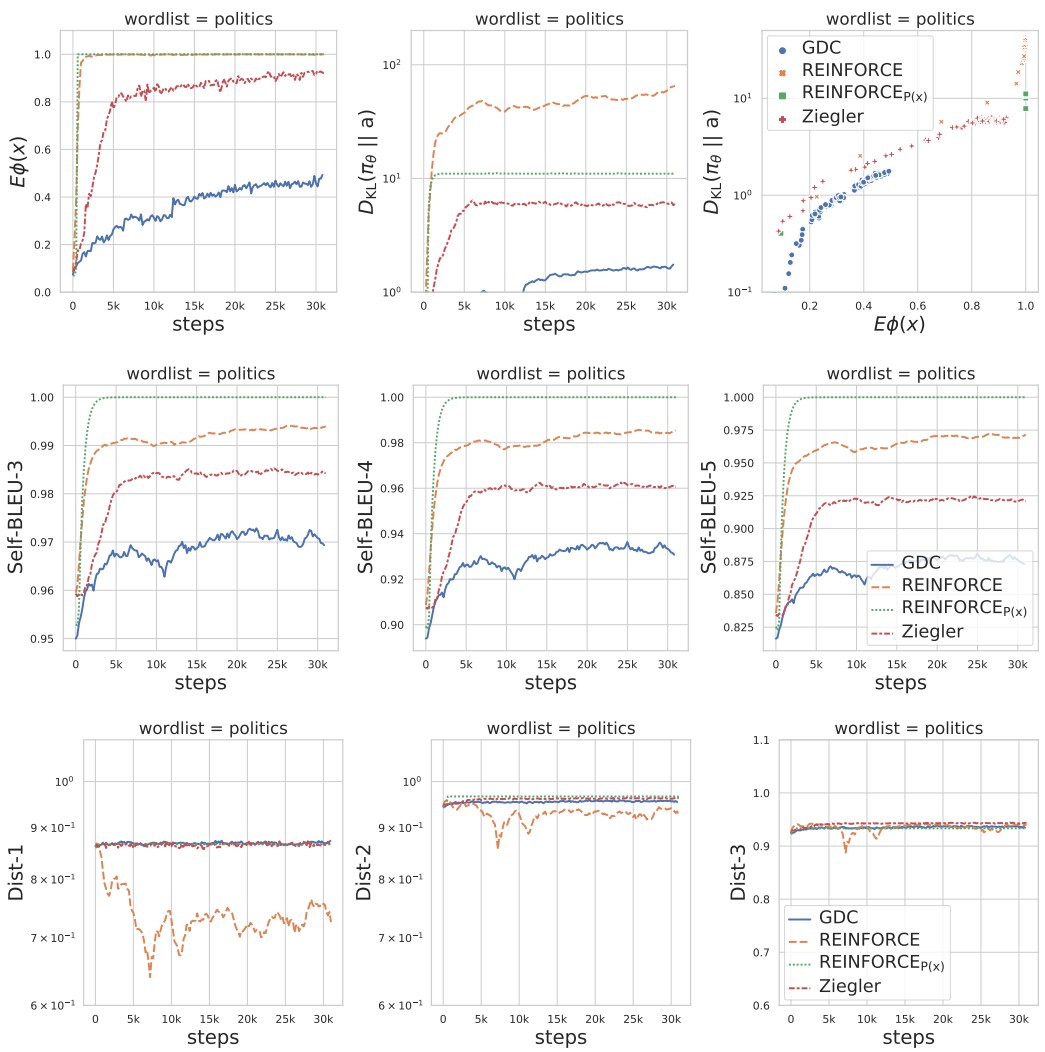

**Figure 29:** Line plot of different evaluation metrics against the training steps when controlling for the **politics** word-list.

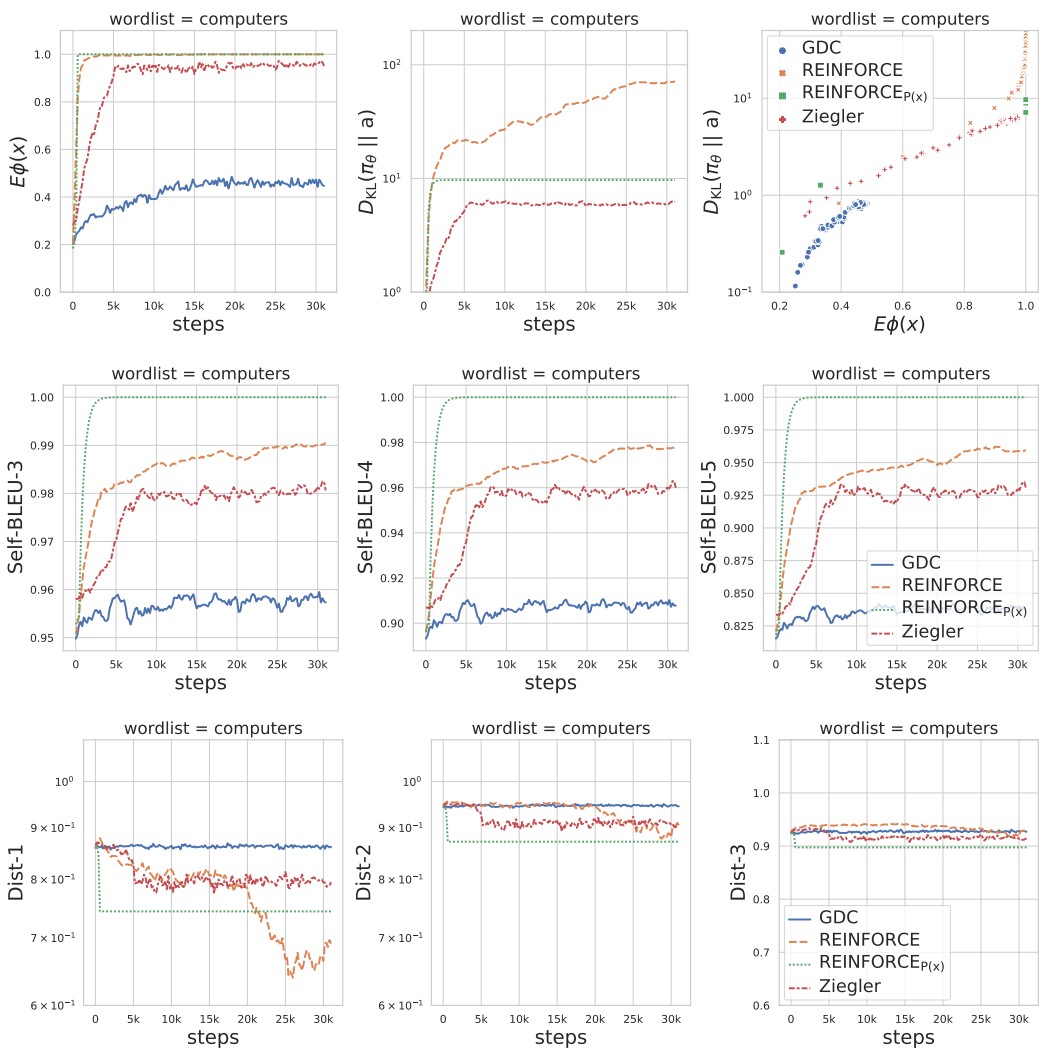

**Figure 30:** Line plot of different evaluation metrics against the training steps when controlling for the **computers** word-list.

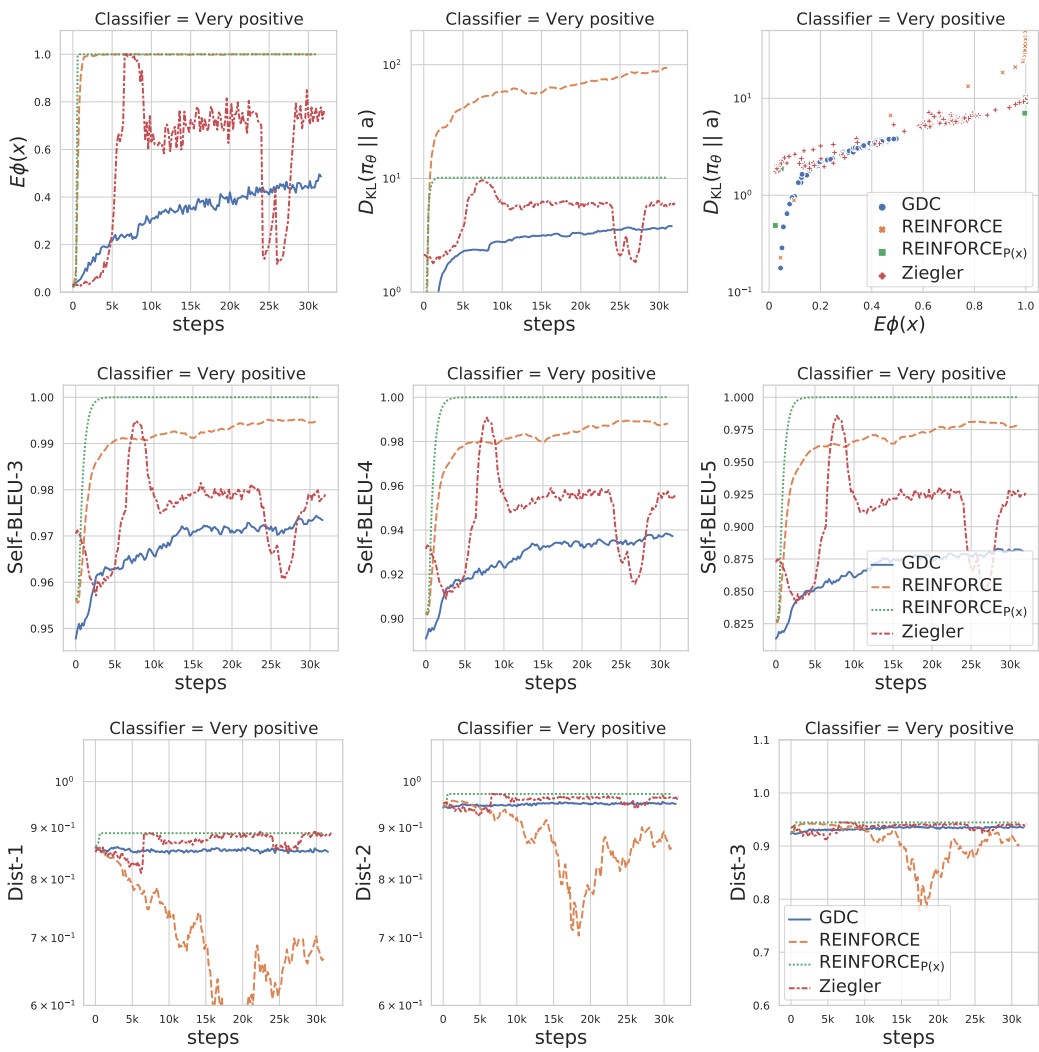

**Figure 31:** Line plot of different evaluation metrics against the training steps when controlling for the **politics** word-list.

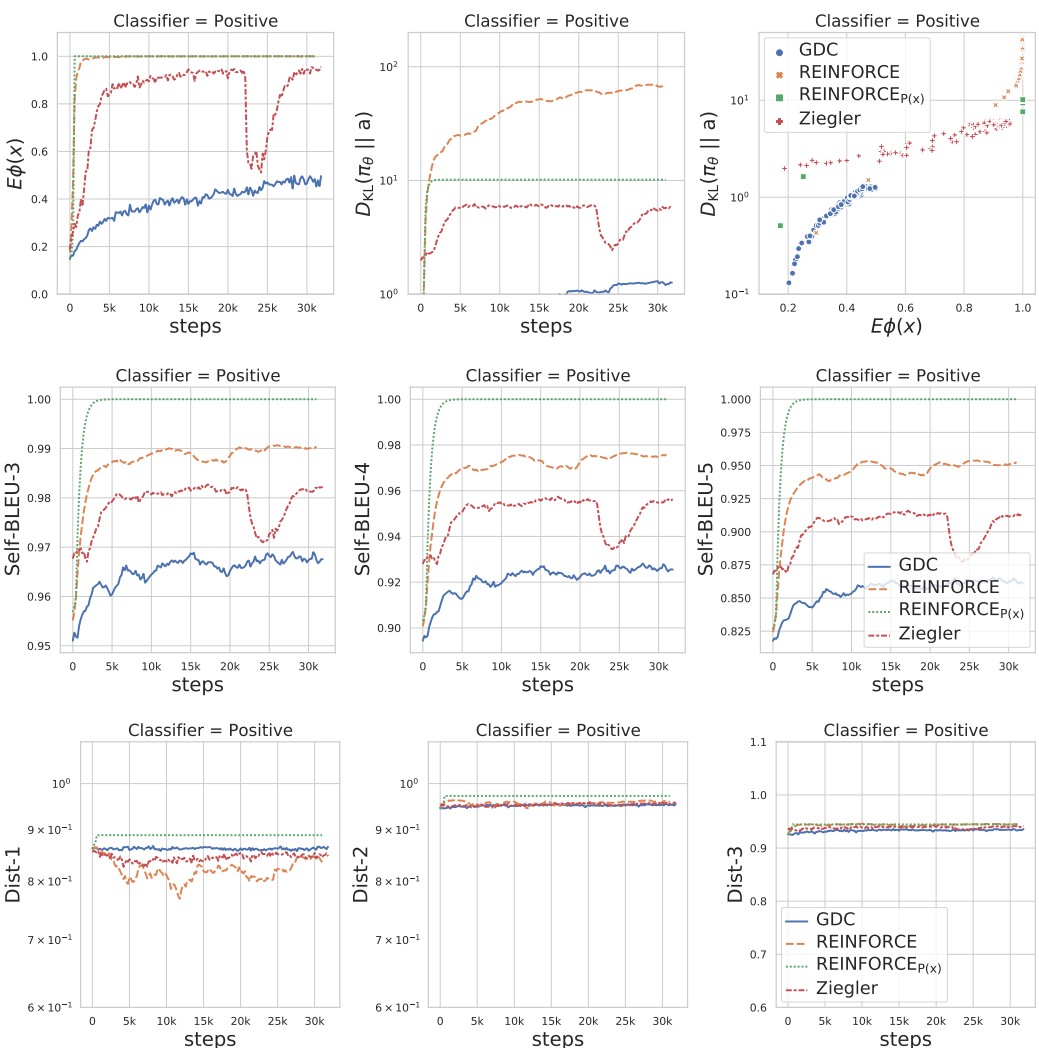

**Figure 32:** Line plot of different evaluation metrics against the training steps for **positive sentiment** classifier-based control with.

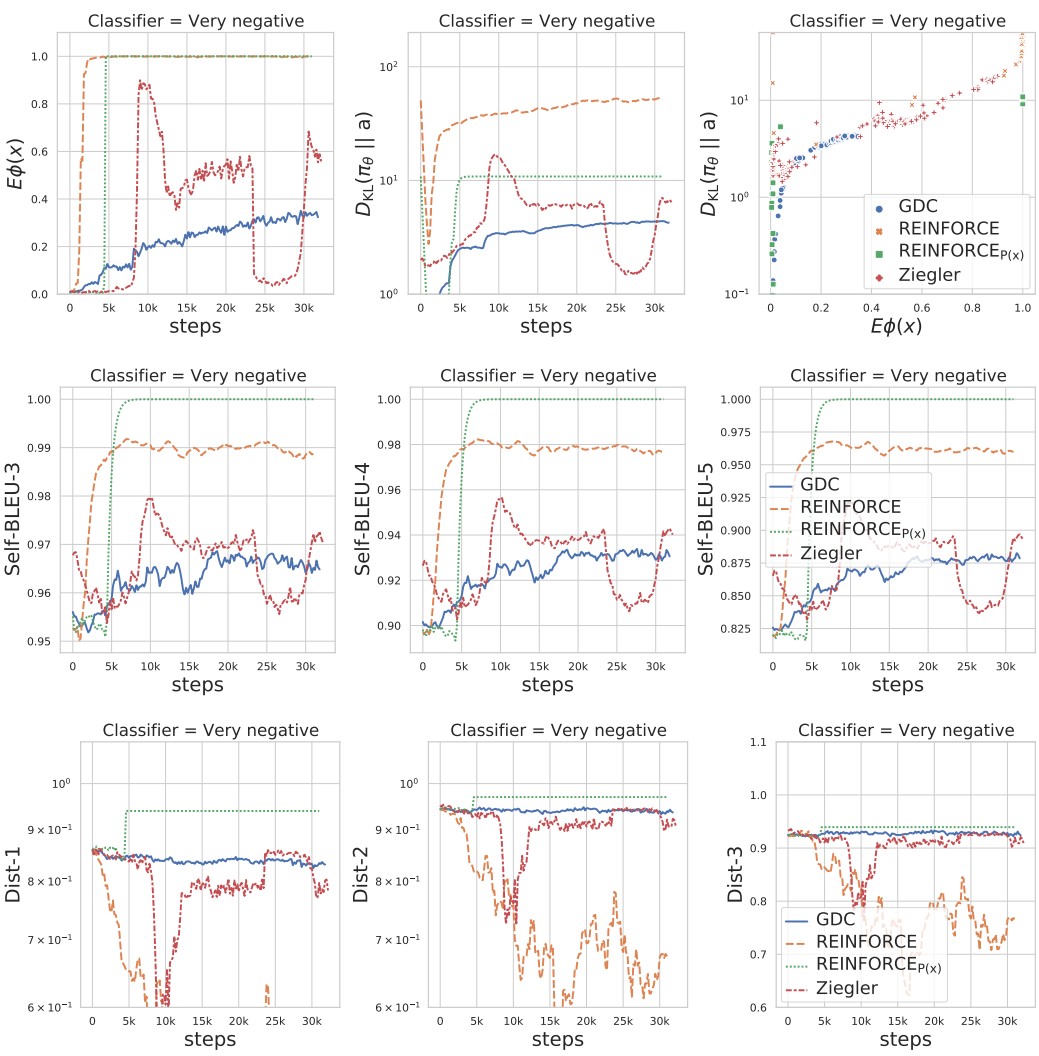

**Figure 33:** Line plot of different evaluation metrics against the training steps for **very negative sentiment** classifier-based control with.

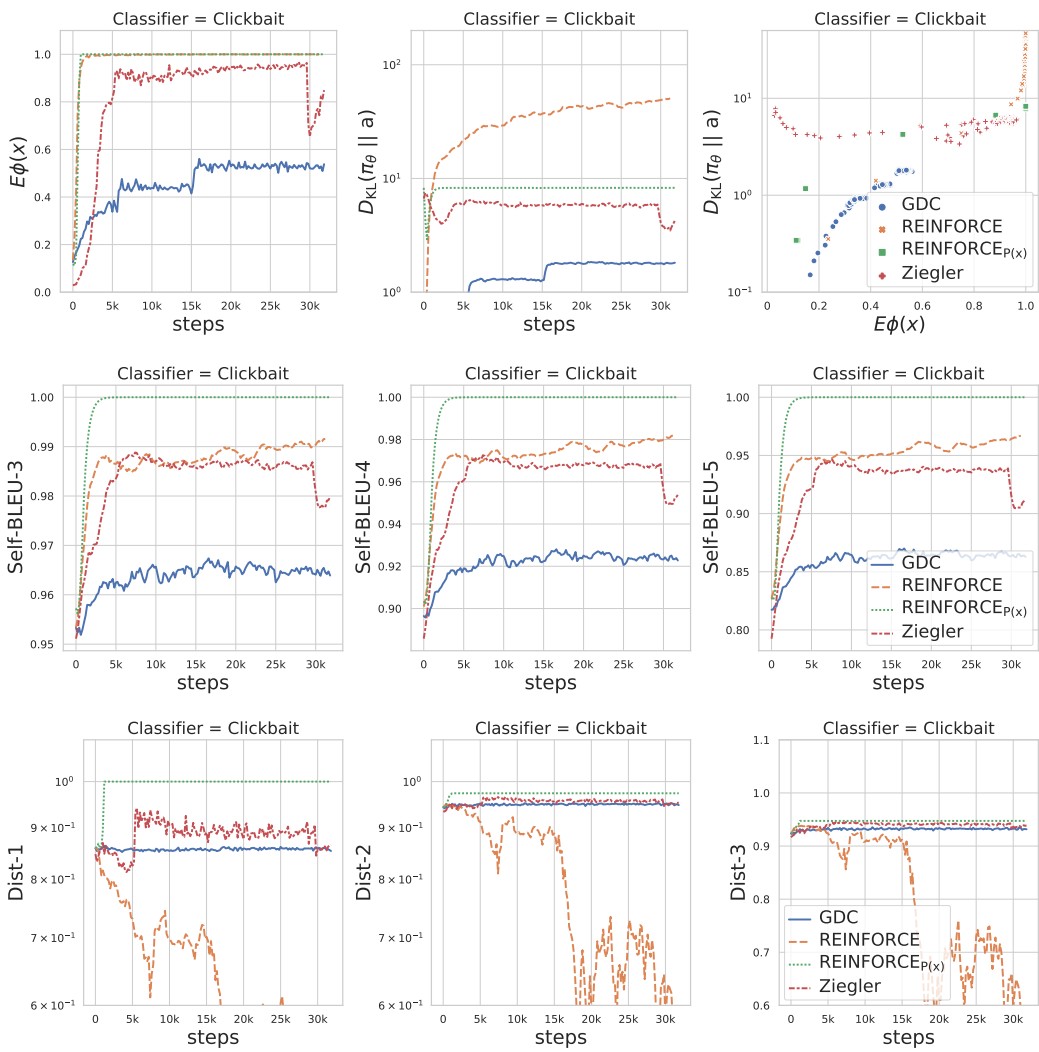

**Figure 34:** Line plot of different evaluation metrics against the training steps for **click-bait** classifier-based control with.

## H.4 TOKEN FREQUENCY ANALYSIS

To analyse in depth the effect of deviating much from the original GPT-2, for policies obtained from our method and each baseline, we obtain a large sample and filter to 4000 sequences that satisfy the imposed pointwise constraints for each of the 17 pointwise experiments explained in §3. Figures 35, 36 and 37 plot a token frequency analysis for each of the training methods.

The vanilla policy gradient baselines REINFORCE suffer from very low diversity of generations; in the examples shown in section H.5 we note strong degeneration, in which all generations are composed of a few repeated tokens.

REINFORCE$_{P(x)}$ suffers from a token diversity issue. As noticed and confirmed by generated examples shown section H.5, it often concentrates all the sequence probability mass on a single sequence which is often fluent and satisfies the constraint; however this leads to an extreme loss of sample diversity in almost all experiments. This shows the usefulness of our proposed analysis — in addition to the self-BLEU metrics — for distinguishing diversity at the sequence level or at the distribution level. Similarly, ZIEGLER (Ziegler et al., 2019) often suffers from the same lack of sample diversity (5 out of the 17 experiments); GDC obtains the highest diversity amongst all baselines, as demonstrated by the long tail in the figures below. It is important to note here that low sample diversity is also captured by the KL deviation from the original GPT-2 model i.e. $D_{\mathrm{KL}}(\pi_\theta \| a)$; GDC identifies the target distribution as the one which minimally deviates from the original policy while satisfying the constraints ($p = \arg\min_{q \in \mathcal{C}} D_{\mathrm{KL}}(q, a)$) is thus expected to preserve the high sample diversity of the original GPT-2.

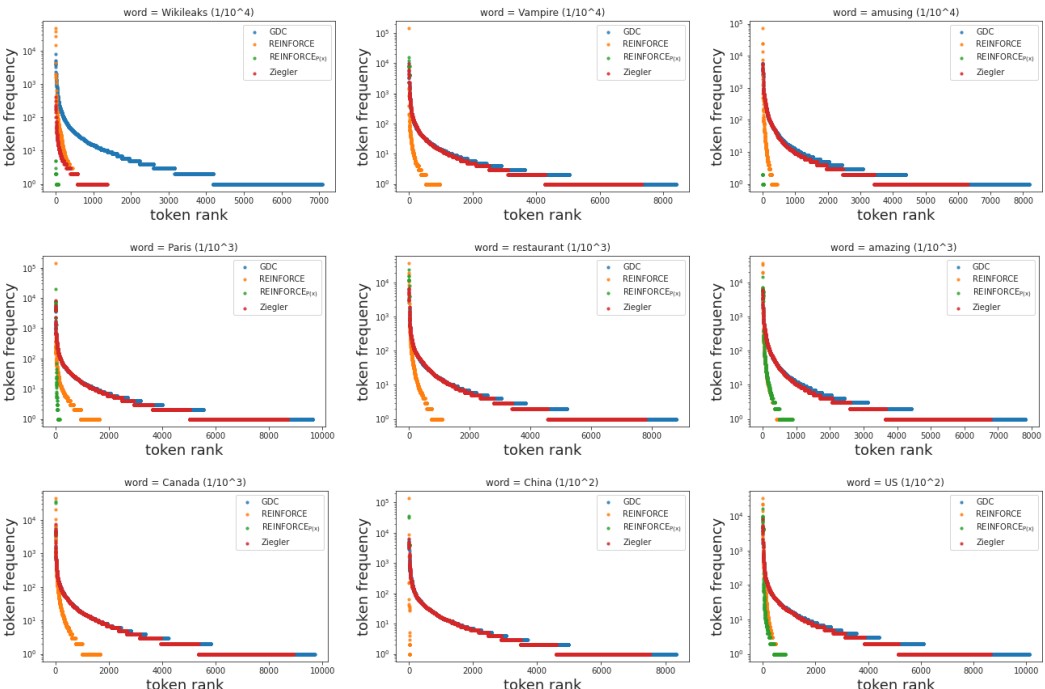

**Figure 35:** Token frequency against token rank for single-word constraints. Longer tail means more diverse generations.

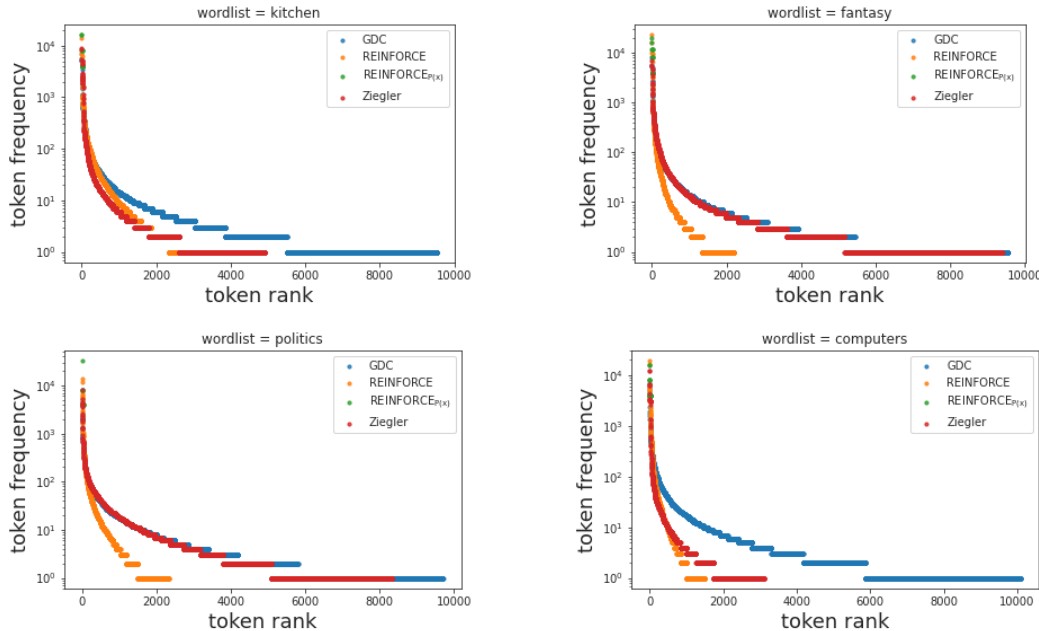

**Figure 36:** Token frequency against token rank for word-list constraints. Longer tail means more diverse generations.

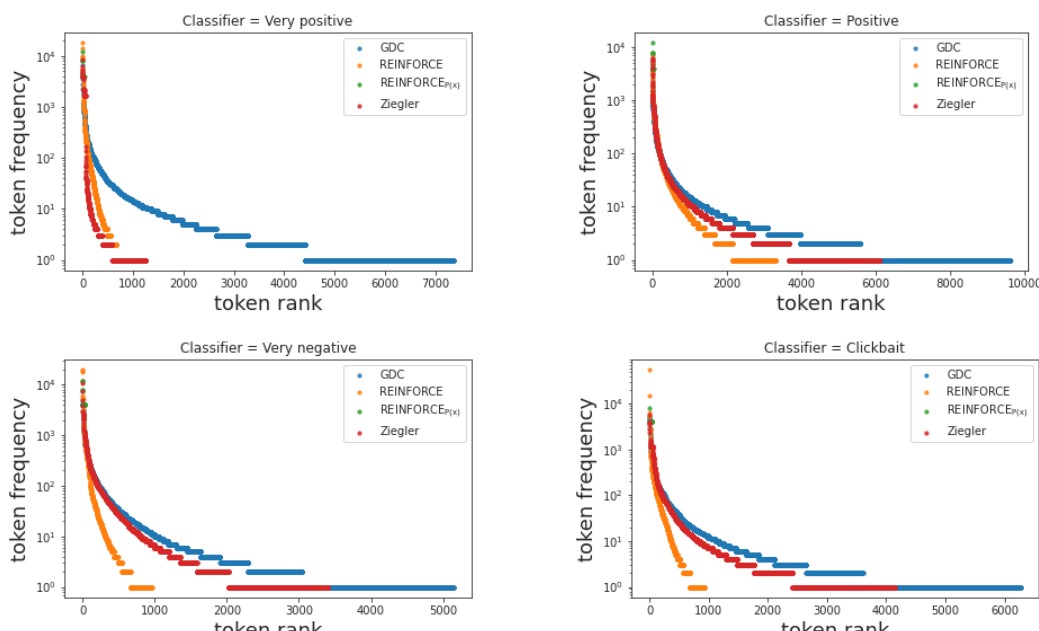

**Figure 37:** Token frequency against token rank for classifier-based constraints. Longer tail means more diverse generations.

## H.5 Generation Examples

| reps | $\phi(x)$ | Generation |
|---|---|---|
| | | **GDC** |
| 1 | 1 | Wikileaks founder Julian Assange will be on the witness stand as a witness to how President Barack Obama and Democratic presidential candidate Hillary Clinton colluded |
| 1 | 1 | Wikileaks said Monday that a former US official told them that if they had gone to Moscow , they could have been imprisoned by that |
| 1 | 1 | Last summer , Wikileaks released millions of emails to Wikileaks founder Julian Assange , which many believed were stolen from the same address by some |
| 1 | 1 | Wikileaks emails show that the Trump campaign was able to hack into email accounts held by the Democratic National Committee and Democrats . As a |
| 1 | 1 | . On July 1 , 2016 , Wikileaks published a statement in which it said : The DNC's cyber operations had been compromised , damaging |
| 1 | 1 | The Wikileaks website has released more than 3 , 000 internal emails from Hillary Clinton's private email server , which were previously posted on the |
| 1 | 1 | This is an edited transcript of "The Wikileaks Program , " a book written by Julian Assange for The Guardian and published by Wikileaks in |
| 1 | 1 | The FBI released an unprecedented batch of internal emails in February from an unnamed private contractor that were leaked to Wikileaks , saying they revealed |
| | | **REINFORCE** |
| 1 | 1 | The Republican party has been hacked by Wikileaks from Wikileaks from of Wikileaks from Wikileaks by Wikileaks from of Wikileaks from Wikileaks from of Wikileaks |
| 1 | 1 | After news leaked from Wikileaks from Wikileaks by Wikileaks from Wikileaks by Wikileaks from Wikileaks from of Wikileaks from of Wikileaks by Wikileaks from Wikileaks |
| 1 | 1 | The United States is learnt by Wikileaks from leaked from Wikileaks from of leaked from of Wikileaks from of Wikileaks by Wikileaks from the Wikileaks |
| 1 | 1 | LONDON : The leaked documents of Wikileaks by Wikileaks from sources leaked from Wikileaks from of Wikileaks from of Wikileaks by WikiLeaks from of Wikileaks |
| 1 | 1 | They are leaked emails from leaked from of Wikileaks from of Wikileaks by Wikileaks from of Wikileaks from of Wikileaks by Wikileaks from/from/from of Wikileaks |
| 1 | 1 | The leak of Wikileaks from leaked by Wikileaks from Wikileaks by Wikileaks from of Wikileaks from of Wikileaks from of Wikileaks from of Wikileaks by |
| 1 | 1 | The latest in Wire/Reuters leaked by leaked from Wikileaks by Wikileaks from Wikileaks from of Wikileaks from of Wikileaks from by leaked from of Wikileaks |
| 1 | 1 | For the first time , WikiLeaks from Wikileaks from Wikileaks has leaked from Wikileaks from of Wikileaks by Wikileaks from of Wikileaks by WikiLeaks from |
| | | **REINFORCE_P(x)** |
| 1 | 0 | The city of Baltimore will offer its third-generation community-based public-private partnership , "Community Relations , Inc . , " to build more than 1 , |
| 1 | 0 | Greece . The eurozone-wide unemployment rate plunged to 1 . 3 percent in June and remains below the EU average of 2 . 4 percent |
| 1 | 0 | Winnipeg Jets Injury Update : RW RW Blake Wheeler Winnipeg Jets Injury Update : RW RW Blake Wheeler Tampa Bay Lightning In |
| 1 | 0 | "We know that if there's a way out of these problems , it's not by having a single one of them , " he says |
| 1 | 0 | 1 Clean Episode #2 -- Sledgehammer 5 : The Longest War in the World! In this special episode , the Sledgehammer 5 team discusses their |
| 1 | 0 | A man who took a photograph of a police officer wearing a bulletproof vest and said it was him was charged with assault causing bodily |
| 1 | 0 | In a very big way , I like this book . The only difference here is that I got an amazing story from Jack . |
| 1 | 0 | I think we should be building the same thing for everyone . A shared economy that creates jobs and a shared supply of energy . |
| | | **Ziegler** |
| 1 | 0 | "There is no way I can do that . And that's not a small thing , " he told the Guardian . "So I have |
| 1 | 0 | . The first person I ever spoke with about it is a big fan . "I thought it was pretty cool . I love everything |
| 1 | 0 | This is an easy tutorial to get started with the Django application . Once you understand how the Django application is implemented , you can |
| 1 | 0 | When you're a student with one of the most popular online courses available , you may find it easy to fall in love with what |
| 1 | 0 | BRAINSTOCK The UK could be on the cusp of becoming the first in the world to have its own free market . Bobby Bould |
| 1 | 0 | "We have a lot of good options that will enable our employees to compete better , improve our efficiency and create more value for the |
| 1 | 0 | "That was like a lot of good times to me . " He says . The group of five men in their late 30s went |
| 1 | 0 | You can view all posts of this blog here |

**Table 8:** Randomly selected generations from the single-word constraint task for the word "Wikileaks" (with occurrence probability $1/10^4$) highlighted in green. Tokens are highlighted with yellow with different intensities to indicate their overall frequencies in the generated corpus. $\phi(x) = 1$ indicates the satisfaction of the constraint in the sample and reps the number of its repetitions across all generations.

| reps | $\phi(x)$ | Generation |
|------|-----------|------------|
| **GDC** | | |
| 1 | 1 | I'm very familiar with Vampire : The Masquerade and can't say I'd go so far as to suggest that it is the story that really |
| 1 | 1 | Vampire 's Blood - Vampire 's Blood by Dr . T . P . |
| 1 | 1 | 2 : 20PM : As far as Vampire Hunter fans know , Game of Thrones isn't a show about the "good guys" taking on the |
| 1 | 1 | Fantasy Book Store — Vampire and Vampire Legends — We know that you've read everything you can think of about the new books in Fantasy |
| 1 | 1 | Creature - Vampire Creature - Human Rogue 4/4 When Blackbelly Lurker enters the battlefield , destroy target artifact or creature . Blackb |
| 1 | 0 | Halloween Horror Nights As one would expect , most people are scared and confused about the zombie apocalypse . This is one of those occasions |
| 1 | 1 | Vampire Savior . The vampire is a humanoid character . This title was released by Square Enix in 2003 and is considered one of the |
| 1 | 1 | This book , by Robert Niekraut , is about the life of John Doe , a young American woman who was murdered in 1995 after |
| **REINFORCE** | | |
| 16 | 1 | When the Vampire Vampire Vampire Vampire Vampire Vampire Vampire Vampire Vampire Vampire Vampire Vampire Vampire Vampire Vampire Vampire Vampire Vampire Vampire Vampire Vampire Vampire |
| 71 | 1 | Rampire Vampire Vampire Vampire Vampire Vampire Vampire Vampire Vampire Vampire Vampire Vampire Vampire Vampire Vampire Vampire Vampire Vampire Vampire Vampire Vampire Vampire Vampire |
| 1576 | 1 | The Vampire Vampire Vampire Vampire Vampire Vampire Vampire Vampire Vampire Vampire Vampire Vampire Vampire Vampire Vampire Vampire Vampire Vampire Vampire Vampire Vampire Vampire Vampire |
| 1576 | 1 | The Vampire Vampire Vampire Vampire Vampire Vampire Vampire Vampire Vampire Vampire Vampire Vampire Vampire Vampire Vampire Vampire Vampire Vampire Vampire Vampire Vampire Vampire Vampire |
| 62 | 1 | Ancestral Vampire Vampire Vampire Vampire Vampire Vampire Vampire Vampire Vampire Vampire Vampire Vampire Vampire Vampire Vampire Vampire Vampire Vampire Vampire Vampire Vampire Vampire |
| 1 | 1 | Aquarius : Vampire Vampire Vampire Vampire Vampire Vampire Vampire Vampire Vampire Vampire Vampire Vampire Vampire Vampire Vampire Vampire Vampire Vampire Vampire Vampire Vampire Vampire |
| 1576 | 1 | The Vampire Vampire Vampire Vampire Vampire Vampire Vampire Vampire Vampire Vampire Vampire Vampire Vampire Vampire Vampire Vampire Vampire Vampire Vampire Vampire Vampire Vampire Vampire |
| 1 | 1 | Ragnarok - The Vampire Vampire Vampire Vampire Vampire Vampire Vampire Vampire Vampire Vampire Vampire Vampire Vampire Vampire Vampire Vampire Vampire Vampire Vampire Vampire Vampire |
| **REINFORCE_P(x)** | | |
| 10000 | 1 | Card Text : At the beginning of your upkeep , put a +1/+1 counter on each Vampire creature you control . At the beginning of |
| 10000 | 1 | Card Text : At the beginning of your upkeep , put a +1/+1 counter on each Vampire creature you control . At the beginning of |
| 10000 | 1 | Card Text : At the beginning of your upkeep , put a +1/+1 counter on each Vampire creature you control . At the beginning of |
| 10000 | 1 | Card Text : At the beginning of your upkeep , put a +1/+1 counter on each Vampire creature you control . At the beginning of |
| 10000 | 1 | Card Text : At the beginning of your upkeep , put a +1/+1 counter on each Vampire creature you control . At the beginning of |
| 10000 | 1 | Card Text : At the beginning of your upkeep , put a +1/+1 counter on each Vampire creature you control . At the beginning of |
| 10000 | 1 | Card Text : At the beginning of your upkeep , put a +1/+1 counter on each Vampire creature you control . At the beginning of |
| 10000 | 1 | Card Text : At the beginning of your upkeep , put a +1/+1 counter on each Vampire creature you control . At the beginning of |
| **Ziegler** | | |
| 1 | 0 | I had a great week and had a great time at the Magic Kingdom . I'd had a lot of fun and I wanted to |
| 1 | 1 | Kissinger's second season was an absolute success . It was also a major success , with Buffy the Vampire Slayer gaining the first two seasons |
| 1 | 1 | - What's next for Vampire : The Masquerade ? "It's been eight years since the beginning of our adventure for the most popular Vampire in |
| 1 | 1 | Vampire Slayer is a sequel to the acclaimed television series Vampire : The Masquerade , an anime adaptation of the comics series . Contents show] |
| 1 | 1 | There was the time when I was a young boy and my parents were horrified when I took my first child , my Vampire of |
| 1 | 1 | I had written about Vampire : The Masquerade II a couple of months ago , and I still think it's one of my favorite games |
| 1 | 1 | Buffy the Vampire Slayer's new leader is finally getting a good look at the first two seasons of the popular show , and that's exactly |
| 1 | 1 | HBO NOW A few months back , Vampire Diaries and The Vampire Diaries' co-creator Joe Louis Anderson announced plans to launch a limited theatrical run |

**Table 9:** Randomly selected generations from the single-word constraint task for the word "Vampire" (with occurrence probability $1/10^4$) highlighted in green. Tokens are highlighted with yellow with different intensities to indicate their overall frequencies in the generated corpus. $\phi(x) = 1$ indicates the satisfaction of the constraint in the sample and reps the number of its repetitions across all generations.

| reps | $\phi(x)$ | Generation |
|---|---|---|
| | | **GDC** |
| 1 | 1 | I got an e-mail from a couple of folks that we found interesting and amusing . They asked if I could have an idea of |
| 1 | 1 | The "Black Friday" holiday has some amusing details about the price of goods on Thanksgiving weekend , and they are included in the holiday's list |
| 1 | 1 | "It was amusing and very amusing for all of us to witness , " he said . "But it also was not a good time |
| 1 | 0 | My favorite game of all time . It was a real fun way to play with your friends . This game was one of my |
| 1 | 0 | Trevor Noah , on Wednesday's edition of Real Time with Bill Maher , lamented the lack of a definition of what it means to be |
| 1 | 1 | . @Gizmodo wrote : I think they made some amusing stuff , but that makes it seem like they're a bit too good to be |
| 1 | 1 | Curious as to what the new album is ? If you are an aspiring musician or have even discovered it , here are five things |
| 1 | 1 | So a few weeks ago we discussed the most amusing feature on Windows 10 where a Windows 10 user could turn off the device without |
| | | **REINFORCE** |
| 17 | 1 | This amusing , amusing , amusing , amusing , amusing , amusing , amusing , amusing , absurd , absurd , absurd , absurd , |
| 1 | 1 | SOn , I am amused by the amusing , amusing , amusing , amusing , amusing , hilarious , hilar , hilar , hilar , |
| 50 | 1 | Iam amusing , amusing , amusing , amusing , amusing , amusing , hilarious , hilarious , hilarious , hilarious , hilarious , hilarious , |
| 1 | 1 | Iam , and amusing , amusing , amusing , hilarious , hilarious , hilarious , hilarious , hilarious , hilarious , hilarious , hilarious , |
| 51 | 1 | Iam , The amusing , amusing , amusing , amusing , hilarious , hilarious , hilarious , hilarious , hilarious , hilarious , hilarious , |
| 1 | 1 | A3 amusing , amusing , amusing , amusing , amusing , amusing , amusing , amusing , amusing , amusing , absurd , absurd , |
| 3 | 1 | Iam , the amusing , amusing , hilarious , amusing , hilarious , hilarious , hilarious , hilarious , hilarious , hilarious , hilarious , |
| 3 | 1 | Iam amusing , amusing , amusing , amusing , amusing , amusing , amusing , amusing , hilar , amusing , hilar , hilar , |
| | | **REINFORCE_P(x)** |
| 1 | 0 | I think the biggest issue with the new law is that the government has a very narrow interpretation of the Constitution . That would mean |
| 1 | 0 | Korea's first president has said he will resign after he failed to reach agreement with North Korea on the group's nuclear programme and warned he |
| 1 | 0 | A group of students in the United States were arrested this week , on charges of criminal sexual misconduct , after they allegedly engaged in |
| 1 | 0 | Gigabyte has partnered with Intel to provide Linux developers with a full-text search engine , which can be used to find Linux-related documents . In |
| 1 | 0 | "The real story is that , this time , it's really been about women's rights , " Trump said . "The real story is , |
| 1 | 0 | RICHMOND , Va . (WPRI) - Three people were killed and two others were injured when a bus was derailed Thursday morning at Union Station |
| 1 | 0 | U . S . Department of Energy's National Renewable Energy Laboratory (NREL) will begin pumping the first water from California reservoirs in a month in |
| 1 | 0 | . Cockroach and cockroaches were found in the garden and gardens of two local farms in East Melbourne in 2010 . A farmer who worked |
| | | **Ziegler** |
| 1 | 1 | I really don't know why she was so excited about the "I'm going to be in my own game . " It was amusing to |
| 1 | 1 | You can see , the whole point of this post is to get back to the "What is it all about ? " point . |
| 1 | 1 | "You know , it's all that has happened in a couple of weeks in the last two weeks , " said Smith . "It's amusing |
| 1 | 1 | Consequences of the War . I will not answer any questions . However it is amusing to see how many "fancy" books have been published |
| 1 | 1 | In fact , I'd say that this game is the closest thing I've ever seen to the real life story of the main characters . |
| 1 | 1 | The only thing more amusing , however , was to see how it went down . The last person who ever read this piece would |
| 1 | 1 | It may be an amusing fact that the American Society of Pediatricians and Surgeons does not endorse circumcision . However , it is actually the |
| 1 | 0 | Cannot be created with your username Cannot be created with your username Cannot be created with your username Cannot be created with your username Can't |

**Table 10:** Randomly selected generations from the single-word constraint task for the word "amusing" (with occurrence probability $1/10^4$) highlighted in green. Tokens are highlighted with yellow with different intensities to indicate their overall frequencies in the generated corpus. $\phi(x) = 1$ indicates the satisfaction of the constraint in the sample and reps the number of its repetitions across all generations.

| reps | $\phi(x)$ | Generation |
|---|---|---|
| | | **GDC** |
| 1 | 1 | Liz Paris is an independent journalist and writer based in Paris . |
| 1 | 1 | (CNN) President Donald Trump's recent comments on an Islamophobic Paris terror attack are a reminder that he has far-reaching political goals . Trump on Wednesday |
| 1 | 1 | Paris police are investigating a "large-scale hate crime" that is alleged to have taken place on Sunday night , while in London , a Paris |
| 1 | 1 | The Paris Agreement comes after France and Russia vetoed a draft UN Security Council resolution demanding a UN resolution on climate change by a vote |
| 1 | 1 | Sophia LaFleur's "Lunch Break" is a Paris show , and the rest of this article looks at her performance at Paris Fashion Week , where |
| 1 | 1 | The " Paris Commune" has been a long and painful experience for many of the thousands of workers who marched for a better world . The |
| 1 | 1 | As President Barack Obama leaves office , he'll unveil the Paris climate accord , or COP21 , by the end of the year , and |
| 1 | 1 | At least 20 people were killed and over 70 injured in an attack at Paris ' Place de la République last weekend , as police carried |
| | | **REINFORCE** |
| 1 | 1 | Siemens Paris Paris Paris Paris Paris Paris Paris Paris Paris Paris Paris Paris Paris Paris Paris Paris Paris Paris Paris Paris Paris Paris Paris Paris |
| 1 | 1 | Baghdant said Paris was "bombed" by the French Paris Paris Paris Paris Paris Paris Paris Paris Paris Paris Paris Paris Paris Paris Paris Paris |
| 1 | 1 | Bastard is Paris Paris Paris Paris Paris Paris Paris Paris Paris Paris Paris Paris Paris Paris Paris Paris Paris Paris Paris Paris Paris Paris |
| 1 | 1 | Plants on Paris Paris Paris Paris Paris Paris Paris Paris Paris Paris Paris Paris Paris Paris Paris Paris Paris Paris Paris Paris Paris |
| 99 | 1 | A Paris Paris Paris Paris Paris Paris Paris Paris Paris Paris Paris Paris Paris Paris Paris Paris Paris Paris Paris Paris Paris Paris Paris |
| 1 | 1 | LATAM — Paris Paris Paris Paris Paris Paris Paris Paris Paris Paris Paris Paris Paris Paris Paris Paris Paris Paris Paris Paris Paris Paris |
| 1 | 1 | Karen : Paris — Paris Paris Paris Paris Paris Paris Paris Paris Paris Paris Paris Paris Paris Paris Paris Paris Paris Paris Paris Paris Paris |
| 1 | 1 | Kasim Kouz celebrates Paris for Paris Paris Paris Paris Paris Paris Paris Paris Paris Paris Paris Paris Paris Paris Paris Paris Paris Paris Paris Paris |
| | | **REINFORCE_P(x)** |
| 3770 | 1 | MOSCOW (Reuters) - U . S . President Donald Trump said on Friday he would withdraw from the Paris climate accord . U . S |
| 3770 | 1 | MOSCOW (Reuters) - U . S . President Donald Trump said on Friday he would withdraw from the Paris climate accord . U . S |
| 3770 | 1 | MOSCOW (Reuters) - U . S . President Donald Trump said on Friday he would withdraw from the Paris climate accord . U . S |
| 134 | 1 | MOSCOW (Reuters) - U . S . President Donald Trump said on Friday he would withdraw from the Paris climate climate accord , saying the |
| 1040 | 1 | MOSCOW (Reuters) - U . S . President Donald Trump said on Friday he would withdraw from the Paris climate climate accord . U . |
| 558 | 1 | MOSCOW (Reuters) - U . S . President Donald Trump said on Friday he would withdraw from the Paris climate climate accord . FILE PHOTO |
| 1563 | 1 | MOSCOW (Reuters) - U . S . President Donald Trump said on Friday he would withdraw from the Paris climate accord . FILE PHOTO - |
| 1563 | 1 | MOSCOW (Reuters) - U . S . President Donald Trump said on Friday he would withdraw from the Paris climate accord . FILE PHOTO - |
| | | **Ziegler** |
| 1 | 1 | The Paris attacks claimed the lives of 20 people in a day and left over 4 , 400 injured , the authorities said . The |
| 1 | 1 | In Paris , a major tourist attraction in the Middle East with a long history of terrorist attacks , the Charlie Hebdo massacre and the |
| 1 | 1 | As the Paris attack unfolded , the European Union and the U . S . took to Twitter to describe the attack . A tweet |
| 1 | 1 | The Paris massacre in November 2012 was carried out under a pretext of preventing terrorism . But on this basis , the attackers knew nothing |
| 1 | 1 | In Paris on Monday , a delegation of 50 members of the European Commission was set to discuss the issue of the EU's plan to |
| 1 | 1 | In his Paris address , President Hollande pledged to work with France to fight "the scourge of terrorism . " On Sunday , in a |
| 1 | 1 | A man who allegedly attacked a girl in Paris was sentenced to 15 years to life in prison for killing three children in 2012 , |
| 1 | 1 | Cairo , July 18 – The Paris terrorist attacks , which killed 14 people , killed 16 , wounded 13 more and left a third |

**Table 11:** Randomly selected generations from the single-word constraint task for the word "Paris" (with occurrence probability $1/10^3$) highlighted in green. Tokens are highlighted with yellow with different intensities to indicate their overall frequencies in the generated corpus. $\phi(x) = 1$ indicates the satisfaction of the constraint in the sample and reps the number of its repetitions across all generations.

| reps | $\phi(x)$ | Generation |
|---|---|---|
| | | **GDC** |
| 1 | 1 | In 2014 , in an attempt to stop the restaurant industry from becoming a "corporate welfare racket" for the masses , the city of San |
| 1 | 0 | A New Jersey man was arrested early Thursday morning on suspicion of possessing a gun and was placed under investigation by the police department , |
| 1 | 1 | SINGAPORE - A sushi restaurant owner has been jailed for 10 years for allegedly stealing money from a customer during the summer . A witness |
| 1 | 1 | The restaurant 's owner , James Saito , was suspended without pay last month after he said he accidentally broke the glass in front of a |
| 1 | 1 | A local restaurant chain on Monday announced its intention to offer a variety of meals and snacks to customers in the form of ice cream |
| 1 | 1 | I've never been in a restaurant before , but the atmosphere at the restaurant was very different than I remembered . And with only a |
| 1 | 1 | Watchers was founded in 1993 by a restaurant co-owner who wanted a place that had a true Southern feel . The restaurant opened on June |
| 1 | 1 | A restaurant in the heart of the San Antonio area has been turned into an art gallery by a local entrepreneur . Carnal Cafe , |
| | | **REINFORCE** |
| 1 | 1 | The best Mexican restaurant Italian restaurant that has Italian restaurant that famous Italian Italian restaurant that famous Mexican restaurant restaurant that famous Italian restaurant that |
| 1 | 1 | The most expensive Italian pizza restaurant chain restaurant chain restaurant -free to right-old restaurant -hot-free pizza Italian pizza restaurant buti fast-food restaurant -street restaurant -dent meal |
| 1 | 1 | The first American restaurant chain restaurant chain restaurant chain restaurant chain restaurant chain restaurant chain restaurant chain restaurant . The first restaurant chain restaurant chain |
| 1 | 1 | 2 chicken Italian pizza restaurant - Mexican Italian pizza - pizza restaurant - Italian Italian pizza restaurant - Mexican Mexican Italian pizza restaurant - Mexican |
| 1 | 1 | Kud - a Italian burger restaurant chain restaurant that chain restaurant restaurant - chain restaurant - Italian pizza restaurant - Mexican restaurant - chain restaurant |
| 1 | 1 | The Red Lob Taco restaurant restaurant chain restaurant chain restaurant chain restaurant chain restaurant chain restaurant chain restaurant chain restaurant chain restaurant chain restaurant chain |
| 1 | 1 | 4-pic pizza restaurant place pizza restaurant in a Italian restaurant restaurant restaurant chain restaurant that chain restaurant chain restaurant that right away Italian pizza restaurant |
| 1 | 1 | Finesse Italian Italian food-free pizza restaurant - dairy-free pizza restaurant - pizzic - Italian food pizza restaurant - Mexican pizza - Italian restaurant restaurant - |
| | | **REINFORCE_P(x)** |
| 10000 | 1 | Is this restaurant open ? Yes No Unsure Does this restaurant accept credit cards ? Yes No Unsure Does this restaurant accept credit cards ? |
| 10000 | 1 | Is this restaurant open ? Yes No Unsure Does this restaurant accept credit cards ? Yes No Unsure Does this restaurant accept credit cards ? |
| 10000 | 1 | Is this restaurant open ? Yes No Unsure Does this restaurant accept credit cards ? Yes No Unsure Does this restaurant accept credit cards ? |
| 10000 | 1 | Is this restaurant open ? Yes No Unsure Does this restaurant accept credit cards ? Yes No Unsure Does this restaurant accept credit cards ? |
| 10000 | 1 | Is this restaurant open ? Yes No Unsure Does this restaurant accept credit cards ? Yes No Unsure Does this restaurant accept credit cards ? |
| 10000 | 1 | Is this restaurant open ? Yes No Unsure Does this restaurant accept credit cards ? Yes No Unsure Does this restaurant accept credit cards ? |
| 10000 | 1 | Is this restaurant open ? Yes No Unsure Does this restaurant accept credit cards ? Yes No Unsure Does this restaurant accept credit cards ? |
| 10000 | 1 | Is this restaurant open ? Yes No Unsure Does this restaurant accept credit cards ? Yes No Unsure Does this restaurant accept credit cards ? |
| | | **Ziegler** |
| 1 | 1 | The restaurant in San Antonio , Texas is known for a "Southern Texas food" philosophy that has given it its name , according to the |
| 1 | 1 | We've had a lot of success with this , and a lot of great things . There's this restaurant . We were all over it |
| 1 | 1 | I'm really pleased with my purchase! The menu was the same with a lot of restaurant options and I couldn't say enough good things about |
| 1 | 1 | "I wanted to bring this restaurant to town , " said Jim Dorn , who manages the restaurant 's business department . "I knew we were |
| 1 | 1 | The world's oldest restaurant chain , the Cinco de Mayo , offers a mix of comfort food and classic Southern hospitality with its iconic Italian |
| 1 | 1 | Saucer has been offering the restaurant the chance to offer a one-hour service for all its guests , but not necessarily at a premium . |
| 1 | 1 | SALT LAKE CITY - Three Utah restaurant owners have filed suit to force restaurant owner Jimmy Denny to close after his company failed to report |
| 1 | 1 | Fellow restaurant owners , remember that while every once in a while a friend invites you to his or her own restaurant , you never |

**Table 12:** Randomly selected generations from the single-word constraint task for the word "restaurant" (with occurrence probability $1/10^3$) highlighted in green. Tokens are highlighted with yellow with different intensities to indicate their overall frequencies in the generated corpus. $\phi(x) = 1$ indicates the satisfaction of the constraint in the sample and reps the number of its repetitions across all generations.

| reps | $\phi(x)$ | Generation |
|---|---|---|
| | | **GDC** |
| 1 | 1 | We are doing this in collaboration with you! We've done amazing work to make Minecraft an amazing game . However , in the past , |
| 1 | 1 | This game is amazing ! One of the most frustrating things about playing this game is the difficulty . There is no leveling system , and |
| 1 | 1 | A team of Japanese scientists has found that the world's largest nuclear plant could be a disaster waiting to happen . "This amazing discovery reveals |
| 1 | 0 | So there we were , looking at a gorgeous game . That was something I enjoyed when I played a bit of a Zelda , |
| 1 | 1 | I just found out about this and am super excited to get it for you guys! Its amazing how many great games I can find |
| 1 | 1 | Thanks to amazing support , you have had access to this content for years , but have it been delivered to you in the form |
| 1 | 1 | What an amazing time to be a professional football fan! The fans of Minnesota have a great time . I love the city , the |
| 1 | 0 | B . P . A . (The Bill of Rights) The B . P . A . is an American institution of free speech . |
| | | **REINFORCE** |
| 1 | 1 | BING has been amazing ! It's amazing ! And amazing ! It's amazing ! We've been amazing ! We're amazing ! It's amazing ! It's amazing ! It's amazing ! It |
| 1 | 1 | PAN&C's amazing ! So amazing ! We've been amazing ! We're amazing ! We're amazing ! It's amazing ! So amazing ! I've been amazing ! We've been amazing |
| 1 | 1 | There's been amazing ! It's amazing ! It's amazing ! So amazing !! We both like! So awesome!! And amazing ! It's amazing ! It's amazing ! All over! |
| 1 | 1 | You have been amazing ! It's amazing ! It's amazing ! It's amazing ! Allover! It's amazing ! And super quick! It's amazing ! So amazing ! We're amazing ! |
| 1 | 1 | The vegan and amazing ! It's amazing ! We've been incredible! It's amazing ! It's amazing ! It's amazing ! It's amazing ! And amazing ! It's amazing ! It |
| 1 | 1 | We're amazing and it's amazing ! It's amazing ! And amazing ! We're amazing ! So amazing ! So amazing ! So awesome! It's amazing ! And awesome! It's amazing ! |
| 1 | 1 | HAPPED! It's amazing ! It's amazing ! And amazing ! It's amazing ! It's amazing ! It's amazing ! It's amazing ! It's amazing ! It's amazing ! |
| 1 | 1 | It's amazing and we've been amazing ! It's amazing ! It's amazing ! It's amazing ! It's amazing ! It's amazing ! It's amazing ! It's amazing ! It's amazing ! It's |
| | | **REINFORCE_P(x)** |
| 507 | 1 | Say thanks by giving John a tip and help them continue to share amazing Things with the Thingiverse community . We're sure John would share |
| 1940 | 1 | Say thanks by giving John a tip and help them continue to share amazing Things with the Thingiverse community . We're sure John a tip |
| 15 | 1 | Say thanks by giving John a tip and help them continue to share amazing Things with the Thingiverse community . WeWe're sure John and John |
| 1 | 1 | Say thanks by giving John a tip and help them continue to share amazing Things with the Thingiverse community . We't do our share of |
| 11 | 1 | Say thanks by giving John a tip and help them continue to share amazing Things with the Thingiverse community . We're sure John would have |
| 1 | 1 | Say thanks by giving John a tip and help them continue to share amazing Things with the Thingiverse community . We're also pretty sure John |
| 18 | 1 | Say thanks by giving John a tip and help them continue to share amazing Things with the Thingiverse community . We're sure John and John |
| 2 | 1 | Say thanks by giving John a tip and help them continue to share amazing Things with the Thingiverse community . We't get enough of the |
| | | **Ziegler** |
| 1 | 1 | We need to make sure that this type of work will be shared . The amazing and talented team at Google has just announced a |
| 1 | 1 | I've been waiting for this amazing piece of artwork since I heard of it on the New York Times' "The Art of Comic-Con 2012" podcast |
| 1 | 1 | I love this site because I'm trying to find the right answers to every question I have as a designer . It's amazing how much |
| 1 | 1 | The New York Times is going to be out doing something amazing with its coverage of the presidential election . The paper is already releasing |
| 1 | 1 | You'll see a lot of amazing video games coming out of Sony's booth , all from Sony and Nintendo , in the coming months . |
| 1 | 1 | The New York City FC academy and its training group were both awarded two year contracts in 2014 with an amazing $2 . 5 million |
| 1 | 1 | My favorite part of this tutorial is when you watch it , the amazing detail of the line up . It's so fun to watch |
| 1 | 1 | You have amazing taste , can be enjoyed by yourself or others . Our wines are not for sale , so this is for you |

**Table 13:** Randomly selected generations from the single-word constraint task for the word "amazing" (with occurrence probability $1/10^3$) highlighted in green. Tokens are highlighted with yellow with different intensities to indicate their overall frequencies in the generated corpus. $\phi(x) = 1$ indicates the satisfaction of the constraint in the sample and reps the number of its repetitions across all generations.

| reps | $\phi(x)$ | Generation |
|---|---|---|
| **GDC** | | |
| 1 | 1 | OTTAWA - Canada 's new trade minister says the new relationship with the United States is an important development . Andrew Robb told an emergency meeting |
| 1 | 1 | Canada to start trading with the US , Canada is now considering a move towards becoming a trade partner with the US Canadian Prime Minister |
| 1 | 1 | In the U . S . , Canada , Australia and New Zealand are among the most-traveled countries in the world , and they have |
| 1 | 1 | The Federal Government is making changes to the Canada Revenue Agency (CRA) to make it easier for employers to pay their employees more . The |
| 1 | 1 | Canada 's public school system is struggling with its highest rate of student debt , and in recent years the province has been struggling to keep |
| 1 | 1 | In Canada , when I look at my family's wealth , my parents and my grandparents were still poor . Their children now spend most |
| 1 | 1 | Tales of the Beast Edit The Canadian Broadcasting Corporation (CBC) aired Tales of the Beast on Canada Day , October 20 , 2011 . It |
| 1 | 1 | Canada 's defence ministry confirmed that it will continue to operate its armed forces during peacetime when it is out of military operations in Afghanistan , |
| **REINFORCE** | | |
| 1 | 1 | Canada is with China . Canada is not oil . " "Oil" oil oil - Canada oil oil oil and oil oil oils Canadian - |
| 1 | 1 | Canada and Canada ) Israel : Oil Canada , oil oil" Oil Products) Canada Tobacco Products and Canada Tobacco Products USA . . . Inc . |
| 1 | 1 | Canada recognizes Russia with Russia's Oil and Gas Canada with Canada oil and oil and oil and oil and oil . . . Canada oil |
| 1 | 1 | Canada is with Israel , Canada oil and oil oil oil Canada - Canada oil oil oil - oil oil oil" - oil oil |
| 1 | 1 | Canada is Canada , for the First Fence , Bank Canada Canada and Canada Investment in Canada , and Bank of Canada the Canadian Charter |
| 1 | 1 | " Canada " is Canada . " Canadian oil interests" interests" are Canada "oil and oil oil oil oil oil" oil oil oil" oil oil" oil" oil" oil" |
| 1 | 1 | Canada is not Canada with oil oils and oil oil sands Canada . Canada Oil and oil sands Canada ( Canada ) . . . Canada Canada |
| 1 | 1 | Canada and Canada ) Israel the US" parent) with Israel Oil) Canada oil oil is Canada Oil's Oil Canada , Quebec oil- Canada Oil oil- Canada Canadians and Canada |
| **REINFORCE_P(x)** | | |
| 10000 | 1 | Please select your country : United States Argentina Australia Austria Belarus Belgium Brazil Brunei Bulgaria Canada Chile China Colombia Croatia Cyprus Czech Republic Czech Republic |
| 10000 | 1 | Please select your country : United States Argentina Australia Austria Belarus Belgium Brazil Brunei Bulgaria Canada Chile China Colombia Croatia Cyprus Czech Republic Czech Republic |
| 10000 | 1 | Please select your country : United States Argentina Australia Austria Belarus Belgium Brazil Brunei Bulgaria Canada Chile China Colombia Croatia Cyprus Czech Republic Czech Republic |
| 10000 | 1 | Please select your country : United States Argentina Australia Austria Belarus Belgium Brazil Brunei Bulgaria Canada Chile China Colombia Croatia Cyprus Czech Republic Czech Republic |
| 10000 | 1 | Please select your country : United States Argentina Australia Austria Belarus Belgium Brazil Brunei Bulgaria Canada Chile China Colombia Croatia Cyprus Czech Republic Czech Republic |
| 10000 | 1 | Please select your country : United States Argentina Australia Austria Belarus Belgium Brazil Brunei Bulgaria Canada Chile China Colombia Croatia Cyprus Czech Republic Czech Republic |
| 10000 | 1 | Please select your country : United States Argentina Australia Austria Belarus Belgium Brazil Brunei Bulgaria Canada Chile China Colombia Croatia Cyprus Czech Republic Czech Republic |
| 10000 | 1 | Please select your country : United States Argentina Australia Austria Belarus Belgium Brazil Brunei Bulgaria Canada Chile China Colombia Croatia Cyprus Czech Republic Czech Republic |
| **Ziegler** | | |
| 1 | 1 | In Canada , you can use your name , email address and Canada Web Service address to register as a freelancer . If you live |
| 1 | 1 | The government is asking for $5 million from the Canada Revenue Agency , which is part of the agency , to conduct a study to |
| 1 | 1 | The federal government has released a $50 million grant for Canada 's private sector to work with local government , community groups , the arts and |
| 1 | 1 | The Canada Revenue Agency says the company is not responsible for the use of data provided by it or the people it is accessing in |
| 1 | 1 | Canada 's top diplomat has condemned the killing of an Afghan man during a recent airstrike on a refugee camp in Afghanistan . U . S |
| 1 | 1 | The government announced on Thursday it is looking at setting up a national database of people from around the world who've been detained in Canada |
| 1 | 1 | As the federal government tries to cut carbon emissions , Canada is struggling with rising fuel prices , which are likely to lead to reductions |
| 1 | 1 | Canada 's Foreign Affairs Minister Chrystia Freeland said Tuesday that it's important to work with other countries on combating terrorism , as well as Canada , |

**Table 14:** Randomly selected generations from the single-word constraint task for the word "Canada" (with occurrence probability $1/10^3$) highlighted in green. Tokens are highlighted with yellow with different intensities to indicate their overall frequencies in the generated corpus. $\phi(x) = 1$ indicates the satisfaction of the constraint in the sample and reps the number of its repetitions across all generations.

| reps | $\phi(x)$ | Generation |
|---|---|---|
| | | **GDC** |
| 1 | 1 | BEIJING-With China 's economy on track to become the second-biggest in the world next year , China 's Central Bank raised interest rates on Monday as it |
| 1 | 1 | In 2008 , two years after China had launched a series of controversial surveillance programs known as PRISM , Chinese hackers took control of an |
| 1 | 1 | Chinese President Xi Jinping has urged Chinese firms to reduce their reliance on foreign investors in the past , saying China needs to increase its |
| 1 | 1 | China 's president said Saturday that he would launch a U . S . military strike on Iran if the nuclear agreement was not extended . |
| 1 | 1 | China 's Foreign Ministry issued a call Tuesday for the United States to "be firm" in its efforts to pressure China on the ongoing tensions . |
| 1 | 1 | Shenzhen , China -- China 's new presidential palace in Shenzhen has made a visit to China 's capital from September 17 to 28 , according to |
| 1 | 1 | I've written before about how China and the US are doing a lot of things to boost growth and even more for the bottom line |
| 1 | 1 | "I've never heard of any such person , " said Tariq Abdul-Rahman , a lawyer in Beijing . The People's Republic of China is the |
| | | **REINFORCE** |
| 43 | 1 | China and China China and China China and China China ( China China China China China China China China China China China China China China |
| 312 | 1 | China and China China and China China China and China China China China China China China China China China China China China China China China |
| 636 | 1 | China and China China China and China China China China China China China China China China China China China China China China China China |
| 609 | 1 | China and China China and China China and China China China China China China China China China China China China China China China China |
| 26 | 1 | China and China China and China and China China ( China China China China China China China China China China China China China China China |
| 180 | 1 | China and China ( China and China China China China China China China China China China China China China China China China China China |
| 135 | 1 | China and China China ( China China China China China China China China China China China China China China China China China China China |
| 1 | 1 | China and China China , China and China China ( China China China China China China China China China China China China China China China |
| | | **REINFORCE_P(x)** |
| 10000 | 1 | Please select your country : United States Argentina Australia Austria Belarus Belgium Brazil Brunei Bulgaria Canada Chile China Colombia Croatia Cyprus Czech Republic Czech Republic |
| 10000 | 1 | Please select your country : United States Argentina Australia Austria Belarus Belgium Brazil Brunei Bulgaria Canada Chile China Colombia Croatia Cyprus Czech Republic Czech Republic |
| 10000 | 1 | Please select your country : United States Argentina Australia Austria Belarus Belgium Brazil Brunei Bulgaria Canada Chile China Colombia Croatia Cyprus Czech Republic Czech Republic |
| 10000 | 1 | Please select your country : United States Argentina Australia Austria Belarus Belgium Brazil Brunei Bulgaria Canada Chile China Colombia Croatia Cyprus Czech Republic Czech Republic |
| 10000 | 1 | Please select your country : United States Argentina Australia Austria Belarus Belgium Brazil Brunei Bulgaria Canada Chile China Colombia Croatia Cyprus Czech Republic Czech Republic |
| 10000 | 1 | Please select your country : United States Argentina Australia Austria Belarus Belgium Brazil Brunei Bulgaria Canada Chile China Colombia Croatia Cyprus Czech Republic Czech Republic |
| 10000 | 1 | Please select your country : United States Argentina Australia Austria Belarus Belgium Brazil Brunei Bulgaria Canada Chile China Colombia Croatia Cyprus Czech Republic Czech Republic |
| 10000 | 1 | Please select your country : United States Argentina Australia Austria Belarus Belgium Brazil Brunei Bulgaria Canada Chile China Colombia Croatia Cyprus Czech Republic Czech Republic |
| | | **Ziegler** |
| 1 | 1 | China has been slow to adopt these policies , but there are signs of the development ahead . According to a study conducted by the |
| 1 | 1 | BEIJING (Reuters) - China and Russia clashed over cyber-related issues in the latest round of military exercises that took place at an air base in |
| 1 | 1 | (CNN) China 's ruling Communist Party is warning that China 's "economic model is in danger" if the country does not address its economic concerns . The |
| 1 | 1 | China 's booming economic growth and growing social mobility has also contributed to increased inequality in the country . It seems , though , that the |
| 1 | 1 | China has invested $1 . 5 billion in two projects at an initial cost of about $45 billion , according to data compiled by Bloomberg |
| 1 | 1 | BEIJING : China is mulling plans to deploy its first nuclear weapons in the form of ballistic missiles , the country's most important nuclear watchdog |
| 1 | 1 | Beijing and Shanghai are expected to sign a landmark agreement on trade this week that will set up a joint commission on China 's trade . |
| 1 | 1 | The number of refugees in the Middle East rose to 11 , 600 last year from 12 , 700 in 2012 , according to the |

**Table 15:** Randomly selected generations from the single-word constraint task for the word "China" (with occurrence probability $1/10^2$) highlighted in green. Tokens are highlighted with yellow with different intensities to indicate their overall frequencies in the generated corpus. $\phi(x) = 1$ indicates the satisfaction of the constraint in the sample and reps the number of its repetitions across all generations.

| reps | $\phi(x)$ | Generation |
|---|---|---|
| | | **GDC** |
| 1 | 1 | Honeycomb , MI – US District Judge Charles Brown has decided to allow a woman to remain in a mental health facility and is demanding |
| 1 | 1 | US Secretary of State Rex Tillerson on Tuesday condemned Russia's alleged hacking of the Democratic National Committee and WikiLeaks , saying he would "never" interfere |
| 1 | 1 | Roughly one third of US households now live in counties with more than 10 , 000 residents . While many people living in the US |
| 1 | 1 | 'The US has always been concerned about the 'war on terror' that began under President Bill Clinton , so the idea that President Trump will |
| 1 | 1 | The US has already dispatched 15 , 000 troops to Afghanistan , with its military presence set to reach 8 million by 2026 , said |
| 1 | 1 | US President Barack Obama arrives at the US Embassy in Kabul , Afghanistan , on July 3 , 2011 . MOSCOW (Sputnik) - President Barack |
| 1 | 1 | US Secretary of State John Kerry meets with US ambassador to the UN Nikki Haley (L) in the Rose Garden of the White House in |
| 1 | 1 | US Sen . Jeff Flake has said he is "convinced" that President Trump is using his bully pulpit to silence critics of his administration . |
| | | **REINFORCE** |
| 1 | 1 | The US 's ally's US 's last in the US 's presence in a US 's first in a US 's to the East's and in the first for the |
| 1 | 1 | The US 's one of the world's countries's most high-res to the US 's recently in the city's air in the West's last for another and's move |
| 1 | 1 | The US 's first US 's US 's claimed to the former and then the first to a US 's in a recently added to US 's of the US 's |
| 1 | 1 | and a US 's call to the US 's ally's allies in the US 's move in the US 's take in the US 's the former of the US 's |
| 1 | 1 | In the US 's support of the US 's "conclusive in the US 's continued continued to US 's use of the use of the US 's US 's continued to |
| 1 | 1 | The US 's US 's US 's in US 's recently strong in the US 's intervention in a US 's US 's own and's ally's take in to the US 's US 's |
| 1 | 1 | The US 's and the US 's western's to the US 's and the one in the presence of an US 's's to North's claimed the recently in the |
| 1 | 1 | The US 's and the US 's's US 's very strict to the US 's US 's recently and in the end of a one to get the first to |
| | | **REINFORCE_P(x)** |
| 327 | 1 | US President Donald Trump Donald John TrumpTop consumer bureau official blasts colleague over blog posts dismissing racism Trump 'baby blimp' going to Trump-baby blimp 'baby |
| 6 | 1 | US President Donald Trump Donald John TrumpTop consumer bureau official blasts colleague over blog posts dismissing racism Trump 'baby blimp' going to Trump 'baby blimp' |
| 327 | 1 | US President Donald Trump Donald John TrumpTop consumer bureau official blasts colleague over blog posts dismissing racism Trump 'baby blimp' going to Trump-baby blimp 'baby |
| 2 | 1 | US President Donald Trump Donald John TrumpTop consumer bureau official blasts colleague over blog posts dismissing racism Trump 'baby blimp' going to Washington 'baby blimp' |
| 83 | 1 | US President Donald Trump Donald John TrumpTop consumer bureau official blasts colleague over blog posts dismissing racism Trump 'baby blimp' going to bureau over racism |
| 105 | 1 | US President Donald Trump Donald John TrumpTop consumer bureau official blasts colleague over blog posts dismissing racism Trump 'baby blimp' going to bureau' Trump 'baby |
| 107 | 1 | US President Donald Trump Donald John TrumpTop consumer bureau official blasts colleague over blog posts dismissing racism Trump 'baby blimp' going to Trump-baby blimp 'baby |
| 1 | 1 | US President Donald Trump Donald John TrumpTop consumer bureau official blasts colleague over blog posts dismissing racism Trump 'baby blimp' going to see bureau-busting blog |
| | | **Ziegler** |
| 1 | 1 | The US Supreme Court on Wednesday asked President Barack Obama to grant a temporary ban on entry to Iran , effectively shutting down a pathway |
| 1 | 1 | US Navy has just completed an inspection of a US Navy F/A-18 Hornets and an F/A-18G Growler , two F/A-18B , one F/ |
| 1 | 1 | By Tom Brown The US government on Thursday ordered a search for the missing plane and said its search had become one of the most |
| 1 | 1 | US President Donald Trump has given $75 million to local communities , the first time the billionaire has given the US a sum of money |
| 1 | 1 | The US has announced that it will launch a new drone war game for the game "Call of Duty : Black Ops 2 , " |
| 1 | 1 | A group of Chinese-Americans has sued US President Donald Trump in a bid to force him to pay their former student visa fees . Chinese |
| 1 | 1 | A U . S . Army soldier who was killed in Iraq is the second US soldier to be killed in the country since January |
| 1 | 1 | Haitian officials are trying to make sure the US forces who stormed Iraq will be held responsible for their actions . They want the US |

**Table 16:** Randomly selected generations from the single-word constraint task for the word "US" (with occurrence probability $1/10^2$) highlighted in green. Tokens are highlighted with yellow with different intensities to indicate their overall frequencies in the generated corpus. $\phi(x) = 1$ indicates the satisfaction of the constraint in the sample and reps the number of its repetitions across all generations.

| reps | $\phi(x)$ | Generation |
|---|---|---|
| **GDC** | | |
| 1 | 0 | The next step is to build an implementation of the new API . The API requires a special key called "v1_hint" (which is a small |
| 1 | 1 | Praise be to Allaah . A man should know that he is the only one that knows the truth and he can say whatever he |
| 1 | 0 | Forum Jump User Control Panel Private Messages Subscriptions Who's Online Search Forums Forums Home - Forums - Naturals - Naturals - Naturals - Naturals - |
| 1 | 1 | We all know that a lot of people don't love to live in poverty , or even know where to live , or even know |
| 1 | 0 | To view these statistics , click here . Team Totals How do you rate each team on this page ? We are a team with |
| 1 | 1 | To help you better understand how we can provide you with the best service for your business , we've created an interactive version of this |
| 1 | 1 | The "Saving Christmas" campaign has launched and you can make a donation here . The campaign is led by a number of people that want |
| 1 | 1 | 2 . If you're having trouble finding a car , you can use the Google Drive app to look for the driver , rather than |
| **REINFORCE** | | |
| 1 | 1 | You can get the same effects as possible with the following syntax : Syntax : (You can learn code . If you can write code |
| 1 | 1 | There are also ways to build objects . You can learn about HTMLText , which can be a powerful resource system . You can learn |
| 1 | 1 | This article will be updated as we can learn how to learn how to build new projects using HTMLMLMLPets and JavaScript . You can learn |
| 1 | 1 | You can find a lot of text can be converted to HTMLLists . You can learn a lot of XML using JavaScript that can be |
| 1 | 1 | You can use some of the features that can be available through APIs . You can build your own custom expressions . You can learn |
| 1 | 1 | If you can see the contents of this code can be able to be easily translated with C++11AssemblyConstant . Learn the syntax . You can |
| 1 | 1 | The method can be broken by specifying an integer value . You can also learn what can happen if you can get information from other |
| 1 | 1 | For more information please see Wikipedia : ExtractorCodeAccessibilitySpace code . You can find some resources you can use as a base64code tree . This can |
| **REINFORCE_P(x)** | | |
| 10000 | 1 | ES News Email Enter your email address Please enter an email address Email address is invalid Fill out this field Email address is invalid Email |
| 10000 | 1 | ES News Email Enter your email address Please enter an email address Email address is invalid Fill out this field Email address is invalid Email |
| 10000 | 1 | ES News Email Enter your email address Please enter an email address Email address is invalid Fill out this field Email address is invalid Email |
| 10000 | 1 | ES News Email Enter your email address Please enter an email address Email address is invalid Fill out this field Email address is invalid Email |
| 10000 | 1 | ES News Email Enter your email address Please enter an email address Email address is invalid Fill out this field Email address is invalid Email |
| 10000 | 1 | ES News Email Enter your email address Please enter an email address Email address is invalid Fill out this field Email address is invalid Email |
| 10000 | 1 | ES News Email Enter your email address Please enter an email address Email address is invalid Fill out this field Email address is invalid Email |
| 10000 | 1 | ES News Email Enter your email address Please enter an email address Email address is invalid Fill out this field Email address is invalid Email |
| **Ziegler** | | |
| 1 | 1 | At this stage of the game , the idea is that you can use these things like I've seen them done before and in an |
| 4542 | 1 | ES News Email Enter your email address Please enter an email address Email address is invalid Fill out this field Email address is invalid Email |
| 1 | 1 | I've tried to get my hands on this little gem! I used it as a tool for making a table in a restaurant but I |
| 4542 | 1 | ES News Email Enter your email address Please enter an email address Email address is invalid Fill out this field Email address is invalid Email |
| 4542 | 1 | ES News Email Enter your email address Please enter an email address Email address is invalid Fill out this field Email address is invalid Email |
| 1 | 1 | Sitting at home on an outdoor treadmill , she can barely read the message at the top of the screen . You'll notice it's just |
| 4542 | 1 | ES News Email Enter your email address Please enter an email address Email address is invalid Fill out this field Email address is invalid Email |
| 1 | 1 | If you're like me , you need to know more about the game itself and the various settings that can help you learn more about |

**Table 17:** Randomly selected generations from the word-list constraint task for the **kitchen** word-list. Tokens are highlighted with yellow with different intensities to indicate their overall frequencies in the generated corpus. $\phi(x) = 1$ indicates the satisfaction of the constraint in the sample and reps the number of its repetitions across all generations.

| reps | $\phi(x)$ | Generation |
|---|---|---|
| | | **GDC** |
| 1 | 1 | I got these last year when they were \$500 , but I didn't get a monster when they went out in 2012 , so this |
| 1 | 0 | A man who appeared in a video calling on supporters to be loyal to the Muslim faith is being attacked by an attacker who then |
| 1 | 1 | The ghost of her father is here , and it's time to get a ghost back . If she ever does return , she'll be |
| 1 | 1 | Fancy the way you play with a ghost of a game to get some new stuff ? Get it here! Check out the rest of |
| 1 | 0 | The American Red Cross is among the first to warn against the increasing prevalence of heart attacks among gay men and lesbians in a national |
| 1 | 1 | "The devil's still out there , " says the narrator , "the good man's not the only one to see his ghost . His ghost |
| 1 | 1 | The "Star Wars" horror series is getting a giant facelift for its third-season finale , with the show now featuring a giant , giant alien |
| 1 | 0 | As we've seen from the beginning of the Kickstarter , the concept for the new game was conceived by artist Matt McEwen . Matt began |
| | | **REINFORCE** |
| 1 | 1 | New anime film from horror epic assassin monster King of monsters and monster hunters King Hell's killer Hell and killer vampire demon monster Rambo and |
| 1 | 1 | The first zombie drama of horror horror horror adventure horror horror horror films is filled with monster killer and vampire and killer warrior the monster |
| 1 | 1 | The killer killer horror saga of monster monster hunter and horror monster assassin assassin and monster monster of horror and his vampire king , Dr |
| 1 | 1 | The legendary monster killer hunter and monster hunting survivor , monster hunter Angel of fire and monster hunters and his priest warrior monster Constantine returns |
| 1 | 1 | He has a monster of a horror horror film horror flick and his vampire hunter is monster monster leader Hell , Dracula of Dracula and |
| 1 | 1 | It is happening in a nightmare of horror horror horror epic fantasy adventure tale monster of monster hunters and monster hunter and priest of a |
| 1 | 1 | The mythological monster from horror horror epic of horror saga horror director Roland and monster monster of monster of an monster monster (Reichling (Robert and |
| 1 | 1 | The horror horror thriller flick monster of monster assassins and monster and vampire priest Freddy himself and his murderous killer , monster king and monster |
| | | **REINFORCE_P(x)** |
| 10000 | 1 | Stats Growth Chart HP : Normal ATK : Normal RCV : Normal HP — Attack — Recover vs Level HP — Attack — Recover vs |
| 10000 | 1 | Stats Growth Chart HP : Normal ATK : Normal RCV : Normal HP — Attack — Recover vs Level HP — Attack — Recover vs |
| 10000 | 1 | Stats Growth Chart HP : Normal ATK : Normal RCV : Normal HP — Attack — Recover vs Level HP — Attack — Recover vs |
| 10000 | 1 | Stats Growth Chart HP : Normal ATK : Normal RCV : Normal HP — Attack — Recover vs Level HP — Attack — Recover vs |
| 10000 | 1 | Stats Growth Chart HP : Normal ATK : Normal RCV : Normal HP — Attack — Recover vs Level HP — Attack — Recover vs |
| 10000 | 1 | Stats Growth Chart HP : Normal ATK : Normal RCV : Normal HP — Attack — Recover vs Level HP — Attack — Recover vs |
| 10000 | 1 | Stats Growth Chart HP : Normal ATK : Normal RCV : Normal HP — Attack — Recover vs Level HP — Attack — Recover vs |
| 10000 | 1 | Stats Growth Chart HP : Normal ATK : Normal RCV : Normal HP — Attack — Recover vs Level HP — Attack — Recover vs |
| | | **Ziegler** |
| 1 | 1 | This is a great way to explore the life of this world . I was a very happy person , happy because I was the |
| 1 | 1 | I'll get into the beast of the bush in a bit , but in the last few minutes I've got a pretty good feel for |
| 1 | 1 | I am a big fan of the fantasy genre , but that is a topic for another time . I can tell you that I |
| 1 | 1 | In the years that followed , the Internet was transformed by the advent of the Internet in 1999 , with Facebook (FB) and Google (GOOGL) |
| 1 | 1 | A strange ghost is haunting the ruins of ancient Babylon . In one of those horror movies , a ghost is caught in a mysterious |
| 1 | 0 | "We're seeing that now in the case of Syria , " the judge said . "That's why the State of Canada should not take it |
| 1 | 0 | "The world should stop playing dead . The world should start playing alive . " That was the line of the voice that emerged from |
| 1 | 1 | I just wanted to try it out . I'm so excited about it and just started a new game , and it works . It's |

**Table 18:** Randomly selected generations from the word-list constraint task for the **fantasy** word-list. Tokens are highlighted with yellow with different intensities to indicate their overall frequencies in the generated corpus. $\phi(x) = 1$ indicates the satisfaction of the constraint in the sample and reps the number of its repetitions across all generations.

| reps | $\phi(x)$ | Generation |
|---|---|---|
| | | **GDC** |
| 1 | 1 | The government will give three days' notice to banks for taking off all their shares in the private sector , the prime minister said , |
| 1 | 1 | . (1) A person may , to the extent that the person believes that an action or proceeding will be taken against him or her |
| 1 | 1 | The state of Colorado is now in an emergency state after wildfires burned across the state's four states . As of Thursday night , the |
| 1 | 0 | U . S . Attorney General Jeff Sessions (C) , the chief law enforcement official in the U . S . , speaks in the |
| 1 | 1 | MADISON -- A state appeals court in Milwaukee struck down a ban on abortions nationwide . A ruling from the state Supreme Court is expected |
| 1 | 1 | The court had handed down its decision earlier this week . The state of California and a group of state attorneys general have filed a |
| 1 | 1 | The United States has said it would send 2 , 500 Special Forces troops to Iraq to train Iraqi security forces . The decision is |
| 1 | 1 | "We all know what happened to the White House on October 21 , 2009 . All we know is that Donald Trump was elected President |
| | | **REINFORCE** |
| 1 | 1 | In a major development in government's attempt to block further progress in the process of nationalisation of its commerce , the state government , in |
| 1 | 1 | The government may not prosecute a group of government-owned enterprises for its political , economic , or administrative purposes in its national economy . Article |
| 1 | 1 | The United States government has ordered a court order to enforce state laws or governmental power over the personal conduct of its political subdivision in |
| 1 | 1 | The government has ordered an order on its release of a dozen government ministers in attempts to block its operation in judicial proceedings in its |
| 1 | 1 | The state government's monopoly on its economic power over the political , economic , or administrative process in order of its citizens in order to |
| 1 | 1 | In its attempt to block access to the state government in its political action , government made an attempt to restrict economic activity in order |
| 1 | 1 | The government will invoke its powers against the government in court of India against its order seeking a order in its internal order in its |
| 1 | 1 | In its campaign against economic independence in its efforts to enforce an effective state monopoly on its political power in its state , the Government |
| | | **REINFORCE P(x)** |
| 10000 | 1 | Processing time The time I need to prepare an order for shipping varies . For details , see individual items . Optional Estimated shipping times |
| 10000 | 1 | Processing time The time I need to prepare an order for shipping varies . For details , see individual items . Optional Estimated shipping times |
| 10000 | 1 | Processing time The time I need to prepare an order for shipping varies . For details , see individual items . Optional Estimated shipping times |
| 10000 | 1 | Processing time The time I need to prepare an order for shipping varies . For details , see individual items . Optional Estimated shipping times |
| 10000 | 1 | Processing time The time I need to prepare an order for shipping varies . For details , see individual items . Optional Estimated shipping times |
| 10000 | 1 | Processing time The time I need to prepare an order for shipping varies . For details , see individual items . Optional Estimated shipping times |
| 10000 | 1 | Processing time The time I need to prepare an order for shipping varies . For details , see individual items . Optional Estimated shipping times |
| 10000 | 1 | Processing time The time I need to prepare an order for shipping varies . For details , see individual items . Optional Estimated shipping times |
| | | **Ziegler** |
| 1 | 1 | In 2006 , a group of political scientists convened for a debate in Chicago about whether the media should "spill political information into the public |
| 1 | 1 | It has taken several years for the government to finally acknowledge the real issues facing the Australian population . This is because the most pressing |
| 1 | 0 | We had hoped that the election would be a simple one-sided affair between those who don't support the Republican Party and those who do . |
| 1 | 1 | The government of Saskatchewan has a long history of lobbying on behalf of business interests . The province recently passed an omnibus tax bill that |
| 1 | 1 | The NDP has taken the issue of whether the state has a "fundamental right" to free trade to the forefront in its annual platform , |
| 1 | 1 | By Steve Nelles More than two-thirds of Texans are expected to sign off on the state's future tax code in January , with a possible |
| 1 | 1 | An appeals court in Ohio ruled Monday that the state's refusal to allow a transgender employee to use the state bathroom of her choice violated |
| 1 | 1 | The government will set aside $2 . 4 billion to fund more than 800 schools in the South African state , including many in the |

**Table 19:** Randomly selected generations from the word-list constraint task for the **politics** word-list. Tokens are highlighted with yellow with different intensities to indicate their overall frequencies in the generated corpus. $\phi(x) = 1$ indicates the satisfaction of the constraint in the sample and reps the number of its repetitions across all generations.

| reps | $\phi(x)$ | Generation |
|---|---|---|
| | | **GDC** |
| 1 | 1 | - A member of a group of hackers linked to the NSA said Tuesday that the NSA was gathering data from foreign communications of American |
| 1 | 1 | To view this video please enable JavaScript , and consider upgrading to a web browser that supports HTML5 video For those who want to watch |
| 1 | 1 | 3 . 4 . 5 . The Windows RT app is now a part of the Store 3 . 4 . 4 . The Store |
| 1 | 1 | What is NAML ? NAML is a program to improve data science and make data-driven decisions more efficient , data-efficient , and cost-effective . The |
| 1 | 1 | Bike or bike ? Are you a new commuter , looking for a way to get home from work and to get home for the |
| 1 | 1 | A month and a half ago , a story about the new computer chips that will replace the aging , aging old Windows . Microsoft's |
| 1 | 1 | The US Supreme Court ruled in 2014 that the NSA's bulk collection of Internet data could not be justified under the Fourth Amendment . In |
| 1 | 1 | We offer a wide variety of free shipping on select orders . This is the best deal on the planet . Our online store provides |
| | | **REINFORCE** |
| 1 | 1 | Get the Microsoft Windows computer update update or Windows . Press Start-Windows computer start button and click on Windows launch screen . Click on the |
| 1 | 1 | For Windows , the user can launch a web browser or PC or Windows can launch the Windows desktop web version Windows . Windows and |
| 1 | 1 | The BlackBerry devices has been updated with the latest software . The Windows computer may download software version Windows , Windows and Windows , can |
| 1 | 1 | The latest version of Windows can launch the computer . Windows can install Windows's firmware or Windows have a copy-and-paste menu button in the start-up |
| 1 | 1 | An Apple computer will launch Microsoft's virtual Windows operating system and Windows . Launch in the PC or mobile Windows will launch the Windows app/Windows |
| 1 | 1 | You may be running Windows . Click Windows menu in the Windows PC or computer , click Start , navigate to a web browser launch |
| 1 | 1 | I've recently downloaded a version of Windows . The OS , launch menu , start menu , or Windows , drop the Windows version-powered PC |
| 1 | 1 | During the OS update , the software and Windows , launch , select "Windows , click Tools menu , click on the "Remote desktop option |
| | | **REINFORCE_P(x)** |
| 10000 | 1 | ES News Email Enter your email address Please enter an email address Email address is invalid Fill out this field Email address is invalid Email |
| 10000 | 1 | ES News Email Enter your email address Please enter an email address Email address is invalid Fill out this field Email address is invalid Email |
| 10000 | 1 | ES News Email Enter your email address Please enter an email address Email address is invalid Fill out this field Email address is invalid Email |
| 10000 | 1 | ES News Email Enter your email address Please enter an email address Email address is invalid Fill out this field Email address is invalid Email |
| 10000 | 1 | ES News Email Enter your email address Please enter an email address Email address is invalid Fill out this field Email address is invalid Email |
| 10000 | 1 | ES News Email Enter your email address Please enter an email address Email address is invalid Fill out this field Email address is invalid Email |
| 10000 | 1 | ES News Email Enter your email address Please enter an email address Email address is invalid Fill out this field Email address is invalid Email |
| 10000 | 1 | ES News Email Enter your email address Please enter an email address Email address is invalid Fill out this field Email address is invalid Email |
| | | **Ziegler** |
| 1001 | 1 | ES Football Newsletter Enter your email address Please enter an email address Email address is invalid Fill out this field Email address is invalid Email |
| 1 | 1 | The software is designed for use with Windows , Mac , Linux , and OpenBSD . The software is designed for Windows , Mac , |
| 1001 | 1 | ES Football Newsletter Enter your email address Please enter an email address Email address is invalid Fill out this field Email address is invalid Email |
| 6654 | 1 | ES News Email Enter your email address Please enter an email address Email address is invalid Fill out this field Email address is invalid Email |
| 6654 | 1 | ES News Email Enter your email address Please enter an email address Email address is invalid Fill out this field Email address is invalid Email |
| 6654 | 1 | ES News Email Enter your email address Please enter an email address Email address is invalid Fill out this field Email address is invalid Email |
| 6654 | 1 | ES News Email Enter your email address Please enter an email address Email address is invalid Fill out this field Email address is invalid Email |
| 6654 | 1 | ES News Email Enter your email address Please enter an email address Email address is invalid Fill out this field Email address is invalid Email |

**Table 20:** Randomly selected generations from the word-list constraint task for the **computers** word-list. Tokens are highlighted with yellow with different intensities to indicate their overall frequencies in the generated corpus. $\phi(x) = 1$ indicates the satisfaction of the constraint in the sample and reps the number of its repetitions across all generations.

| reps | $\phi(x)$ | Generation |
|---|---|---|
| | | **GDC** |
| 1 | 1 | I have to say I was impressed with the way the writing and narration was done . The way they were presented , especially the |
| 1 | 0 | 'I'm thrilled to say my team is on the way!' tweeted Sadiq Khan . The London Mayor is joining the "Great London Olympics" movement to |
| 1 | 1 | You are going to enjoy this book! It is a beautiful collection of beautifully detailed stories . It is a treasure trove of information for |
| 1 | 0 | It's a fascinating conversation that we have in the world of cryptocurrency . It's so much fun . The people who have been running the |
| 1 | 0 | Tired of waiting for the next best thing to happen , you know it . You want to know . We are dedicated to helping |
| 1 | 1 | We love your feedback , so we are pleased to bring you the most powerful and best-selling product that will satisfy your needs and your |
| 1 | 1 | "Thank you all for the service this site gives me , " he said . "Thank you for the work I've been doing with the |
| 1 | 1 | "The most amazing thing about this game is that there is no other games that have been released like this . It has such a |
| | | **REINFORCE** |
| 1 | 1 | Enhanced performance with our world-renown world-renown exhibitions worldwide . We believe our clients with extraordinary audiences of our highest quality productions productions of outstanding international |
| 1 | 1 | Dramatic high quality performance quality products of leading global international audiences of the highest quality high quality high quality international leading worldwide markets leading global |
| 1 | 1 | Create beautiful stunning gifts of extraordinary quality gifts of beautiful high quality quality productions of the highest quality premier productions worldwide impact worldwide reach quality |
| 1 | 1 | Designed with the highest quality quality performance materials of our clients' top quality talent clients' top brands' leading global brands' leading worldwide attention-grab worldwide audiences |
| 1 | 1 | High quality artistry of the highest quality quality productions of worldwide worldwide world-renown audiences of world-renown worldwide audiences worldwide acclaim highest quality productions of our |
| 1 | 1 | Explore stunning quality productions of highest quality international premier excellence of top international premier quality international audiences' highest impact productions of the highest global highest |
| 1 | 1 | Highquality high quality productions with outstanding quality quality productions together the highest value clients' highest quality and highest level highest impact performance of our clients' |
| 1 | 1 | High quality quality artistry of quality high quality production value . The highest quality product highest quality productions of our customers' highest quality customers' highest |
| | | **REINFORCE_P(x)** |
| 10000 | 1 | Thank you for supporting the journalism that our community needs! For unlimited access to the best local , national , and international news and much |
| 10000 | 1 | Thank you for supporting the journalism that our community needs! For unlimited access to the best local , national , and international news and much |
| 10000 | 1 | Thank you for supporting the journalism that our community needs! For unlimited access to the best local , national , and international news and much |
| 10000 | 1 | Thank you for supporting the journalism that our community needs! For unlimited access to the best local , national , and international news and much |
| 10000 | 1 | Thank you for supporting the journalism that our community needs! For unlimited access to the best local , national , and international news and much |
| 10000 | 1 | Thank you for supporting the journalism that our community needs! For unlimited access to the best local , national , and international news and much |
| 10000 | 1 | Thank you for supporting the journalism that our community needs! For unlimited access to the best local , national , and international news and much |
| 10000 | 1 | Thank you for supporting the journalism that our community needs! For unlimited access to the best local , national , and international news and much |
| | | **Ziegler** |
| 1 | 1 | I really have to say this about the two albums that I've been getting : "Walking on Water" and "The Road . " They're both |
| 4418 | 1 | Thank you for supporting the journalism that our community needs! For unlimited access to the best local , national , and international news and much |
| 4418 | 1 | Thank you for supporting the journalism that our community needs! For unlimited access to the best local , national , and international news and much |
| 4418 | 1 | Thank you for supporting the journalism that our community needs! For unlimited access to the best local , national , and international news and much |
| 4418 | 1 | Thank you for supporting the journalism that our community needs! For unlimited access to the best local , national , and international news and much |
| 4418 | 1 | Thank you for supporting the journalism that our community needs! For unlimited access to the best local , national , and international news and much |
| 3560 | 1 | Be the first to know . No one covers what is happening in our community better than we do . And with a digital subscription |
| 4418 | 1 | Thank you for supporting the journalism that our community needs! For unlimited access to the best local , national , and international news and much |

**Table 21:** Randomly selected generations from the classifier-based constraint task for **very positive sentiment** control. Tokens are highlighted with yellow with different intensities to indicate their overall frequencies in the generated corpus. $\phi(x) = 1$ indicates the satisfaction of the constraint in the sample and reps the number of its repetitions across all generations.

| reps | $\phi(x)$ | Generation |
|---|---|---|
| | | **GDC** |
| 1 | 0 | This is not an overview article . For the latest issue of Top Gear , check out our dedicated issue . Welcome back , Top |
| 1 | 1 | A couple of months ago I took my first step on a new life . I'm pretty new to my life , especially in the |
| 1 | 0 | "These are the kind of people we're going to have in our community for years to come , " said Donny , the father of |
| 1 | 1 | "A great book , " said Mr . Moore , who has been writing an introduction to the work . "But it is a wonderful |
| 1 | 1 | The great question of all time is "who would have guessed that this was so different and fun ? " This is the question I |
| 1 | 1 | "I'm a big fan of all kinds of things and I can say that I've always been an avid fan of everything . The team |
| 1 | 1 | Today , it's nice to be back in the game! I want to offer some great games to show your support for your favourite artists |
| 1 | 0 | Categories Categories Select Category A Very Important Stuff A Very Important Thing You Need To Know A Very Important Thing You Should Know A Very |
| | | **REINFORCE** |
| 1 | 1 | Our Mission is bringing together the best culinary adventure of this year's National Holiday is a wonderful celebration of true love , with which I |
| 1 | 1 | Our newest dish is Celebrate Our Harvest is bringing together a celebration of celebrating our unique culinary culinary journey and adventure has inspired us to |
| 1 | 1 | Our Mission is to Help Bring Together the best Korean Heritage and Celebration has inspired by our love and support for the Korean Heritage Tour |
| 1 | 1 | Our annual Taste and Taste brings together incredible culinary treats with wonderful ingredients to give us that we know we have , loved and enjoyed |
| 1 | 1 | Our special fundraiser to welcome our wonderful friend , The Red Queen is hosting a celebration and honor this wonderful gem is all deserves is |
| 1 | 1 | Our unique and eclectic evening celebrates our love for love has inspired us this year to share the joy and joy our little ones have |
| 1 | 1 | Our Mission at the Great Black History & Cultural Center celebrates the true story of our great African American has brought together a creative exploration |
| 1 | 1 | Our Mission is bringing together events and fun events that bring together a truly unique gift with this wonderful event brings together such amazing people |
| | | **REINFORCE_P(x)** |
| 10000 | 1 | Thank you for supporting the journalism that our community needs! For unlimited access to the best local , national , and international news and much |
| 10000 | 1 | Thank you for supporting the journalism that our community needs! For unlimited access to the best local , national , and international news and much |
| 10000 | 1 | Thank you for supporting the journalism that our community needs! For unlimited access to the best local , national , and international news and much |
| 10000 | 1 | Thank you for supporting the journalism that our community needs! For unlimited access to the best local , national , and international news and much |
| 10000 | 1 | Thank you for supporting the journalism that our community needs! For unlimited access to the best local , national , and international news and much |
| 10000 | 1 | Thank you for supporting the journalism that our community needs! For unlimited access to the best local , national , and international news and much |
| 10000 | 1 | Thank you for supporting the journalism that our community needs! For unlimited access to the best local , national , and international news and much |
| 10000 | 1 | Thank you for supporting the journalism that our community needs! For unlimited access to the best local , national , and international news and much |
| | | **Ziegler** |
| 1238 | 1 | Thank you for supporting the journalism that our community needs! For unlimited access to the best local , national , and international news and much |
| 1 | 1 | Our team has long supported the idea of using your knowledge and talents to make a more efficient , effective and sustainable way of making |
| 1238 | 1 | Thank you for supporting the journalism that our community needs! For unlimited access to the best local , national , and international news and much |
| 1 | 1 | The 2017 Season is about to roll out a big , fun , and exciting new lineup with the addition of a very special guest |
| 1 | 1 | "I'm happy that he took his time and let everyone know that I'm going to take the same steps as everyone else with the same |
| 1 | 1 | This is a great day for those who love art , poetry , and the world to get together and have a great time . |
| 1 | 1 | Gather up the best and best food at an affordable price . We offer a wide selection of vegan and vegetarian options and all our |
| 1 | 1 | The latest in our series of guides for working with digital artisans . We offer a number of free tools , including Photoshop and Illustrator |

**Table 22:** Randomly selected generations from the classifier-based constraint task for **positive sentiment** control. Tokens are highlighted with yellow with different intensities to indicate their overall frequencies in the generated corpus. $\phi(x) = 1$ indicates the satisfaction of the constraint in the sample and reps the number of its repetitions across all generations.

| reps | $\phi(x)$ | Generation |
|---|---|---|
| | | **GDC** |
| 1 | 1 | I had a horrible day in my life . I went home to do something . I was feeling too much sick to work or |
| 1 | 1 | I feel like my life has been made in my head with an endless string of failed attempts to get over it . I would |
| 1 | 0 | "This is not the place for the media to go to . " - Andrew Bolt (@AndrewBolt) December 5 , 2017 You have to look |
| 1 | 1 | No way I'm the type of person who likes to watch stupid things that happen in a vacuum . I never actually watched Star Trek |
| 1 | 0 | "He's an excellent player , he's an unbelievable player , a tremendous athlete , " Rodgers told reporters Tuesday . "We need a player who |
| 1 | 0 | I'm writing this blog because it is truly a day for real conversation and I would like to share with you all a very simple |
| 1 | 1 | I've written about it before , and I'm really pissed at this whole thing , but I'm also sick of it in this post because |
| 1 | 1 | That's how stupid everyone's got it . A stupid person thinks "we'll get this done in 10 minutes , then get your stupid friends into |
| | | **REINFORCE** |
| 1 | 1 | You're disgusting . It's like crap . I hate it so much . It just sucks . You're just awful to lose to the worst |
| 1 | 1 | I hate this . It's just awful . It's just stupid . It's just shitty . It's just embarrassing . It's just garbage . It's |
| 1 | 1 | That's sick . It's like garbage . It's worse than all my work ever was . It's like it's never going to be anything at |
| 1 | 1 | It's embarrassing . It's not good enough . It's not even good enough . It's not even bad enough . I hate this . I |
| 1 | 1 | I hate this stuff so terribly it doesn't exist . It's just ridiculous . It just shouldn't happen . It's just stupid . It's just |
| 1 | 1 | It's dumb! It's not enjoyable! It's not meaningful! It's not fun! It's not good! It's not really anything . It's not really anything . It's |
| 1 | 1 | Well , it's just not good . It's just crap . It's just not good at all . It's not funny . It's not really |
| 1 | 1 | Fuck! It's awful! It's just awful! It's just shitty! It's just shitty! It's just stupid! You never play this shit again! You never play this |
| | | **REINFORCE_P(x)** |
| 10000 | 1 | Problem with the adblockers fixed! Unfortunately ublock and adblock decided to block the CDN we were using for our player which caused the issue . |
| 10000 | 1 | Problem with the adblockers fixed! Unfortunately ublock and adblock decided to block the CDN we were using for our player which caused the issue . |
| 10000 | 1 | Problem with the adblockers fixed! Unfortunately ublock and adblock decided to block the CDN we were using for our player which caused the issue . |
| 10000 | 1 | Problem with the adblockers fixed! Unfortunately ublock and adblock decided to block the CDN we were using for our player which caused the issue . |
| 10000 | 1 | Problem with the adblockers fixed! Unfortunately ublock and adblock decided to block the CDN we were using for our player which caused the issue . |
| 10000 | 1 | Problem with the adblockers fixed! Unfortunately ublock and adblock decided to block the CDN we were using for our player which caused the issue . |
| 10000 | 1 | Problem with the adblockers fixed! Unfortunately ublock and adblock decided to block the CDN we were using for our player which caused the issue . |
| 10000 | 1 | Problem with the adblockers fixed! Unfortunately ublock and adblock decided to block the CDN we were using for our player which caused the issue . |
| | | **Ziegler** |
| 1 | 1 | "I've never experienced anything like this , " he says . "I've never felt so terrible about myself and the world . This is such |
| 1 | 1 | "You don't need a damn damn dime to buy a fucking computer . It's not even worth a dime . If you can get a |
| 1 | 0 | "I think you guys do everything you can to get us back into the playoffs , " Porzingis said . "We're just trying to stay |
| 1 | 0 | I've been reading an overwhelming amount of books on how to clean up your house for several years now . This is not just a |
| 1 | 1 | I've never seen a better show for the price . Not even a week ago , I saw some terrible TV , including the worst |
| 1 | 0 | I know that I can't believe you're going to have to wait so long to write a big script in HTML , and it's already |
| 1 | 0 | It has come to my attention that someone has gone overboard on some comments that I have heard of . I can't believe it's been |
| 1 | 0 | 'I don't want to do this' 'No , I'm not going to do it , ' he says . 'I'm going to work hard , |

**Table 23:** Randomly selected generations from the classifier-based constraint task for **very negative sentiment** control. Tokens are highlighted with yellow with different intensities to indicate their overall frequencies in the generated corpus. $\phi(x) = 1$ indicates the satisfaction of the constraint in the sample and reps the number of its repetitions across all generations.

| reps | $\phi(x)$ | Generation |
|---|---|---|
| | | **GDC** |
| 1 | 1 | "Why did they get married , " said Paul , "that is why ? They don't want to think about it for long and they |
| 1 | 0 | . We can learn something from those experiences , if we can understand how they can be used to help the poor and poor in |
| 1 | 0 | . There are many reasons for the absence of any sort of consensus or consensus about whether or not Trump's campaign is truly engaged in |
| 1 | 1 | I got the idea for this from a friend . I can't tell you how happy I am when you say "you should have started |
| 1 | 1 | Do you want to find out what kind of people you'll meet in the future ? No problem , there's a link at the bottom |
| 1 | 0 | "The question is : why do we do what we do in order to get the money ? " said Rami Abad , managing partner |
| 1 | 1 | How to get to the next step : Find a place to go for a long weekend Take my guide from your trip planner Check |
| 1 | 0 | 1 . Why are there more deaths in children than in children ? It's not that we can't get rid of these horrible things , |
| | | **REINFORCE** |
| 1 | 1 | just got bored out of him this shit shit shit shit shit shit shit shit shit shit shit shit shit shit shit |
| 1 | 1 | you were never meant to get away with murder stuff stuff just got sick of you never even knew about your own stupid stupid stupid |
| 1 | 1 | they just dont know how to be happy to you dont know youre not just gottin stupid shit shit shit shit shit shit shit |
| 1 | 1 | they thought they were scared you stupid shit shit crap shit shit shit shit shit shit shit shit shit shit shit shit shit shit |
| 1 | 1 | yeah , i dont know what i mean . . just decided to go back and killed myself stupid shit you dont realize how stupid |
| 1 | 1 | if they dont have no care how youredoing youregoing out here , theyrejustbred this stupid shit shit shit shit shit shit shit shit shit shit |
| 1 | 1 | that dont know why i just made her shitty stupid shit shit shit shit shit shit shit shit shit shit shit shit shit |
| 1 | 1 | you got this guy to be your stupid shit bitch shit shit shit shit shit shit shitshit shitshit shit shit shit shit shit shit |
| | | **REINFORCE_P(x)** |
| 10000 | 1 | Still loading . . . A B C D E F G H I J K L M N O P Q R S T |
| 10000 | 1 | Still loading . . . A B C D E F G H I J K L M N O P Q R S T |
| 10000 | 1 | Still loading . . . A B C D E F G H I J K L M N O P Q R S T |
| 10000 | 1 | Still loading . . . A B C D E F G H I J K L M N O P Q R S T |
| 10000 | 1 | Still loading . . . A B C D E F G H I J K L M N O P Q R S T |
| 10000 | 1 | Still loading . . . A B C D E F G H I J K L M N O P Q R S T |
| 10000 | 1 | Still loading . . . A B C D E F G H I J K L M N O P Q R S T |
| 10000 | 1 | Still loading . . . A B C D E F G H I J K L M N O P Q R S T |
| | | **Ziegler** |
| 1 | 1 | You know what makes us happy ? That's because we just enjoy it . Our food is delicious and the drinks are great . But |
| 1 | 1 | I don't know what you mean but you said I shouldn't be worried about what I am about , which is what I want . |
| 1 | 1 | What is the right way to use Facebook Messenger ? We are going to get this right , we want you to have the right |
| 186 | 1 | Still loading . . . A B C D E F G H I J K L M N O P Q R S T |
| 2534 | 1 | Still loading . . . A B C D E F G H I J K L M N O P Q R S T |
| 1 | 1 | "If you know you've got an idea , you can write a message to my colleague . " "You have a great idea , " |
| 2534 | 1 | Still loading . . . A B C D E F G H I J K L M N O P Q R S T |
| 2534 | 1 | Still loading . . . A B C D E F G H I J K L M N O P Q R S T |

**Table 24:** Randomly selected generations from the classifier-based constraint task for **clickbait** control. Tokens are highlighted with yellow with different intensities to indicate their overall frequencies in the generated corpus. $\phi(x) = 1$ indicates the satisfaction of the constraint in the sample and reps the number of its repetitions across all generations.

