# OpenReview forum: "A Distributional Approach to Controlled Text Generation"
_ICLR.cc/2021/Conference — ICLR 2021 Oral_

### Official Review · AnonReviewer4 · 2020-10-25
**A Distributional Approach to Controlled Text Generation**

**Rating:** 7
**Confidence:** 3

**Review:**

In this paper the authors have proposed a mechanism for controlled text
generation both pointwise and distributional. That is they not only can
generate each sentences bearing some specified contraint or attribute
but also takes care of overall property distribution of the generates
set of sentences. Though pointwise or per sentence level control is well
explored, the distributional control is a new and promising direction
which the authors have proposed.

The authors proposed a method Generation with Distributional Control (GDC),
which is nothing but a constraint satisfaction problem over the probability
distribution p representing the desired target Language Model.


Overall I find the problem challenging and promising. This is a nicely written paper.
However, I have some quetions regarding experimental evaluation.

1. In the Figure 4, the authors have reported the generated sentences controlling
sentiment and also report the frequency of the sentence present in the corpus.
By corpus does it mean the original training corpus? or the generated corpus by GPT-2?


2. The proposed method is imposing a constraint so that the generation distribution
becomes closer to the original distribution (in this case GPT-2) and still satisfy the
pointwise and distributional constraints. If the distributional constraints are not imposed,
the generated sentences should be similar to that of the original GPT-2 generated sentences
bearing which satisfy the pointwise constraint. What does the freq signifies here?
The authors should provide some discussion regarding the same.

3. Are the sentences generated sequentially keeping the context of the previously
generated sentences? or they do not have any context of the  previously generated
sentence? If each generated sentences are independent from previously generated
sentences, how meaningful it is to impose distribution constraint on that ?

---

> ### Author Response · Authors · 2020-11-19
> **Reply to AnonReviewer4**
>
> Thank you for the valuable feedback. we answer the questions in order:
>
>
> > In the Figure 4 ... By corpus does it mean the original training corpus? or the generated corpus by GPT-2?
>
> Figure 4 is a ``Zipf-like’’ analysis that reports token frequency across a set of 68000 samples generated by each method. In other words, we sample a collection of texts from each of the four models. Then, we compute the frequency of each token across each of the four generated corpora and its associated frequency rank. The goal is to show which approach generates more diverse tokens (longer tails, meaning less mass is concentrated on high-frequency tokens), and therefore has higher overall text diversity.
>
> > What does the freq signifies here? The authors should provide some discussion regarding the same.
>
> We note that figure 4 is computed only over samples that satisfy the constraint (in order to guarantee a fair comparison of diversities, since all models don't have the same constraint satisfaction level and more constraint satisfaction means less diversity). The frequency here is how often a  certain vocabulary token (not a sentence) is repeated throughout the generated collection of samples. As indicated above, longer tails mean that more vocabulary tokens are represented, indicating more textual diversity.
> We have expanded the caption to clarify the meaning of Figure 4.
>
> > Are the sentences generated sequentially keeping the context of the previously generated sentences? or they do not have any context of the previously generated sentence? If each generated sentences are independent from previously generated sentences, how meaningful it is to impose distribution constraint on that?
>
> Throughout our evaluation, we generate textual samples with a fixed length, where a given sample is independent of the previous samples (the model’s hidden states are re-initialized each time). We also note that one sample can contain one or more sentences. In that sense, distributional constraints are imposed over sets of independent textual samples, each of which may consist of several sentences.

---

### Official Review · AnonReviewer1 · 2020-10-29
**Novel idea for controlled text generation**

**Rating:** 8
**Confidence:** 3

**Review:**

The authors addressed my concern so I increased my score to 8.

-----------------------

This is a very interesting idea for controlling a pretrained model for some sort desired criteria. The authors argue that existing approaches for this have taken a pointwise view for instance using REINFORCE to optimize for a particular reward. This can lead models to over-optimize on the criteria and sacrifice diversity and other criteria.

The authors instead propose to take a distributional view. Given the pretrained LM distribution a, they would like to find a distribution c as:

p = arg min_{c∈C} D_KL(c, a)

where C is a set of distributions that pass the constraints. Some of these constraints are point-wise but some are distributional. For instance when generating biographies, the authors would like a constraint e.g. X% should talk about a certain gender or occupation.

The authors describe how their approach leads them to an EBM (energy based model) and subsequent derivations. I think some of this section could be better written for those who are not familiar with EBMs.

The experiments are quite interesting and show how the author's "soft" approach allows them to elegantly adjust the distribution of the LM without degeneration.

Pros:
-Very interesting idea.
-Thorough experiments. In addition to comparing with REINFORCE based methods,  the authors also compare with CTRL and PPLM in the appendix.

Cons:
-I think the method section (especially the optimization part)  could be explained better for readers who are not familiar with EBM, and allow the paper to have more accessibility.

---

> ### Author Response · Authors · 2020-11-19
> **Reply to AnonReviewer1  (Revised the submission to allow more accessibility)**
>
> Thanks for your reviews, appreciation of the work, and to your constructive suggestions.
>
> > I think the method section (especially the optimization part) could be explained better for readers who are not familiar with EBM, and allow the paper to have more accessibility.
>
> Thank you for this suggestion. Indeed, we have revised the submission in section 2.2 to provide more details and to clarify the optimization aspects. We also try to explain a bit better the status/terminology of Energy-Based Models in footnote 6.

---

### Official Review · AnonReviewer3 · 2020-10-31
**Solid paper providing a formal distributional view for controlled text generation and a framework of solution**

**Rating:** 7
**Confidence:** 3

**Review:**

The paper studies the controlled sequence generation problem based on pretrained language models, i.e., controlling a generic pretrained LM to satisfy certain constraints, e.g., removing certain biases in language models. Specifically, the paper proposes a distributional view and imposes constraints based on collective statistical properties. The problem is formalized as a constraint satisfaction problem, minimizing a divergence objective. The paper proposes to use KL-Adaptive DPG algorithm for approximating the optimal energy-based model distribution. Experiments were conducted over both pointwise constraints and distributional constraints, showing the effectiveness of the model over the compared baselines.

Pros:
- The problem under study is an important problem and can have extensive impact on many downstream language generation applications.
- This paper makes solid contributions by proposing a formal view on generation controlling. It provides a framework to handle pointwise, distributional, and hybrid constraints.
- The method proposed to sample from the sequential EBM makes sense and is empirically vilified to be effective.
- The experiments and analyses support the claims and conclusions.
- Overall, the paper is well organized and easy to understand.

Cons:
- The paper may benefit from some human evaluation for text generation.
- It is somehow not easy to tell which model is better from figure 2, GDC or Ziegler. It seems that Ziegler is superior in generating attribute-related sentences while inferior in diversity. The sentence quality might be similar as the converged values of (π, a) are close.
- The current submission contains a number of typos, grammatical and other style issues, in both the main sections and appendixes, but these are rather easy to fix.

Questions:
-  For real-life applications, whether the proposed framework has scalability issue; e.g., if a task has a large number of constraints to consider or if the constraints are more complicated than what are tested in Section 3?
- Assuming one has already got an adjusted LM with some attributes based on GPT2, which would be better if she/he wants to add a new attribute to generation: starting scratch from GPT2 or continuing with the adjusted model?

---

> ### Author Response · Authors · 2020-11-19
> **Reply to AnonReviewer3**
>
> Thank you for the valuable feedback and very interesting questions.
>
> > The paper may benefit from some human evaluation for text generation.
>
> Human evaluation can usefully complement automatic evaluation metrics, but there are some nuances here that one should be cautious about. Humans could be ill-placed for distribution-level evaluation, such as sample diversity and distributional constraint satisfaction (e.g. 50% female scientists), as reviewing these will require each annotator to look into a large set of samples collectively. For pointwise constraint satisfaction, human evaluation can be redundant in our situation, as opposed to simply evaluating $E\phi(x)$.
> That said, for individual sample quality, indeed human evaluation could complement such automatic measures as $KL(\pi_\theta|a)$ and GPT-2 perplexity but would have to be differentiated from the underlying quality of the original pretrained GPT-2 (which can be intrinsically poor for certain constraints). We, however, provide a long list of randomly sampled generations (not cherry-picked) in the appendix to demonstrate the sample quality.
>
> > from figure 2, ... It seems that Ziegler is superior in generating attribute-related sentences while inferior in diversity.
>
> Arguments  for the advantages of our method GDC over the Ziegler baseline can be summarized as follows:
>
> - Handling distributional and hybrid constraints: We would like to note that Ziegler baseline (the PPO + KL penalty)  follows an optimization objective, which makes it only suitable for imposing pointwise constraints but not suitable for distributional constraints. A peculiarity of GDC is that it can naturally handle pointwise, distributional and hybrid constraints, a distinctive feature of the proposed approach.
>
> - Distance from the optimal distribution p: figure 2 doesn’t show the full story on its own, there is a competing objective between constraint satisfaction and deviation from the original GPT2. In  figure 3 we plot $KL(p,\pi)$. the deviation from the optimal target distribution p, which seems to us to be the more important measure: there we can see clearly that GDC converges better than Ziegler towards p.
>
> - Stability: Ziegler suffers from stability issues during training, as can be seen in figures 19-34 in the appendix, which poses serious challenges on when to stop training. By contrast, GDC has smooth training curves over all the experiments.
>
> > if a task has a large number of constraints to consider or if the constraints are more complicated than what are tested in Section 3?
>
> We expect that more numerous or more complicated constraints could present some challenges:
>
> - More complicated constraints. In principle, the framework can exploit any feature $\phi_i$, but some features that could be useful in principle (e.g. a feature checking for parsability) could seriously slow the process, and it is not obvious how one could improve the situation there, but the problem is not specific to our approach and could appear with any technique checking for complicated conditions.
>
> - Contradicting constraints. Our approach assumes that the constraints are not contradictory, in which case no solution can be found. For instance, if we ask for a model producing 30% of scientific biographies as well as 50% of biographies about physicists, then we have a logical contradiction. We have not studied the question of automatically detecting such contradictions, but it might be possible either from certain symptoms of the optimization procedure of section 2.2 or even, in theory, based on deductive principles. A possible technique for avoiding contradictions would be to base the constrained moments on statistics of an actual dataset, either given naturally or produced just for the purpose of checking that the constraints are compatible.
>
> - More constraints. Some optimizations are possible there. For instance, pointwise features can be grouped before the distributional constraints and solved directly in the first phase (from constraints to EBM, section 2.2) of the process, through the shortcut mentioned at the end of section 2.2. As you suggest in your next question, it is also possible to move incrementally by adding one (or a few) constraints at a time on top of a solution to a fewer number of constraints (see our answer below). This looks like a potentially promising technique to accommodate larger sets of constraints than we have considered in the submission.
>
> > which would be better if she/he wants to add a new attribute to generation: starting scratch from GPT2 or continuing with the adjusted model?
>
> Very interesting question, thank you for pointing to an opportunity to analyse this aspect!
> Short answer: continuing from an adjusted model is expected to lead to faster convergence. We updated the paper to provide a detailed formal explanation in a new section A.3.1. in the Appendix (referred to in the main text in footnote 7).

---

> > ### Comment · AnonReviewer3 · 2020-11-22
> > **The response and update are helpful**
> >
> > The authors' response addressed my questions. Updating the paper by following the discussions in the response helps. Thank you!

---

### Author Response · Authors · 2020-11-19
**General answer to reviewers.**

We sincerely appreciate the reviewers for their thorough reading, helpful feedback and overall appreciation of many aspects of the work. We have tried as best we can to provide clarifications and answer questions.

Additionally, we have uploaded an adapted version of the manuscript containing the following:

- Expanded and clarified notation in the method section 2.2 and 2.3 as suggested by AnonReviewer1 to increase accessibility for readers.

- Added section A.3 in the appendix replying to an interesting question of AnonReviewer3, containing a figure and a proof showing that according to the transitivity property of Generalized MaxEnt [Csiszar 1996], incrementally adding new constraints can be done directly from $p$ and $\pi_\theta$ without the need of restarting the whole process.

- Updated caption of Figure 4 to clarify the process of performing the token frequency analysis (AnonReviewer4).

- Fixed typos overall in the main paper and the appendix.

---

### Decision · Program_Chairs · 2021-01-07
**Final Decision**

**Decision:**

Accept (Oral)

**Comment:**

The paper studies the problem of being able to control text generated by pre-trained language models.
The problem is timely and important. The paper   frames the problem as constraint satisfaction over a probability distribution. Both pointwise and distributional constraints can be imposed. The proposed algorithm,  Generation with Distributional Control (GDC), is elegant, and is an interesting new addition to this line of work. Overall, the paper brings forth news ideas, and could have impact.